# Improved upper-ocean thermo-dynamical structure modeling with combined effects of surface waves and M₂ internal tides on vertical mixing: a case study for the Indian Ocean

Zhanpeng Zhuang[1, 2, 3], Quanan Zheng [4], Yongzeng Yang[1,2,3], Zhenya Song[1,2,3], Yeli Yuan[1,2,3], Chaojie Zhou[5], Xinhua Zhao[6], Ting Zhang[1], Jing Xie[7]

[1]First Institute of Oceanography, and Key Laboratory of Marine Science and Numerical Modeling, Ministry of Natural Resources, Qingdao 266061, China

[2]Laboratory for Regional Oceanography and Numerical Modeling, Pilot National Laboratory for Marine Science and Technology, Qingdao 266237, China

[3]Shandong Key Laboratory of Marine Science and Numerical Modeling, Qingdao 266061, China

[4]Department of Atmospheric and Oceanic Science, University of Maryland, College Park, Maryland 20740-20742, USA

[5]Hainan Institute of Zhejiang University, Yazhou Bay Science and Technology City, Sanya 572025, China

[6]Jiangsu Marine Resources Development Research Institute, Jiangsu Ocean University, Lianyungang 222005, China

[7]School of Information and Control Engineering, Qingdao University of Technology, Qingdao 266520, China

*Correspondence to*: Zhanpeng Zhuang (zhuangzp@fio.org.cn)

**Abstract.** The surface waves and internal tides have great contribution to the vertical mixing processes in the upper ocean. In this study, three mixing schemes, including the non-breaking surface-wave-generated turbulent mixing, the mixing induced by the wave transport flux residue, and the internal-tide-generated turbulent mixing, are introduced to study the effects the surface waves and the internal tides on the vertical mixing. The three schemes are jointly incorporated into the Marine Science and Numerical Modeling (MASNUM) ocean circulation model as a part of the vertical diffusive terms, which are calculated by the surface wave parameters simulated from the MASNUM wave model and the surface amplitudes of the mode-1 M₂ internal tides extracted from the satellite altimetry data using a two-dimensional plane wave fit method. The effects of the mixing schemes on the Indian Ocean modeling are tested by five climatological experiments. The surface waves and internal tides lead to enhance the vertical mixing processes in the sea surface and ocean interior, respectively. The combination of the mixing schemes is able to strengthen the vertical water exchange and draw more water from the sea surface to the ocean interior. The simulated results gain significant improvement in the thermal structure, the mixed layer depth and the surface currents if the three schemes are all adopted.

## 1 Introduction

Turbulence in the ocean is hard to be described superficially and characterized dynamically. Fortunately, in recent years a great progress in understanding the turbulence has been achieved by a combination of experiments, simulations, and theories (Baumert et al., 2005; Umlauf and Burchard, 2020). Turbulence has great contribution to the vertical mixing processes in the upper ocean, which is important for regulating the sea surface temperature (SST) and the thermal structure. Accurate parameterization of the vertical mixing process is the key for the ocean general circulation models (OGCMs) to simulate the realistic ocean dynamic and thermal environment. However, the factors influencing the vertical mixing in the upper ocean still remain unclearly, so that there are substantial biases in the simulated SST, mixed layer depth (MLD) and dynamic quantities within the ocean interior such as potential vorticity, temperature and salinity for the most ocean models (Ezer, 2000; Qiao et al., 2010; Wang et al., 2019; Song et al., 2020; Zhuang et al., 2020).

In the sea surface layer, turbulence can be generated by wind and surface waves (Agrawal et al., 1992; Qiao et al., 2004; Babanin, 2017), Langmuir circulation (Li and Garrett, 1997; Li and Fox-Kemper, 2017; Yu et al., 2018), and surface cooling at night (Shay and Gregg, 1986). Among them wind energy input to the surface waves is estimated as 60 - 70 TW (Wang and Huang, 2004) much greater than all other mechanical energy sources (Wunsch and Ferrari, 2004). Most of the wave energy is dissipated locally through wave breaking (Donelan, 1998), and enhances the turbulent mixing near the sea surface. Meanwhile, previous studies indicated that the non-breaking surface waves (NBSWs) enable to affect the depths much greater than that of wave breaking (Huang et al., 2011), and even penetrate into the sub-thermocline ocean (Babanin and Haus, 2009; Wang et al., 2019). Despite the parameterization schemes of the wave-induced mixing have been developed and adopted in OGCMs, it still remains controversial about the effects of wave-induced turbulence mixing in the upper ocean (Huang and Qiao, 2010; Kantha et al., 2014).

Generally, the effects of the surface waves on the upper-ocean dynamic processes include the momentum transport by the Stokes drift through the "Coriolis-Stokes" forcing (Li et al., 2008; Zhang et al., 2014; Wu et al., 2019), enhanced near-surface mixing by wave breaking (Donelan, 1998), and modulation of the surface wind stress by wave roughness (Craig and Banner, 1994; Sullivan et al., 2007; Yang et al., 2009). The "Coriolis-Stokes" forcing induced by the surface waves has a positive impact on the simulated current profile in the whole wind-driven layer, since the ocean Ekman transport and the Ekman spiral profile are modified (Polton et al., 2005; Wu et al., 2019). A non-breaking wave-induced mixing scheme for shear driven turbulence was proposed, in which the viscosity and diffusivity can be calculated as the functions of the Stokes drift (Huang and Qiao, 2010; Qiao et al., 2010). The turbulent mixing induced by the wave-current interaction occurs in the subsurface layers due to the Langmuir turbulence, which can improve the ocean circulation modeling (Huang and Qiao, 2010; Qiao et al., 2016; Yu et al., 2018). For the small- and meso-scale motions, the effects of the surface waves are also significant by modifying the surface current gradient variability and the eddy transport when the Turbulent Langmuir number is small (Jayne and Marotzke, 2002; Romero et al., 2021), and the effects will become larger when the model resolution increases (Hypolite et al., 2021). On the whole, the effects of the NBSWs on the dynamical structure are not negligible.

In the bulk of the stratified ocean interior, it is believed that the internal waves are one of the dominant sources to induce turbulent mixing (Munk and Wunsch, 1998; Wunsch and Ferrari, 2004). The total internal wave energy input was estimated as $2.1\pm0.7$ TW (Kunze, 2017) with most of the uncertainty in the observations of the near-inertial waves produced by winds (Alford, 2001; Furuichi et al., 2008) and the internal lee-waves (Scott et al., 2011; Wright et al., 2014). Based on the internal wave-wave interaction theory, parameterization schemes for internal-wave-induced turbulence mixing are proposed in terms of shear and/or strain (e.g. Gregg and Kunze, 1991; Gregg et al., 2003; Kunze et al., 2006; Huussen et al., 2012). However, the usefulness of the parameterizations, which are put forward based on particular data set, should be severely limited (Polzin et al., 2014). The development of the dynamical interpretation and parameterization of the internal-wave-induced turbulence mixing is still an ongoing process. Meanwhile, the internal tides (ITs) are essentially the internal waves generated by barotropic tidal flow with the tidal frequency. Previous investigators have demonstrated that the internal tides are important and even dominant in the energy budgets of the ocean interior (Wunsch and Ferrari, 2004; Zhao et al., 2016). In this study, we analyze the effects of the turbulent mixing generated by the $M_2$ internal tides on the ocean circulation. Actually, the $M_2$ IT, which is one of the main tidal constituents, is chosen to analyze the IT-generated turbulent mixing. There are three main reasons. Firstly, as one of the main tidal constituents (including $M_2$, $S_2$, $N_2$, $O_1$, and $K_1$), the $M_2$ ITs have the largest energy among the semi-diurnal ITs, therefore, the turbulence generated by $M_2$ internal tides should be dominant and typical for this study. Secondly, $M_2$ ITs are ubiquitous in the world oceans, and lose little energy in propagating across its critical latitudes ($28.88°$ S/N) (Zhao et al., 2016). Finally, this study is still preliminary research on the contribution of surface wave- and internal tides-induced vertical mixing in the upper ocean, we just choose one of the main tidal constituents to test the effects. Other constituents such as $S_2$, $N_2$, $O_1$, and $K_1$ will be evaluated in future. The internal tides are extracted from the satellite altimeter data using a two-dimensional plane wave fit method (Zhao et al., 2016; Zhao, 2018).

The internal wave energy in the ocean interior, which generates the turbulence processes and the diapycnal diffusivity (Jayne, 2009; st. Laurent et al., 2012), is redistributed from large- to small-scale motions by the wave-current interactions. The dynamic processes were modulated through shearing and straining actions of the fine-scale internal waves (Gregg and Kunze, 1991; Kunze et al., 2006; Jayne, 2009). As a key mechanism, the subharmonic instability may transfer the energy from the internal tides to the shear-induced turbulent diapycnal mixing (MacKinnon and Gregg, 2005; Pinkel and Sun, 2013). The parameterization of the turbulent mixing induced by the internal waves was introduced into the ocean models and makes the simulated mixing coefficients and dynamic processes, including horizontal currents and meridional overturning circulation, agree better with the Large Eddy Simulation (LES) results or observations than the original schemes (Kunze et al., 2006; Jayne, 2009; Huussen et al., 2012; Shriver et al., 2012). However, the effects of the IT-generated turbulent mixing on the dynamical processes has not been understood clearly.

The Indian Ocean (IO) is the third largest ocean in the world and has an important low latitude connection to the Pacific Ocean through the Indonesian Archipelago (Fig. 1). On one hand, the mean wind pattern of the South Indian Ocean (SIO) is similar to the Atlantic and Pacific Oceans, with westerly winds at high latitude (Southern Ocean) and trade winds at low latitudes; on the other hand, a complex annual cycle associated with the seasonally reversing monsoons is dominant in the

North Indian Ocean (NIO). As a result, the wind waves, which are the prominent feature of the ocean surface, undergo large seasonal variations in the NIO (Kumar et al., 2013; Kumar et al., 2018). Previous investigations showed that the annual and seasonal (during summer monsoon period, i.e. June - September) average significant wave height (SWH) in the NIO ranges from 1.5 - 2.5 m and 3.0 - 3.5 m, respectively, based on the European Centre for Medium-Range Weather Forecasts (ECMWF) ReAnalysis V5 (ERA5) reanalysis product (Anoop et al., 2015). In the SIO, the average SWH between 35° S and 22° S is consistently higher by about 1.5 times that in the NIO because of the higher wind speed (Kumar et al., 2013). Furthermore, based on the satellite altimetry data and high-resolution numerical simulations, a regional map of the internal tides in the IO was constructed by previous studies. The results show that the Madagascar-Mascarene regions, the Bay of Bengal and the Andaman Sea are considered to be hot spots for generation of semidiurnal internal tides (Ansong et al., 2017; Zhao, 2018), while the central IO for diurnal internal tides (Shriver et al., 2012). In summary, all these efforts gave us a strong hint that the surface waves and the internal tides in the IO could not be neglected in the studies of ocean dynamics and modeling. As mentioned above, the NBSW and the IT are two of the key factors for the vertical mixing processes, which are important for the simulated SST, MLD, the meridional overturning circulation and larger-scale property budgets in the IO (Jayne and Marotzke, 2002; Qiao et al., 2010; Huussen et al., 2012; Kumar et al., 2013; Zhuang et al., 2020).

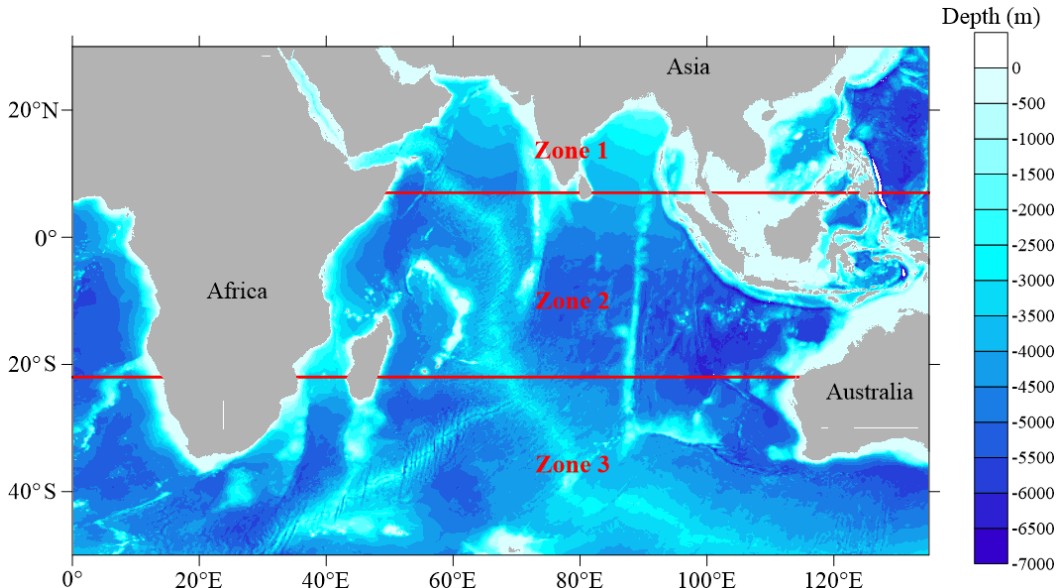

**Figure 1: Bathymetric map (color codes in m) in the Indian Ocean. Red lines (7° N and 22° S) show the zone partition in the present study**

Previous studies, such as Simmons et al. (2004) and Nagai and Hibiya (2015), constructed baroclinic ocean models to compute the energy flux from barotropic tides into internal waves. The Navier-Stokes equations with accurate tidal potential forcing, tidal open boundary conditions and non-hydrostatic approximation were calculated to simulate the generation, development, propagation and dissipation processes of the ITs in high-resolution numerical experiments. The induced turbulent

mixing coefficients then can be estimated in terms of the local dissipation efficiency, the barotropic to baroclinic energy
conversion and the buoyancy frequency. In fact, the estimation of the IT-generated turbulent mixing in these previous studies
was implicit. The simulated internal tides processes will become inaccurate if the temperature and current structure cannot be
modeled accurately. On the contrary, we attempt to derive an analytic and explicit expression of the vertical diffusive terms
induced by the NBSWs and ITs, based on the theory about the turbulence dynamics and the surface and internal wave statistics.
The mixing schemes introduced in this study will be calculated directly in terms of the parameters of the NBSWs and ITs. The
present study just provides another way and preliminary attempt to study the mixing processes induced by the internal tides.
It should be more convenient to improve the simulation further because the mixing schemes are independent with the ocean
model.

In this study, the vertical mixing schemes induced by non-breaking surface waves and internal tides are incorporated into
the MASNUM ocean circulation model (Han, 2014; Han and Yuan, 2014; Zhuang et al., 2018). The vertical mixing schemes
are introduced in Section 2. Section 3 describes the model and experiment design. Model results are given in Section 4. The
relevant discussion is given in Section 5 and the conclusions are summarized in Section 6.

## 2 Vertical Mixing Schemes

### 2.1 Non-breaking Surface-wave-generated Turbulent Mixing

Previous studies indicated that the NBSWs are able to enhance the turbulent mixing in the upper ocean (Babanin and
Haus, 2009; Dai et al., 2010; Huang and Qiao, 2010; Qiao et al., 2016). The ability to simulate the SST and the MLD can be
improved obviously via the incorporation of the related NBSW-induced turbulent mixing schemes into the OGCMs (Lin et al.,
2006; Xia et al., 2006; Song et al., 2007; Aijaz et al., 2017; Wang et al., 2019). According to Yuan et al. (2011) and Yuan et
al. (2013), Zhuang et al. (2020) expressed the vertical viscosity, $B_{us}$, and diffusivity, $B_{Ts}$, generated by the non-breaking surface
waves (NBSW) as follows,

$$
\begin{cases}
B_{us} = \left(\dfrac{7}{4}\right)^{1/2} \cdot h_{sw}^2 \cdot \left(\displaystyle\sum_{i=1}^{3}\sum_{j=1}^{3}\left|\dfrac{\partial u_i}{\partial x_j}\right|\right)_{sw}^2 \Bigg/ \left(\displaystyle\sum_{i=1}^{3}\sum_{j=1}^{3}\left|\dfrac{\partial u_i}{\partial x_j}\right|_{x_3=0}\right)_{sw} \\[3em]
B_{Ts} = \dfrac{1}{\sigma}\left(\dfrac{7}{4}\right)^{1/2} \cdot h_{sw}^2 \cdot \left(\displaystyle\sum_{i=1}^{3}\sum_{j=1}^{3}\left|\dfrac{\partial u_i}{\partial x_j}\right|\right)_{sw}^2 \Bigg/ \left(\displaystyle\sum_{i=1}^{3}\sum_{j=1}^{3}\left|\dfrac{\partial u_i}{\partial x_j}\right|_{x_3=0}\right)_{sw}
\end{cases}
, \tag{1}
$$

where $h_{sw}$ is the SWH, $\left(\displaystyle\sum_{i=1}^{3}\sum_{j=1}^{3}\left|\dfrac{\partial u_i}{\partial x_j}\right|\right)_{sw}$ is the averaged velocity shear module of the sea surface waves and can be calculated

as

$$
\left(\displaystyle\sum_{i=1}^{3}\sum_{j=1}^{3}\left|\dfrac{\partial u_i}{\partial x_j}\right|\right)_{sw}^2 = 2\iint\limits_{k} \Phi_{sw} K_{sw}^2 \omega_{sw}^2 \dfrac{\cosh\left[2K_{sw}(x_3-H)\right]}{\sinh^2\left[K_{sw}\cdot(-H)\right]}\, dk_1 dk_2 , \tag{2}
$$

where $\omega_{sw}$ is the surface wave frequency in a typical frequency range: $\omega_{sw} > N$, where $N$ denotes the Brunt-Väisälä frequency,

$K_{sw}$ is the wave number, $H$ is the water depth. $\Phi_{sw} = \eta_{sw} \cdot \eta_{sw}^*$ is the wave number spectrum of $h_{sw}$, $\eta_{sw}$ is the Fourier kernel

function of $h_{sw}$, i.e., $h_{sw} = \iint \eta_{sw} \exp\{i(k_1 x + k_2 y - \omega t)\} dk_1 dk_2$, here superscript "*" means the conjugate value, $k_1$ and $k_2$ are the

horizontal components of the wave number in x- and y- directions.

## 2.2. Mixing induced by surface wave transport flux residue

Apart from the NBSWs, the residue of the wave transport flux is also able to contribute to inducing the mixing to the
ocean circulation through the Reynolds average upon characteristic wavelength scale (Yang et al., 2009; Yang et al., 2019). Yang et al. (2009) proposed a mixing scheme for the wave transport flux residue (WTFR), which has been adopted in the OGCMs (Shi et al., 2016; Yu et al., 2020). The results show that the simulated SST and MLD are remarkably improved especially in summer and in the strong current regions. In the tropical cyclone conditions, the performance of the model to simulate ocean response could also be greatly improved, if the wave transport flux residue mixing scheme is introduced. The
coefficients of the wave transport flux residue mixing are expressed as follows,

$$
\begin{cases}
B_{SM1} = \iint\limits_{\bar{k}} \omega k_1 E(\bar{k}) \dfrac{\cosh[2K(x_3 - H)]}{\sinh^2[K \cdot (-H)]} dk_1 dk_2 \\
B_{SM2} = \iint\limits_{\bar{k}} \omega k_2 E(\bar{k}) \dfrac{\cosh[2K(x_3 - H)]}{\sinh^2[K \cdot (-H)]} dk_1 dk_2
\end{cases},
\tag{3}
$$

where $E(\bar{k})$ represents the wave number spectrum, which can be calculated from the wave spectrum model, other variables are the same as that in Eq. (2).

## 2.3. Internal-tide-generated Turbulent Mixing

In the stratified ocean interior, the internal tides are able to provide about half of the mechanical power required for the ocean interior turbulent mixing (Wunsch and Ferrari, 2004; Zhao, 2018; Vic et al., 2019; Whalen et al., 2020). However, the currently field observations are insufficient for constructing the whole internal-tide map in the IO. Satellite altimetry is able to provide sea surface height (SSH) measurements to observe the global internal tides (Ray and Mitchum, 1996). Zhao et al. (2016) presented a method to extract the $M_2$ internal tides by fitting plane waves to satellite altimeter data in individual
windows with size of 160 km × 160 km. In this technique, the least square fitting algorithm is adopted to determine the amplitude and phase of one plane wave. This procedure can be repeated three times to extract the three most dominant $M_2$ internal waves, of which superposition gives the final internal tidal solution. In this study, the turbulent mixing generated by $M_2$ semidiurnal internal tides will be derived from the SSH amplitude (Zhao et al., 2016). Other principal tidal constituents will be studied in future. For simplicity, the mode-1 $M_2$ internal tides, which mainly originate from regions with steep
topographic gradients, are considered, because the depth-integrated energy and SSH amplitudes of the mode-2 $M_2$ internal tides are much smaller than mode-1 ones (Zhao, 2018).

Yuan et al. (2013) presented the second-order turbulence closure model to estimate the turbulence kinetic energy and dissipation in terms of the velocity shear module of the non-breaking waves. The sub-surface displacements of internal tides, pressure anomalies and currents can be derived from the SSH amplitudes following vertical models (Zhao and Alford, 2009; Wunsch, 2013; Zhao, 2014). The detailed derivation process about the velocity shear module of the internal tide can be found in Appendix A. For the mode-1 $M_2$ internal tide, the vertical viscosity, $B_{ui}$, and diffusivity, $B_{Ti}$, generated by the velocity shear can be written as follows,

$$\begin{cases} B_{ui} = \left(\dfrac{7}{4}\right)^{1/2} \cdot h_{iw}^2 \cdot \left(\displaystyle\sum_{i=1}^{3}\sum_{j=1}^{3}\left|\dfrac{\partial u_i}{\partial x_j}\right|\right)_{iw}^2 \Bigg/ \left(\displaystyle\sum_{i=1}^{3}\sum_{j=1}^{3}\left|\dfrac{\partial u_i}{\partial x_j}\right|_{x_3=0}\right)_{iw} \\[4ex] B_{Ti} = \dfrac{1}{\sigma}\left(\dfrac{7}{4}\right)^{1/2} \cdot h_{iw}^2 \cdot \left(\displaystyle\sum_{i=1}^{3}\sum_{j=1}^{3}\left|\dfrac{\partial u_i}{\partial x_j}\right|\right)_{iw}^2 \Bigg/ \left(\displaystyle\sum_{i=1}^{3}\sum_{j=1}^{3}\left|\dfrac{\partial u_i}{\partial x_j}\right|_{x_3=0}\right)_{iw} \end{cases} , \tag{4}$$

where $h_{iw}$ is the SSH amplitude of the mode-1 $M_2$ internal tide. $\left(\displaystyle\sum_{i=1}^{3}\sum_{j=1}^{3}\left|\dfrac{\partial u_i}{\partial x_j}\right|\right)_{iw}$ is the averaged velocity shear module of the internal tides in a simple monochromatic form and can be calculated based on the unified linear theory under general ocean conditions (Yuan et al., 2011). The expression can be written as

$$\left(\sum_{i=1}^{3}\sum_{j=1}^{3}\left|\frac{\partial u_i}{\partial x_j}\right|\right)_{iw}^2 = 2h_{iw}K_{iw}^2\omega_{iw}^2 \frac{\left[\phi^2(\omega_{iw})-1\right]^2 \sin^2\left[\int_{-H}^{x_3}\phi(\omega_{iw})K_{iw}dx_3\right]+2\phi^2(\omega_{iw})}{\sin^2\left[\int_{-H}^{0}\phi(\omega_{iw})K_{iw}dx_3\right]} , \tag{5}$$

where

$$\phi(\omega_{iw}) = \sqrt{\frac{N^2 - \omega_{iw}^2}{\omega_{iw}^2 - f^2}} , \tag{6}$$

where $\omega_{iw}$ is the $M_2$ tidal frequency, $K_{iw}$ is the wave number. Under the influence of the Earth's rotation, the dispersion relation can be written as

$$\omega_{iw}^2 = K_{iw}^2 c_n^2 + f^2 , \tag{7}$$

where $f$ is the inertial frequency, $c_n$ is the eigenvalue speed, which is the phase speed in a non-rotating fluid, the expression can be written as

$$c_n^2 = \left[\frac{\int_{-H}^{0} N(x_3)dx_3}{n\pi}\right]^2 , \tag{8}$$

where $n$ is the mode number and set to be 1 here.

## 2.4. Incorporating the Vertical Mixing Schemes into OGCMs

The effects of the new vertical mixing schemes are introduced into the OGCMs. The modified equations can be written as

$$195 \quad \begin{cases} \dfrac{\partial U_i}{\partial t} + U_j \dfrac{\partial U_i}{\partial x_j} = F + \Pi_U \\[3mm] \dfrac{\partial C}{\partial t} + U_j \dfrac{\partial C}{\partial x_j} = G + \Pi_C \end{cases}, \quad (9)$$

where $x_j$ ($j = 1, 2, 3$) are the $x$, $y$, and $z$ axes of the Cartesian coordinates, $U_i$ ($i = 1, 2, 3$) and $C$ denote the mean velocity current components and one of the two tracers including the potential temperature and salinity, respectively. The second terms on the left hand side of Eq. (9) are the advection ones. $F$ represents the sum of the terms on the right hand side of the momentum equations including the Coriolis force, pressure gradient force, horizontal diffusion, molecular viscous force, and external

forcing terms. $G$ represents the sum of the terms on the right hand side of the tracer equations including horizontal diffusion, molecular diffusivity force, and heat/fresh-water flux terms. $\Pi_U$ and $\Pi_C$ denote the modified vertical diffusive terms and can be expressed as

$$\begin{cases} \Pi_U = \dfrac{\partial}{\partial x_3}\left(K_m \dfrac{\partial U_i}{\partial x_3}\right) + \dfrac{\partial}{\partial x_3}\left(B_{us} \dfrac{\partial U_i}{\partial x_3}\right) + \left(-B_{SM1}\dfrac{\partial U_i}{\partial x_1} - B_{SM2}\dfrac{\partial U_i}{\partial x_2}\right) + \dfrac{\partial}{\partial x_3}\left(B_{ui} \dfrac{\partial U_i}{\partial x_3}\right) \\[3mm] \Pi_C = \dfrac{\partial}{\partial x_3}\left(K_h \dfrac{\partial C}{\partial x_3}\right) + \dfrac{\partial}{\partial x_3}\left(B_{Ts} \dfrac{\partial C}{\partial x_3}\right) + \left(-B_{SM1}\dfrac{\partial C}{\partial x_1} - B_{SM2}\dfrac{\partial C}{\partial x_2}\right) + \dfrac{\partial}{\partial x_3}\left(B_{Ti} \dfrac{\partial C}{\partial x_3}\right) \end{cases}. \quad (10)$$

The new vertical diffusive terms $\Pi_U$ and $\Pi_C$ can be divided into four parts as shown in Eq. (10). The first term on the right side

denotes the original diffusive term, where $K_m$ and $K_h$ are vertical eddy viscosity and diffusivity calculated by the classic Mellor-Yamada 2.5 (M-Y 2.5) scheme (Mellor and Yamada, 1982). The M-Y 2.5 scheme is a level-2.5 turbulence model based on the modification of the material derivative and diffusive terms. The mixing coefficients $K_m$ and $K_h$ can be calculated as the turbulence characteristics $ql$ multiplied by a stability function associated with the Richardson number, where $q$ represents the turbulent fluctuation velocity and $q^2/2$ is the turbulence kinetic energy, $l$ is the mixing length scale. The turbulence kinetic

energy $q^2/2$ can be estimated from the local shear production, buoyancy and dissipation based on the $q^2$-$q^2l$ closure equations in the atmospheric boundary. Actually, the M-Y 2.5 scheme was proposed based on an assumption of the rigid surface and did not consider the effects of the surface and internal waves (Qiao et al., 2004; Huang and Qiao, 2010; Huang et al., 2011), which is regarded as one of the major reasons for the insufficient mixing in the upper ocean simulation. The remaining terms represent the new diffusive terms generated by surface waves and internal tides, which are described in Section 2.1 - 2.3.

## 3 Model Description and Numerical Experiment Design

### 3.1 Ocean circulation model

The three-dimensional MASNUM ocean circulation model (Han and Yuan, 2014; Zhuang et al., 2018) is used to evaluate the effects of the NBSW- and IT-generated turbulent mixing and the WTFR-induced mixing. The two-level single-step Eulerian forward-backward time-differencing scheme and the σ-Z-σ hybrid vertical coordinate are adopted in the MASNUM ocean model. The forward-backward scheme with a spatial smoothing method should be superior to the leapfrog scheme because of more stability and more computational efficiency (Han, 2014). Han (2014) and Han and Yuan (2014) have tested the modeling ability of the MASNUM model compared with the POM, the results showed that the MASNUM model could produce quite identical simulation results as the existing models with only half computer cost.

The model domain is in an area of 50° S - 30° N, 0° - 135° E (Fig. 1) with a horizontal resolution of 1°/6 by 1°/6. 5 surface σ layers, 31 intermediate Z layers and 3 bottom σ layers are used in the vertical direction in order to obtain the vertical grid spacing with a high resolution in the upper ocean. The topography of the model is down sampled from the global General Bathymetric Chart of the Oceans 2008 (GEBCO_08) with a resolution of 1′ by 1′. The minimum depth is set to 5 m. The maximum depth is set to 5,000 m avoiding the artificial influences at deep water depths. The topography has been smoothed using the dual-step five-point-involved spatial smoothing method (Han, 2014) to make the calculation more stable. The topographic gradients were not considered to be a key factor in this study because the climatological experiments in this study are inappropriate to directly simulate the ITs.

The initial temperature and salinity are interpolated based on the annually mean Levitus94 data (Levitus and Boyer, 1994; Levitus et al., 1994) with the horizontal resolution of 1° by 1° and 33 vertical layers. The initial velocities are set to 0. The gravity-wave radiation conditions (Chapman, 1985) were used as the lateral boundary conditions, which are very important for the basin-scale modeling, in this study. The simulated variables, including velocities, temperature, salinity and SSH, on the lateral boundary grids are calculated in an explicit numerical form. In the explicit form, the values of the related variables obtained from the daily global climatologic model results by the MASNUM model with the horizontal resolution of 1°/2 by 1°/2 are also used. The lateral boundary conditions are time-dependent with an updating period of 1 day. The surface forcing including the momentum, heat and wind stress fluxes are calculated from the monthly mean surface fields of the National Centers for Environmental Prediction / National Center for Atmospheric Research (NCEP/NCAR) reanalysis data set with the horizontal resolution of 1°/4 by 1°/4. We calculated the multi-year monthly mean surface forcing results based on the time series of the NCEP/NCAR data from 1948 to 2021, then the model is driven by the monthly mean surface forcing results, which repeats in every climatologic year. The time step size of the barotropic mode is set to 30 s, while that of the baroclinic mode is 900 s. The model is integrated from the quiescent state for 10 climatological years. The simulated results in the last 1 year are compared with the monthly World Ocean Atlas 2013 (WOA13) and the Ocean Surface Current Analyses Real-time (OSCAR) climatologic data, which can be regarded as the true solution of the climatological numerical experiments.

It is worth noting that the time interval of 10 years should be appropriate for ocean simulation from the quiescent state to a relatively stable circulation background. Because the average kinetic energy, which can be regarded as a model stability index, fluctuated obviously in the first two years, then became stable gradually in the third year, and was completely steady from the forth to the tenth year. The conclusion is similar to many previous studies (e.g. Xia et al., 2006; Qiao et al., 2010; Han, 2014; Yu et al., 2020).

## 3.2 Wave spectrum model

The MASNUM wave spectrum model (Yuan et al., 1991, 1992; Yang et al., 2019) is used to simulate the parameters of the surface wave in the IO. The energy balanced equations are solved in the model based on the wave number spectrum space. The characteristics inlaid scheme is adopted for the wave energy propagation to improve the original wave model (Yuan et al., 1992). The wave model has been validated by the observations (Yu et al., 1997) and widely accepted in ocean engineering and numerical simulation (e.g. Qiao et al., 1999; Xia et al., 2006; Qiao et al., 2010; Shi et al., 2016; Yang et al., 2019; Yu et al., 2020; Sun et al., 2021). The results showed that the simulated SWH and mean wave period were consistent with the satellite observations.

The model domain, the resolution, the topography and the surface wind stress flux data are consistent with that in the MASNUM ocean circulation model. The boundary conditions are from the JONSWAP spectrum (Hasselmann et al., 1973). The wave model is integrated from the quiescent state for 10 climatological years with the same period as the ocean circulation model. Actually, the configuration of the wave model is simpler than the OGCM, and the model design in this study is almost the same as that in Xia et al. (2006) and Qiao et al. (2010). Therefore, we believe that the experiment using the MASNUM wave model is able to characterize the spatial pattern and variation of the surface waves in the IO.

The wave spectrum $E(\vec{k})$ is calculated from the MASNUM wave model, and then $\omega_{sw}$, $K_{sw}$, and $h_{sw}$ can be estimated. Thus, the new mixing coefficients including $B_{us}$, $B_{Ts}$, $B_{SM1}$, $B_{SM2}$, $B_{ui}$, and $B_{Ti}$ are calculated directly from Eqs. (1) - (4).

## 3.3 Experimental design

To assess the effects of the NBSW, WTFR and IT on the vertical mixing and the simulated thermal structure in the upper ocean, five experiments (Table 1) are denoted as Exp 1 - 5 and designed as follows:

Exp 1 (benchmark experiment): The model is integrated with the classic M-Y 2.5 turbulence closure model (Mellor and Yamada, 1982), which is broadly used in the OGCMs.

Exp 2: Same as Exp 1, except with the classic M-Y 2.5 scheme and the NBSW-generated turbulent mixing scheme. This experiment is designed to evaluate the effect of the NBSW.

Exp 3: Same as Exp 1, except with the classic M-Y 2.5 scheme and the NBSW- and IT-generated turbulent mixing schemes. This experiment is designed to evaluate the effects of the NBSW and the IT. The experiment with the M-Y 2.5 scheme and the IT-generated turbulent mixing scheme (Exp 3.5) is omitted in this study, because the deviation of the

temperature between Exp 1 and Exp 3.5 is too small. The possible reason is that the IT is often considered to enhance the vertical mixing in the ocean interior from the thermocline to the abyssal regions (Munk and Wunsch, 1998; Wunsch and Ferrari, 2004; Kunze et al., 2006), therefore, it should be insufficient for the incorporation of only the IT into the M-Y 2.5 scheme to draw the warmer water from the surface into the interior. This implies that only the IT is unable to improve the upper-ocean simulation.

Exp 4: Same as Exp 1, except with the classic M-Y 2.5 scheme and the WTFR-induced mixing scheme. Comparisons between Exp 2 and Exp 4 are implemented to evaluate the two mechanisms that the surface waves affect the upper-ocean vertical mixing.

Exp 5: Same as Exp 1, except with the classic M-Y 2.5 scheme and the NBSW- and IT-generated turbulent mixing and the WTFR-induced mixing schemes. This experiment is designed to evaluate the effects of the NBSW, the IT and the WTFR.

**Table 1: Numerical experiment design**

|  | NBSW | WTFR | IT |
|---|---|---|---|
| Exp 1 | No | No | No |
| Exp 2 | Yes | No | No |
| Exp 3 | Yes | No | Yes |
| Exp 4 | No | Yes | No |
| Exp 5 | Yes | Yes | Yes |

It is worth noting that the climatological experiments, which should be regarded as the multi-year mean simulation, are designed in this study, so it is inappropriate for the simulated results to be compared with the Argo data, because there should be considerable difference between the climatologic data and the real-time in-situ observations. The WOA13 data, which is the multi-year (1955 - 2012) mean results, and the multi-year (1993 - 2021) mean OSCAR data will be a good choice to evaluate the ocean climatological modeling.

## 4. Results

In this section, the comparable results for the climatological temperature construction in the upper ocean are used to assess the effects of the NBSW, the IT, and the WTFR on the vertical mixing.

### 4.1 Comparison of vertical diffusive terms

As a typical example, the vertical distribution of the monthly mean vertical temperature diffusive terms in logarithmic scale along zonal transect of 10.5° S in January and July are shown in Figs. 2 and 3. As expressed in Eq. (10), the calculated

vertical diffusive term based on the M-Y 2.5 scheme, which is written as $\frac{\partial}{\partial x_3}\left(K_h \frac{\partial T}{\partial x_3}\right)$ (KHT for short) and calculated from

Exp 1, the NBSW-generated turbulent mixing scheme, which is written as $\frac{\partial}{\partial x_3}\left(B_{Ts} \frac{\partial T}{\partial x_3}\right)$ (BTST) and calculated from Exp 2,

the IT-generated turbulent mixing scheme, which is written as $\frac{\partial}{\partial x_3}\left(B_{Ti} \frac{\partial T}{\partial x_3}\right)$ (BTIT) and calculated from Exp 3 and the WTFR-

induced mixing scheme, which is written as $-B_{SM1} \frac{\partial T}{\partial x_1} - B_{SM2} \frac{\partial T}{\partial x_2}$ (BSMT) and calculated from Exp 4, are presented,

respectively. The BTIT can be calculated independently as a part of the diffusive terms in Exp 3. Figures 2 and 3 show the
comparisons among these diffusive terms along 10.5° S, which is a typical transect to show the difference, in January and July.
There are two reasons for the choice of the 10.5° S Transect. Firstly, the Madagascar-Mascarene regions (0° ~ 25° S in the
West Indian Ocean) are considered to be a hot spot for generation of semidiurnal ITs (Zhao et al., 2016). Both of the NBSWs
and ITs should grow fully and become large enough for the comparison of the diffusive terms. Secondly, the 10.5° S Transect
is typical to show the spatial pattern among the diffusion terms obviously because of the stronger surface waves and ITs
(Kumar et al., 2013; Zhao et al., 2016; Ansong et al., 2017).

One can see that all of the terms decay with the depth below the sea surface. In January, BTST is $> 10^{-2}$ °C s$^{-1}$ in the
upper-30 m layer of most regions, of which the values are too high to show in Figs. 2b and 3b, and obviously greater than other
terms, implying that the NBSW-generated turbulent mixing is dominant in the layers with depths less than 30 m. Similar to
BTST, BSMT (Fig. 2d) is also induced by NBSW and directly generated by the surface wave orbital velocity, but the values
are about 4 to 6 orders smaller than those of BTST. However, in July, BSMT may affect greater depths than BTST and KHT
especially in some regions with large topographic relief. In the ocean interior with depths from 40 m to 130 m, BTIT (Fig. 2c)
is about $10^{-6}$ °C s$^{-1}$ and significantly higher than other three terms in some regions such as the East Atlantic (5° E - 10° E), the
West Indian Ocean (50° E - 70° E) and the West Pacific (122° E - 125° E) because of the effects of the IT. It is worth noting
that the vertical distribution of the eddy diffusivity $K_h$, $B_{Ts}$ and $B_{Ti}$ is very similar to the diffusive terms. Especially in January,
$B_{Ts}$ is the largest in the upper-30 m layers and $B_{Ti}$ is larger generally in the ocean interior with the depth deeper than about 40
m. $K_h$ and $B_{Ts}$ decay with the depth below the sea surface, the delay rate of $B_{Ts}$ is slower obviously than $K_h$, so $B_{Ts}$ is larger than
$K_h$ in the ocean interior. The high-value layers ($>10^{-5}$ m$^2$/s) of the $K_h$ are as thin as about 20 m in January, and up to about 80
m partially in July, while the high-value layers of the $B_{Ts}$ are generally about 70-100 m both in January and July. When the
depth is larger than 40 m, the value of $B_{Ti}$ appear to be about $10^{-5}$-$10^{-3}$ m$^2$/s.

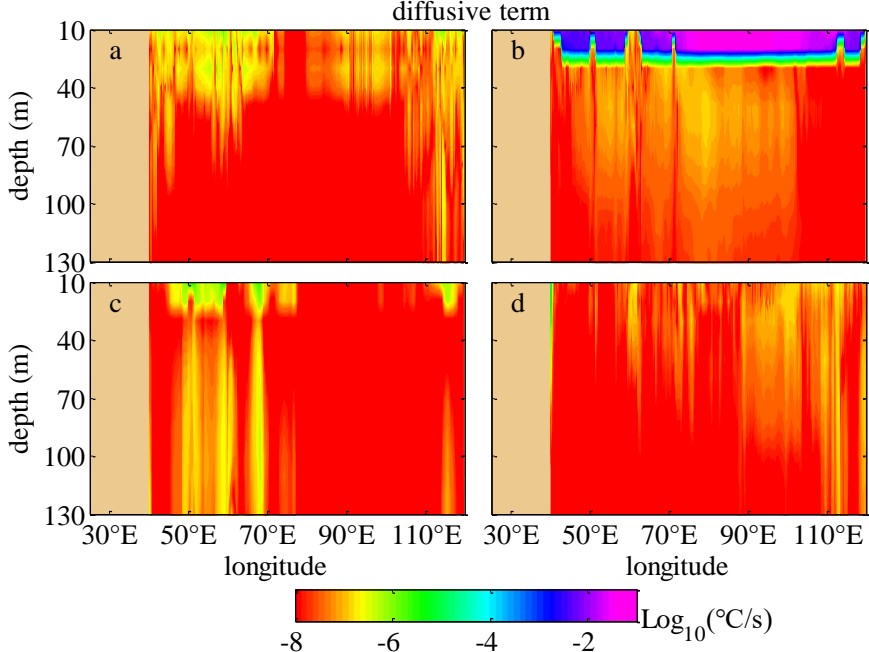


**Figure 2: Vertical profiles of the monthly mean vertical temperature diffusive terms in logarithmic scale along 10.5° S in January, including the diffusive term based on the M-Y 2.5 scheme (a), the NBSW-generated turbulent mixing scheme (b), the IT-generated turbulent mixing scheme (c) and the WTFR-induced mixing scheme (d). Deep yellow areas correspond to the lands**

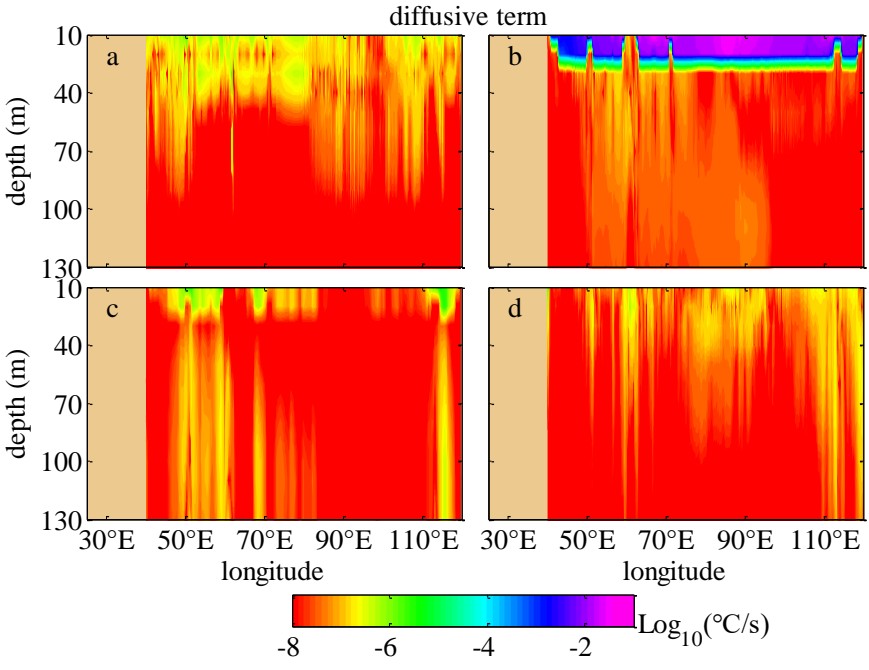

**Figure 3: The same as Fig. 2, but in July**

## 4.2 Effects on simulation of the vertical temperature structure

The climatologic experiments are designed in this study because of the NCEP monthly climatologic sea surface flux forcing fields and the daily global climatologic lateral boundary conditions in the simulation, so the WOA13 monthly climatology data can be used in comparisons as a reference.

Figs. 4 - 7 show the comparisons of the upper-ocean temperature vertical structure between the WOA13 data and the model results of the five experiments along transects of 30.5° S and 7.5° N, corresponding to SIO and north of the Equatorial Indian Ocean (EIO), in January and July. In the southern hemisphere, the 30.5° S Transect is typical to show the effects of the three schemes on the temperature modeling. The temperature structure along 10.5° S, which is used to show the comparison of the diffusive terms in Section 4.1, is omitted here, because there is non-ignorable difference between the WOA13 data and

the simulation results especially in the East Indian Ocean, and the effects of the NBSW in the tropical area are relatively non-obvious, which is regarded as a long-standing issue (Qiao et al., 2010; Zhuang et al., 2020). In the northern hemisphere, the temperature structure along the 7.5° N Transect, which is located in the south of the Arabian Sea and Bay of Bengal and regarded to be representative for modeling evaluation, is discussed. One can see that the difference by subtracting the monthly mean results of Exp 1 from the monthly WOA13 data is the largest among the five experiments along the two transects in

January and July. In the ocean interior, the temperature of Exp 1 is extremely lower than the WOA13 data, which implies that less surface water is transferred into the layers with depths from 30 m to 100 m in Exp 1 because of the insufficient vertical mixing process simulated by the classic M-Y 2.5 scheme. Compared with Exp 1, the difference for Exp 2 decreases remarkably, because the NBSW strengthens the vertical mixing and improve the upper-ocean simulation, which has been proved many times by previous studies (Lin et al., 2006; Huang et al., 2011; Qiao et al., 2016; Zhuang et al., 2020).

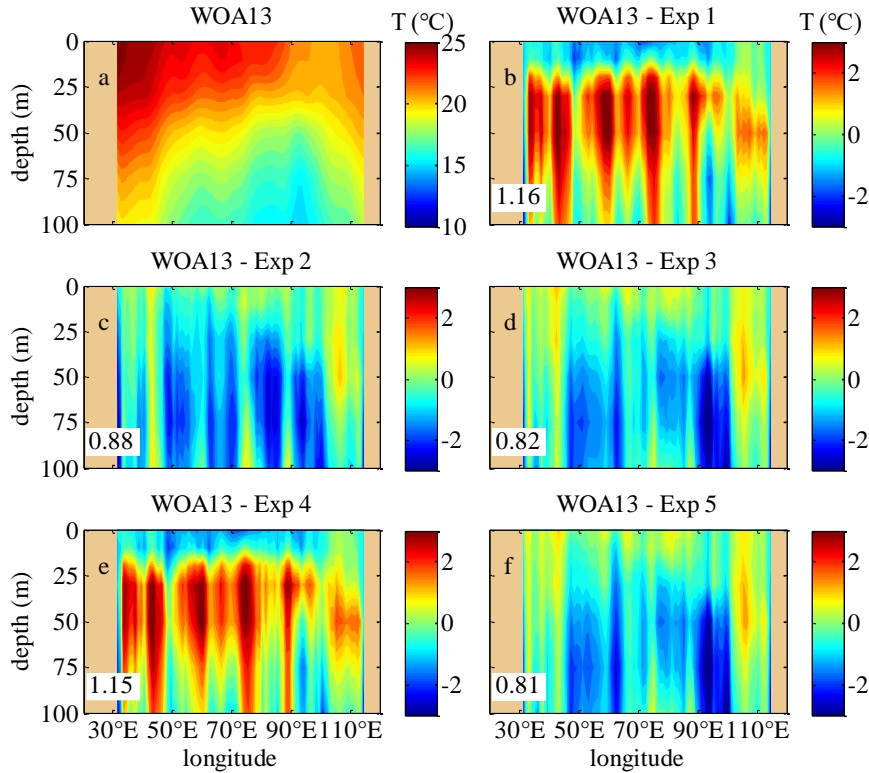


**Figure 4: The vertical temperature profiles along 30.5° S in January. (a) The temperature structure from the monthly WOA13 data (units: °C). (b) - (f) The difference of the temperature calculated by subtracting the monthly mean results simulated in Exp 1 - Exp 5 from the monthly WOA13 data, respectively. The RMSE of the temperature in the upper-100 m regions between the WOA13 data and the model results are given. Deep yellow areas correspond to the lands**

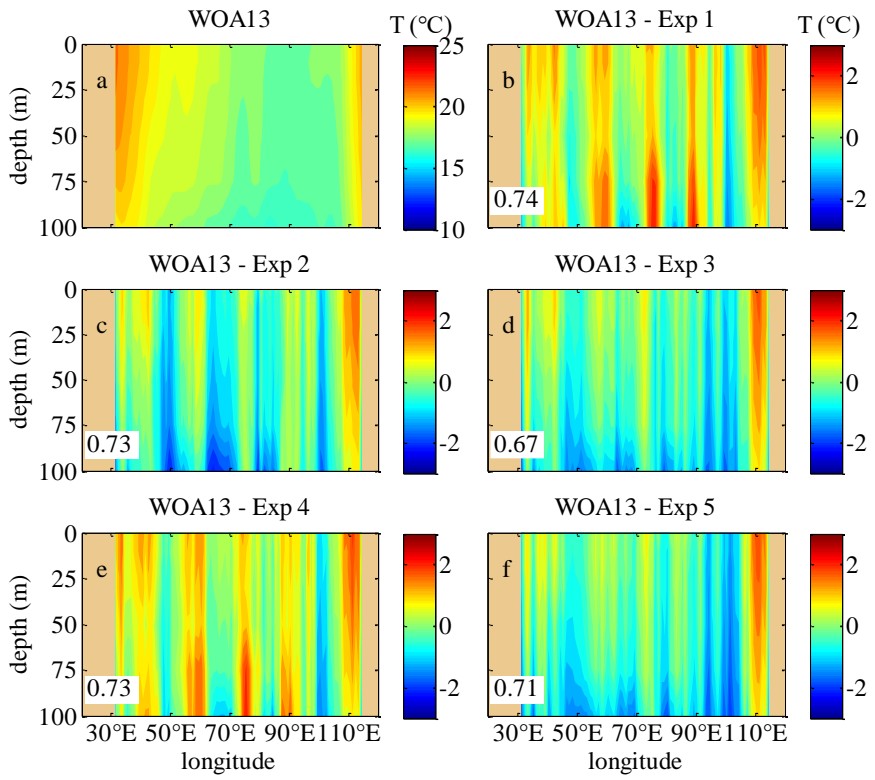


**Figure 5: The same as Fig. 4, but in July**

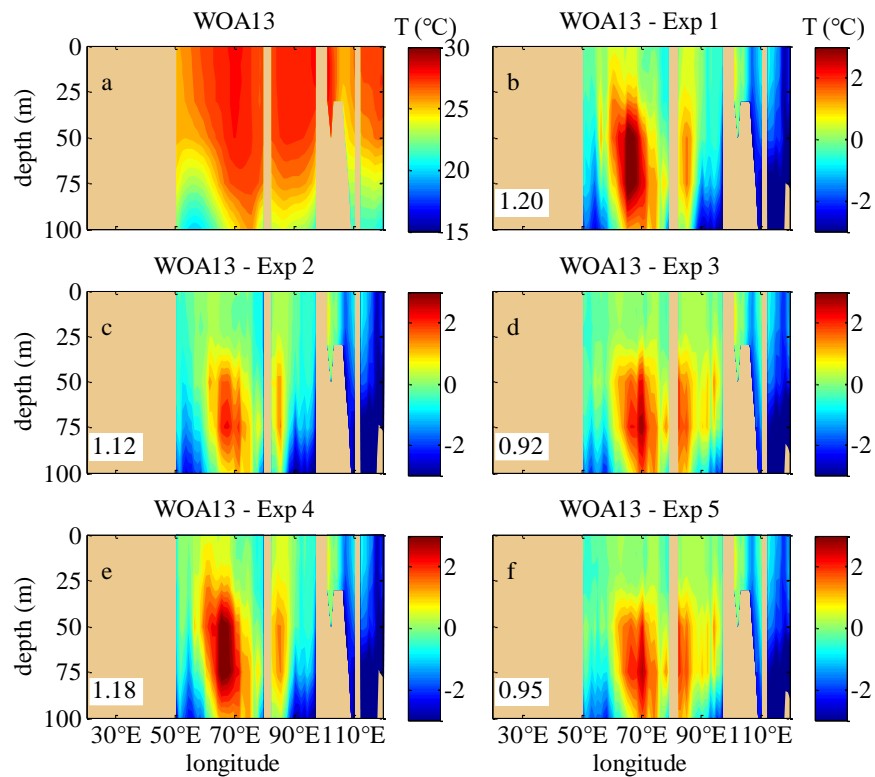

**Figure 6: The same as Fig. 4, but along 7.5° N**

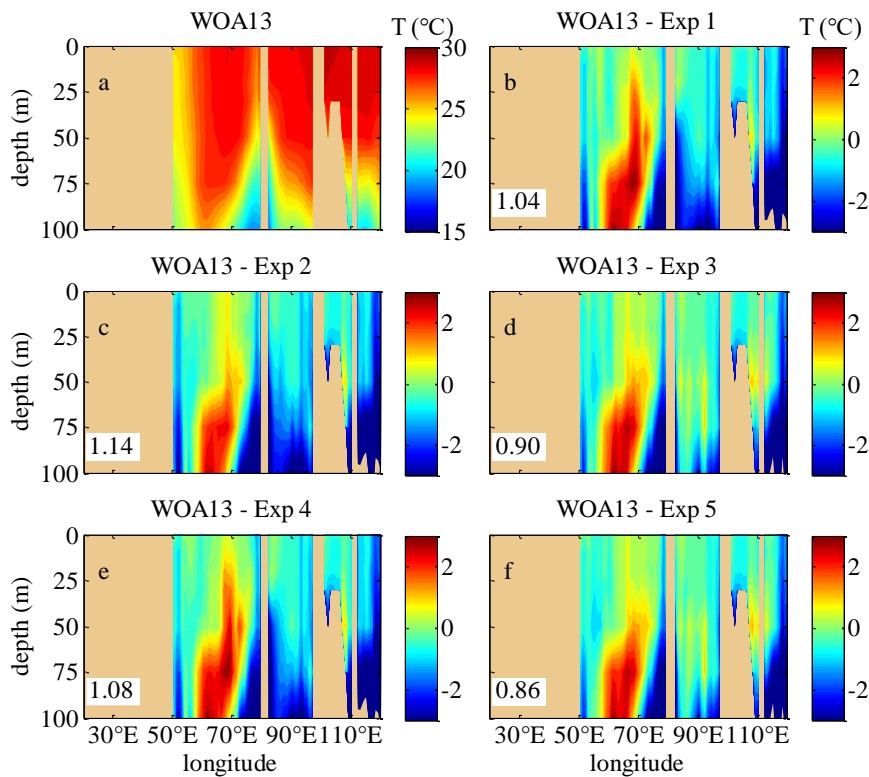

**Figure 7: The same as Fig. 4, but along 7.5° N and in July**

The difference for Exp 3 is much smaller than that of Exp 1 and Exp 2 because of the incorporation of the IT-generated turbulent mixing, especially in the layers with depths between 20 m and 50 m. This implies that the IT strengthens the vertical mixing of the ocean interior and improve the simulation further. It is worth noting that the experiment with the classic M-Y 2.5 scheme and the IT-generated turbulent mixing scheme is omitted, the reason is that the results have not been improved, if only the IT-generated turbulent mixing is incorporated because the simulated surface mixing is insufficient, and even been deteriorated in some regions because the colder water will be drawn from the lower layers with depths deeper than 100 m into the upper ocean.

However, the simulation is slightly improved in Exp 4 compared with Exp 1 because the BSMT, which is induced by the WTFR, is remarkably smaller than the BTW, so the WTFR-induced mixing is too insufficient to significantly improve simulating the upper-ocean temperature structure. Similarly, there is less difference between Exp 3 and Exp 5, implying that the effects of the WTFR on enhancing vertical mixing are much weaker than the NBSW in the surface layers and the IT in the ocean interior.

In Exp 1, the simulated temperature along 30.5° S is cooler than the WOA13 data, while the temperature bias becomes reversed with similar magnitude in Exp 2 and Exp 3. The reason should be that the multi-year monthly mean surface forcing fields, on one hand, were smaller than the actual values, which leads to insufficient heat transfer from the atmosphere to the

ocean. After 10 climatologic years modeling, the temperature in the ocean interior became cooler obviously than the WOA13 data. On the other hand, the NBSWs and ITs enhanced the vertical mixing as well as the heat transfer, so more heat entered into the ocean interior and the SST became cooler. Additionally, the Haney equation (Haney, 1971), which improves the large-scale thermal coupling of ocean and atmosphere, is used to modify the surface heat flux. However, a disadvantage of the Haney modifying method is the destruction of the heat balance, so the solar radiation will increase continuously in the ocean surface. Therefore, the accumulation of the heat during the 10-year modeling will make the temperature bias in Exp 2 and Exp 3 reversed with similar magnitude in Exp 1. Furthermore, the temperature from the annual mean Levitus94 data, which are used as the initial fields, is cooler about 3℃ than that from the WOA13 data partially in the Antarctic Circumpolar Current (ACC) region from surface to 200 m depth, and warmer about 0.5℃ in the NIO and tropics when the depth is deeper than 200 m. This should make the simulated temperature cooler than the WOA13 data in the SIO and become warmer in the NIO and the tropics. The improvement of the NBSWs and ITs on the temperature simulation is obvious because of the smaller errors in Exp 2 and Exp 3, although the difference of the temperature bias between Exp 1 and Exp 2/3 is substantial.

Figure 8 shows the monthly variability of the root mean square errors (RMSEs) of the temperature in the upper-100 m layers between the WOA13 data and the model results. Actually, the RMSEs are calculated based on the simulated temperature only in the whole IO (the regions outside of the IO have been removed) as the following expression

$$\text{RMSE} = \sqrt{\frac{\sum\limits_{i=1}^{im}\sum\limits_{j=1}^{jm}\sum\limits_{k=1}^{ks}\left(t_{m\ i,j,k} - t_{w\ i,j,k}\right)^2}{im \times jm \times ks}} \tag{11}$$

where $t_m$ and $t_w$ represent the model results and the WOA13 data of the monthly mean temperature, im, jm and ks means the number of grids in the whole IO in horizontal and vertical directions. The RMSE can be regarded as a spatially average deviation of the three-dimensional temperature fields. Therefore, the RMSE should be statistically robust because the calculated result is unique if the spatial range of the temperature field is determined.

The study area is divided into three zones (Zones 1 - 3 marked in Fig. 1). The zone partition of the IO in this study is designed based on the previous studies and the dynamic patterns of the IO. On one hand, previous studies (Talley et al., 2011; Kumar et al., 2013; Kumar et al., 2018) showed different zone partitioning criteria which often included the NIO, the SIO and the tropical regions. On the other hand, the principal upper ocean flow regimes of the IO are the subtropical gyre of the SIO and the monsoonally forced circulation of the tropics and NIO. The whole effects of the Indonesian throughflow (ITF) should be also considered. Taking the above factors into account, the whole IO was divided into three parts, Zone 1 represents the NIO including the Arabian Sea and the Bay of Bengal, Zone 2 represents the tropics and the subtropical regions in the SIO with the whole effects of the subtropical gyre and the ITF, and Zone 3 represents the region on the south of Zone 2 in the SIO. In Zone 2, there is a complete cyclonic circulation system between the equator and 20° S, consisting of the westward South Equatorial Current on the south side, the eastward South Equatorial Countercurrent on the north side, the northward East African Coastal Current and the ITF. The effects of the $M_2$ internal tides, which are generated in the northern regions around the Madagascar Island, are produced throughout the whole west region in Zone 2.

In Zone 1, the RMSEs for Exp 2 are smaller than that for Exp 1 in all of the months, indicating the improvement of the NBSW on the upper-ocean simulation in the NIO. Compared with Exp 2, the RMSEs for Exp 3 are smaller in most of the months except November, December and January. This implies that the IT enhances the vertical mixing and improves the simulation further. The possible reason for little effects of the IT from November to January is that, on one hand, the mixed layer depths in the NIO are relatively shallower in boreal winter, so that the averaged velocity shear module of the internal tides is smaller and the IT-induced mixing is weaker, on the other hand, the strength of the surface waves is more intensive, so the NBSW-induced mixing is relatively sufficient. The largest declines occurred in May, when the RMSE decreased 14.0% from 1.72 ℃ (Exp 1) to 1.49 ℃ (Exp 2), and 19.1% from 1.72 ℃ (Exp 1) to 1.40 ℃ (Exp 3).

In Zone 2, the NBSW is ineffective, because the RMSEs for Exp 2 are almost equal to, even larger than, those for Exp 1. This is a long-standing issue about the trivial effects of the NBSW in the tropical area (Qiao et al., 2010; Zhuang et al., 2020), implying that only the NBSW should be not enough to improve the tropical simulation. To solve this issue, the coupled atmosphere-wave-ocean-ice modeling is one of the solution (Song et al., 2012; Wang et al., 2019). Another way is incorporation of the additional mechanism into the OGCMs, such as the IT-generated turbulent mixing added in Exp 3 and Exp 5. The RMSEs for Exp 3 are obviously smaller than that for Exp 1 and Exp 2 in the whole climatologic year except March, implying that the combination of the NBSW and the IT is able to improve the simulation of the temperature structure in the tropical area. Additionally, the RMSEs in Zone 2 are smaller than that in Zones 1 and 3 on the whole, the RMSEs in Zone 2 for Exp 3 and Exp 5 are even less than 0.9 ℃ in half of the climatologic year, indicating much accurate simulation in the tropical area.

In Zone 3, the results are similar to that in Zone 1, the RMSEs for Exp 2 are smaller than that for Exp 1 in most months, and the RMSEs for Exp 3 are the smallest ones among the first three experiments. The largest declines occurred in January, when the RMSE decreases 20.8% from 1.50 ℃ (Exp 1) to 1.18 ℃ (Exp 2), and 25.7% from 1.50 ℃ (Exp 1) to 1.11 ℃ (Exp 3). The situation also indicates the significant improvements of the combination of the NBSW and the IT in simulating the upper-ocean temperature structure.

Furthermore, in Zones 1 - 3, the effects of WTFR are much weaker and similar to that in Figs. 4 - 7, because the RMSEs for Exp 4/5 are almost equal to, and even larger than, that for Exp 1/3. The possible reason is that the values of the WTFR-induced diffusion terms are about 4 to 6 orders smaller than NBSW, which are too low to enhance the vertical mixing especially in the surface layers.

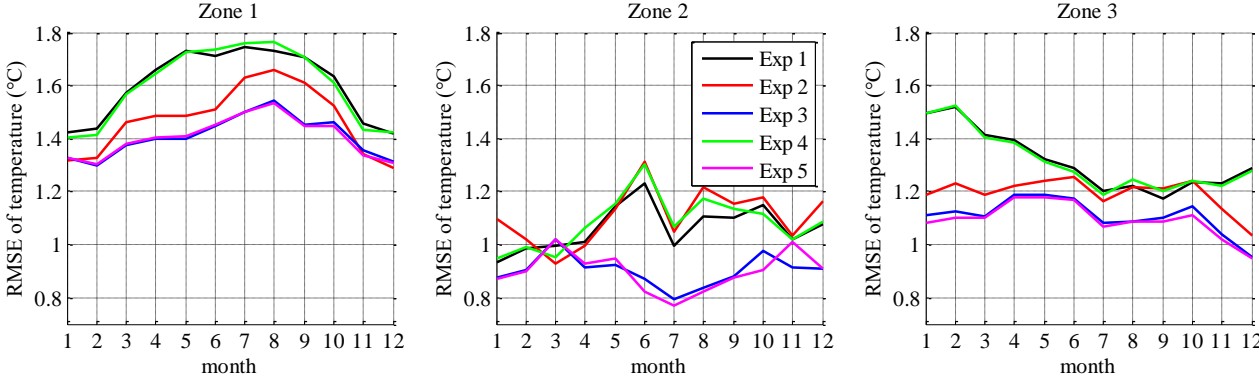


**Figure 8: Variation of the RMSE of the temperature between the simulated monthly mean results in the five experiments and the monthly WOA13 data in Zones 1 - 3 (shown in Fig. 1)**

The thermal structure in the regions with depths from 100 m to 300 m are also compared with the WOA13 data. The simulated temperature is generally cooler than the WOA13 data along 30.5° S with the depths between 100 and 300 m, while

warmer dramatically along 7.5° N. The distribution pattern of the simulated temperature in the ocean interior (from 100 m to 300 m or deeper) seems to appear as cooler in SIO and warmer in NIO and tropics. One of the reasons is that, the Haney equation (Haney, 1971) is used to modify the climatologic surface heat flux and brings excessive heat into the ocean interior in the simulation. Some more accurate surface forcing data with higher resolution will be used in future simulation.

Furthermore, the intermediate and deep water masses in the IO are often effected by the Southern Ocean including

Antarctic Intermediate Water, Circumpolar Deep Water, North Atlantic Deep Water, etc. These cooler water masses are carried by the meridional overturning circulation into the IO throughout south of the South Equatorial Current in the subtropical Indian Ocean (Talley et al., 2011), but the situation did not appear in the simulated current fields. Therefore, another important reason should be that it is hard to simulate accurately the meridional overturning circulation in the present experiments, especially the meridional transport of the heat. This makes the simulated temperature cooler/warmer than the WOA13 data along 30.5° S/7.5°

N when the depth is deeper than about 100 m.

In addition, it is worth noting that the initialization design is also important for the ocean modeling. The comparison between of the annual mean temperature between the Levitus94 and WOA13 data shows that the temperature from the Levitus94 data is cooler obviously than that from the WOA13 data in the ACC regions (45° - 75° E, 35° - 50° S), while warmer generally in the whole IO with the depth from 200 to 500 m. The WOA13 data contains more meso-scale information than the

Levitus94 data. Therefore, the inaccurate initial field should be also one of the reasons why the simulated temperature in the ocean interior is different from the WOA13 data. A series of the high-resolution real-time numerical experiments for the circulation in the IO will be carried out to examine the influence of different initial fields, parameterization schemes, surface fluxes, and open boundary conditions in future. It is worth noting that the detailed analysis for the deep ocean is omitted here because the vertical mixing in the upper ocean (0 ~ 100 m) is the main focus of this study.

Lozovatsky et al. (2022) demonstrated that the internal wave instabilities appear to be a dominant mechanism for generating the energetic mixing based on the analysis of the in-situ observations of the turbulent kinetic energy dissipation rate and buoyancy frequency profiles. Actually, designing a universal and flexible IT-induced mixing scheme for ocean modeling based on the in-situ observations still needs a lot of works. The three schemes introduced in this study are just preliminary research on the contribution of the upper-ocean vertical mixing.

The thermocline structure, which is normally defined as the depth of 20 °C isothermal (Z20), can affect SST variability via vertical water exchange and thereby modulate the air-sea interaction events (Talley et al., 2011). Vertical displacements of the thermocline depth at the equatorial region are regarded as one of the distinctive feature in the IO. Because of the weak westerly at the equator, the IO shows deeper and reversed slope of Z20 as compared to its counterpart in the Pacific Ocean. Analysis of the mean state thermocline structure is very important because the thermocline variability is related to the Indian

Ocean Dipole, ITF and El Niño-South Oscillation at the interannual time scale according to previous studies (e.g. Chambers et al., 1999; Gordon et al., 2003; Liu et al., 2017). Figures 9 show the variation of the Z20 depths along the equator in January and July. As another indicator of the thermal structure in the upper-100 m layers, the depths of 26 °C isothermal (Z26) are also plotted in Figs. 9a and 9b. The RMSEs of the Z20 and Z26 depths are calculated and plotted in Figs. 9c and 9d.

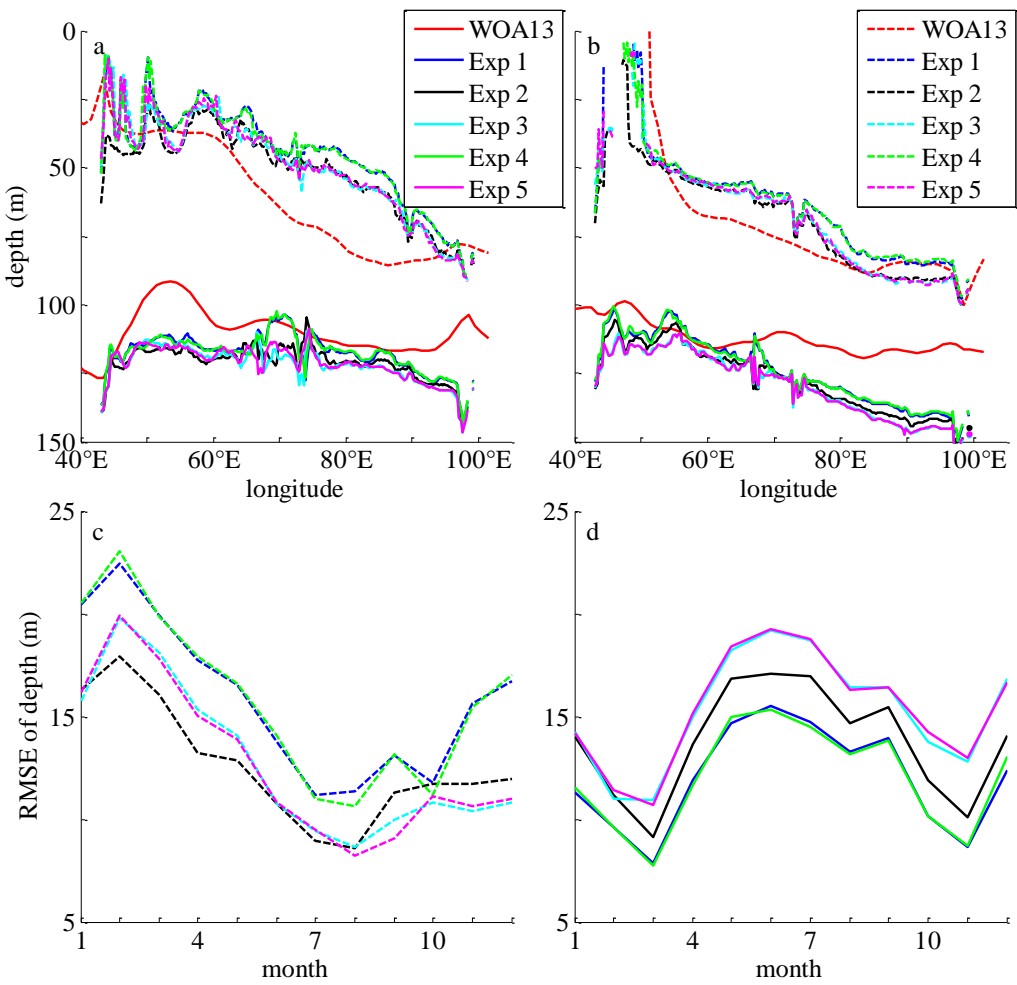

**Figure 9: The comparison of the Z20 and Z26 depths along the equator from the WOA13 data and the model results. a and b: the Z20 (solid curves) and Z26 (dashed curves) depths from the WOA13 data and the model results (Exp 1 ~ Exp 5) in January and July, respectively. The RMSEs of the Z26 (c) and Z20 (d) depths along the equator in the whole climatologic year are also plotted.**

From Figs. 9 one can see that the Z20 and Z26 depths are both shallow in the west and deep in the east, the simulation of the thermal structure in the five experiments depict this pattern successfully, but there is still obvious difference between the

WOA13 data and the results especially in the east regions in January for Z26 depths and in July for Z20 depths. One of the main reason should be that the ITF may be simulated inaccurately because of the inaccurate topography in the Indonesian regions and the open boundary conditions. The accurate simulation of the ITF should be a difficult issue because of the complicated topography and ocean-atmosphere interaction in the Indonesian Archipelago. Many OGCMs are incapable of reproducing the patterns of the ITF (Nagai et al., 2017; Santoso et al., 2022). Another reason is that this area is full of eddies

produced by the horizontal velocity shear but our ocean model still lack an accurate and reasonable parameterization of the eddy-induced mixing, which needs more future works. For Z26 depths, the RMSEs for Exp 1 and Exp 4 are the largest in almost all of the months, this implies that the WTFR-induced mixing has a little effects on the modeling, which is consistent

with the comparison results above (Figs. 4 ~ 8). The NBSW- and IT-generated turbulence mixing can improve the simulated thermal structure as two of the key factors because of the smallest RMSEs (from February to July for Exp 2, and from August to January for Exp 3 and Exp 5). For Z20 depths, the NBSW and the IT have a negative effects on the modeling because the RMSEs for Exp 2 ~ Exp 5 are larger obviously than those for Exp 1. The reason is that the Z20 isothermal simulated in Exp 1 is generally deeper than the WOA13 data because the simulated temperature in the regions with depths from 130 m to 200 m is warmer than the WOA13 data, the enhanced vertical mixing induced by the NBSW and the IT will make the Z20 isothermal deeper further and deviate more from the WOA13 data (solid curves in Figs. 9a and 9b). Therefore, more optimization and improvement of the experimental design will be implemented in future work to make the simulated results more accurate.

In order to evaluate the modeling results further, the existing Argo-derived gridded products, which are named Barnes objective analysis (BOA)-Argo datasets (Li et al., 2017), are also chosen. The climatologic monthly mean BOA-Argo data (multi-year mean from 2004 to 2014) are used and can be downloaded directly from ftp://data.argo.org.cn/pub/ARGO/BOA_Argo/. The BOA-Argo data with 49 vertical levels from the surface to 1950 m depth is produced based on refined Barnes successive corrections by adopting flexible response functions. A series of error analyses are adopted to minimize errors induced by non-uniform spatial distribution of Argo observations. These response functions allow BOA-Argo to capture a greater portion of mesoscale and large-scale signals while compressing small-sale and high-frequency noise the performance of the BOA-Argo dataset demonstrates both an accuracy and retainment of mesoscale features. Generally, BOA-Argo seems compare well with other global gridded data sets (Li et al., 2017).

Figures 10 and 11 show the comparison of the temperature structure between the monthly BOA-Argo data and the model results in January. The vertical distribution are similar to those from the WOA13 data (see Sub-figures a in Figures 4, 6 10 and 11). The difference between the BOA-Argo data and the model results along 30.5° S is also similar to the WOA13 data. Compared with Exp 1, the difference for Exp 2 often decreases remarkably, the difference for Exp 3 is much smaller than that of Exp 1 and Exp 2 because of the incorporation of the IT-generated turbulent mixing, especially in the layers with depths between 20 m and 50 m. In addition, the improvement of the NBSW and IT along 7.5° N is not obvious, this conclusion is also similar to that for the WOA13 data. This implies that the three mixing schemes introduced in this study may not be appropriate in the marginal sea simulation that if full of small- and meso-scale processes. In order to solve the issues about the accuracy, we attempt to design the high-resolution real-time numerical modeling experiments in the NIO (or the Arabian Sea and the Bay of Bengal only), as well as the finer simulation of the surface waves and more accurate estimation of the ITs.

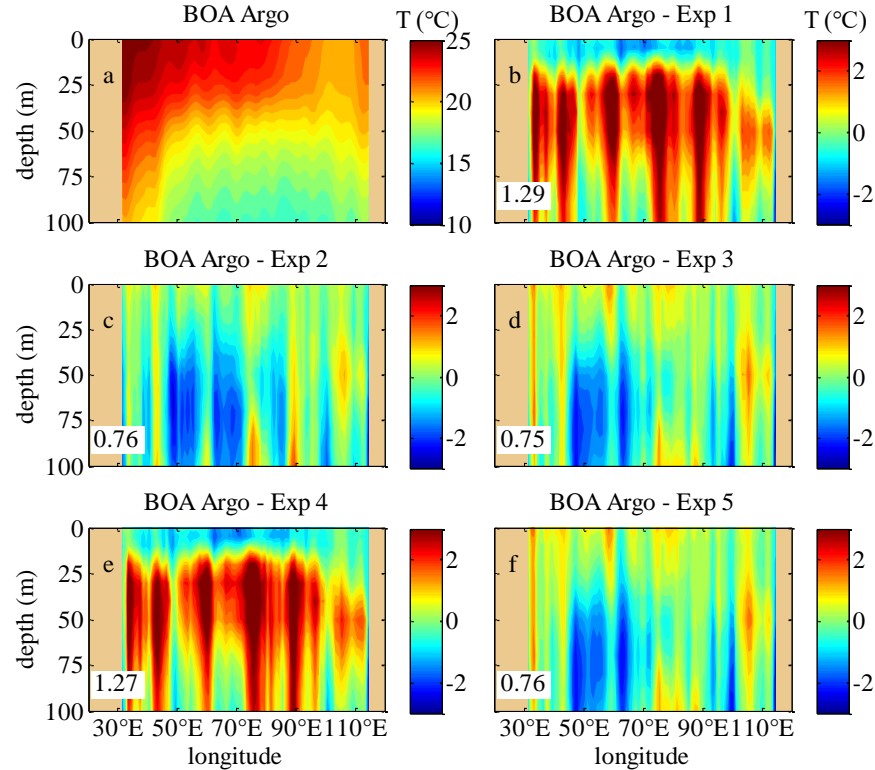


**Figure 10: The vertical temperature profiles along 30.5° S in January. (a) The temperature structure from the monthly BOA-Argo data (units: °C). (b) - (f) The difference of the temperature calculated by subtracting the monthly mean results simulated in Exp 1 - Exp 5 from the monthly BOA-Argo data, respectively. The RMSE of the temperature in the upper-100 m regions between the BOA-Argo data and the model results are given. Deep yellow areas correspond to the lands**

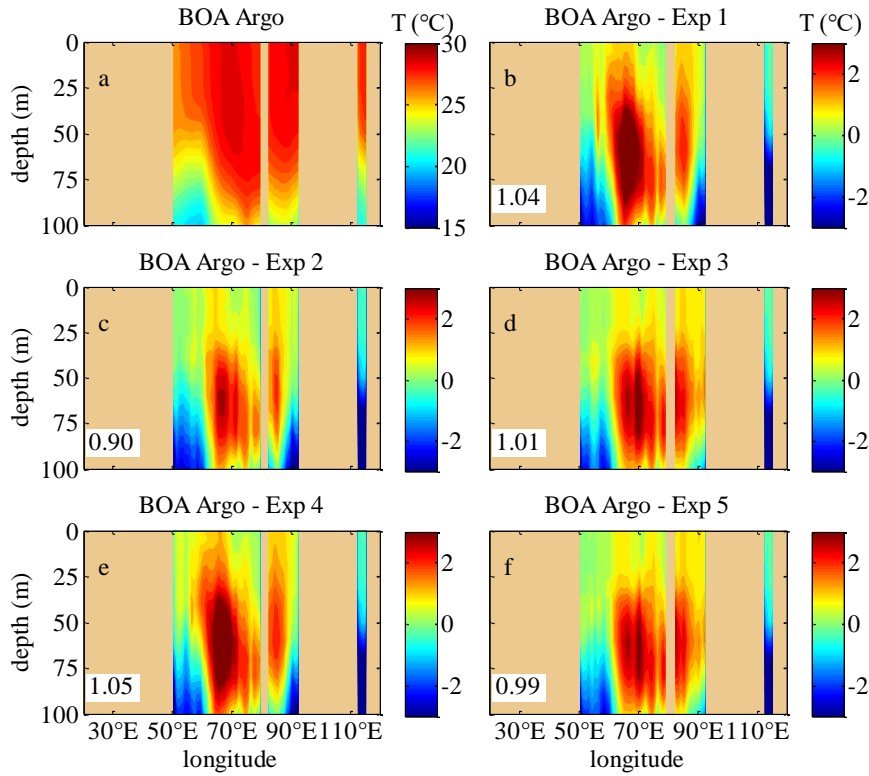

**Figure 11: The same as Fig. 10, but along 7.5° N**

**4.3 Effects on simulation of the mixed layer depth**

The mixed layer (ML), which is characterized by a quasi-uniform temperature and salinity, is crucial in understanding the physical processes in the upper ocean. The MLD variability is influenced by many processes including wind-induced turbulence, surface warming or cooling, air-sea heat exchange and turbulence-wave interaction (Chen et al., 1994; Kara et al., 2003; de Boyer Montégut et al., 2004; Abdulla et al., 2019). There are different methods to define the MLD (Kara et al., 2003). The threshold criterion, which is a widely favoured and simple method for finding the MLD (Kara et al., 2003; de Boyer Montégut et al., 2004), is used in this study. In threshold criterion, the MLD is defined as the depth at which the temperature or density profiles change by a predefined amount relative to a surface reference value. Various temperature threshold criteria were used to determine the MLD globally, such as 0.2 ℃ in de Boyer Montégut et al. (2004), 0.5 ℃ in Monterey and Levitus (1997), 0.8 ℃ in Kara et al. (2003) and 1.0 ℃ in Qiao et al. (2010). Therefore, considering the vertical temperature distribution pattern in the IO, we choose one of the typical threshold criteria (1.0 ℃) to define the MLD and attempt to make the effects of the NBSWs and ITs on the simulated MLD more obvious. In this study, the MLD is defined as a depth where the temperature is lower than the SST by 1.0 ℃ and used to show the upper-ocean thermal structure.

Figures 12 shows the comparisons of the MLDs between the WOA13 data and the model results in January. The MLDs for Exp 1 are generally shallower than the WOA13 in the whole IO and the Southern Ocean because of insufficient simulated mixing processes, which leads to underestimation of the vertical mixing in the upper ocean, especially during summer. The conclusion is similar to the global and regional simulations in previous studies (Kantha and Clayson, 1994; Qiao et al., 2010; Wang et al., 2019; Zhuang et al., 2020). The accumulation of the weak vertical mixing during the 10-year climatologic

modeling will make more heat staying in the surface layer, which will lead to warmer SST and shallower MLDs. In fact, from Figs. 12 one can see that, the obviously shallower MLDs are generally in the ACC regions, where the simulated vertical mixing from the original experiment is weak dramatically. In addition to the ACC regions, the obviously shallower MLDs also appear in the east regions of the Arabian Sea because of the weak vertical mixing. Furthermore, the simulated MLDs in most of the tropical and southern regions of the IO are shallower partially than the WOA13 data. Adopting the threshold criterion of 1.0 ℃,

the simulated MLDs were shallower than the WOA13 data because of the warmer SST and cooler temperature in the ocean interior. Comparisons among the MLDs for Exp 1 – Exp 3 show that the NBSW and IT may enhance the upper-ocean mixing and make the simulated MLDs closer to the WOA13 data. The MLs for Exp 2 and Exp 3 are extremely deepened especially in the tropical IO and the Southern Ocean. The RMSEs of the MLDs between the WOA13 data and the model results decrease 13.2% and 14.6% from 21.9 m (Exp 1) to 19.0 m (Exp 2) and to 18.7 m (Exp 3), respectively. However, the effects of the

WTFR seem to be trivial in the whole area, because there is almost no improvement from Exp 4/5 to Exp 1/3. The RMSEs for Exp 4/5 are larger than that for Exp 1/3, and the larger deviations in Exp 4/5 mostly occur in the Southern Ocean, implying that the WTFR does not work well in the Southern Ocean in austral winter.

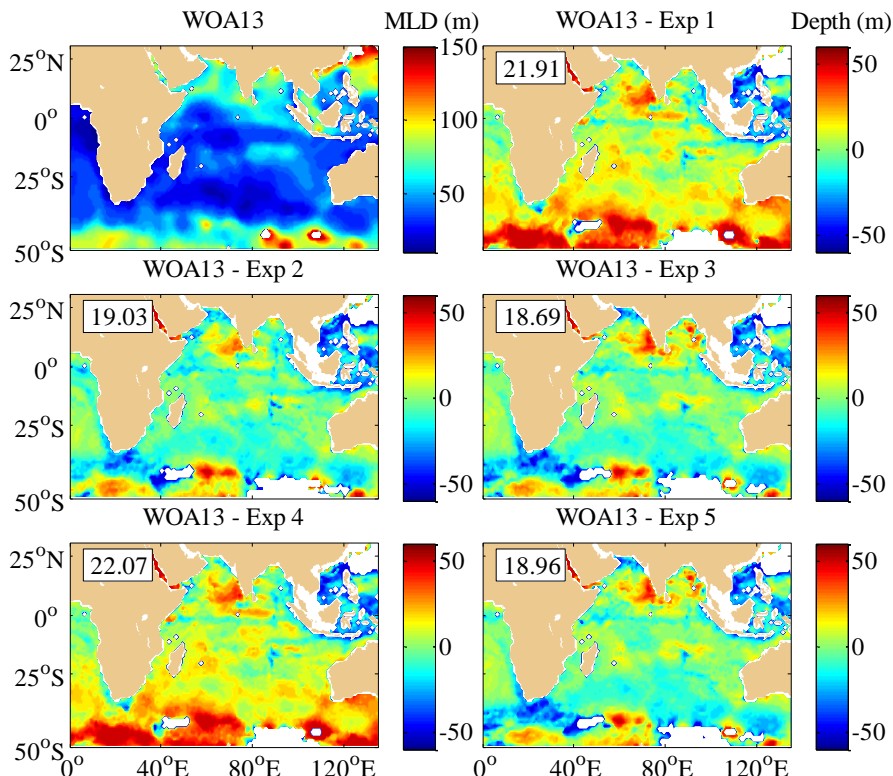

**Figure 12: The distribution of the MLD calculated from WOA13 data and the differences of the MLD between the WOA13 data**
**and the results simulated in Exp 1 – Exp 5. RMSEs of the MLD are shown on the upper left corner of the sub-figures. Deep yellow**
**and white areas correspond to the lands and the calculated MLDs are deeper than 150 m, respectively**

### 4.4 Effects on simulation of ocean currents

In this subsection, the simulated horizontal velocities in the surface layer are analyzed to evaluate the effects of the
NBSWs and the ITs on the ocean currents. The Previous studies indicated that the NBSWs and ITs have complicated impacts
on the simulated currents for the OGCMs (e.g. Huang and Qiao, 2010; Wu et al., 2019). Only the results simulated in Exp 1 –
Exp 3 are discussed in detail and the results in Exp 4 and Exp 5 are omitted here because the effects of the WTFR-induced
mixing are relatively small, this situation is similar to the simulated temperature structure and MLDs in Section 4.2 and 4.3.

Figures 13 show the comparisons of the surface velocities between the monthly mean OSCAR data (Bonjean and
Lagerloef, 2002) and the model results of Exp 1 – Exp 3 in January and July. The simulated surface velocities are chosen as
those at the depth of 2 m interpolated by the model results. The OSCAR surface current products with a horizontal resolution
of 1°/3 ×1°/3 and a time resolution of 5 days are constructed from the altimeter SSH, scatterometer winds, and both radiometer
and in situ SST. The velocities are calculated based on a combined formulation including geostrophic balance, Ekman-
Stommel dynamics and a complementary term for SST gradients (Bonjean and Lagerloef, 2002; Dohan, 2017). Johnson et al.
(2007) demonstrated that the OSCAR products are able to provide accurate estimates of the surface time-mean circulation. A

29-year (1993 - 2021) time series of the OSCAR surface currents data is collected, and used to calculate the monthly mean climatologic currents, which are regarded as the reference in the comparison. In Figs. 13a and 13b, the spatial distribution of the surface current fields is presented as the anticyclonic subtropical gyre in the SIO and the monsoonally forced circulation of the tropics and the NIO (clockwise in boreal summer and anti-clockwise in boreal winter). The eastern boundary current (Leeuwin Current) is not obvious because of too small magnitude of the velocities in the climatologic data. Figures 13c – 13h

present the difference by subtracting the monthly OSCAR data from the mean results of the three experiments. Generally, the simulated results contain most features of the surface currents in the IO, however, the simulated velocities are relatively too large to the southwest of the Indian Peninsula in January, and too small in the Somali Current and the ACC regions in both January and July. The reason should be that the spatio- and temporal-resolution and the accuracy of the surface forcing data are insufficient. One can see that there is only a little difference of the simulated currents among the three experiments. The

relatively smallest RMSEs for Exp 3 indicate that the NBSW and the IT are able to improve the simulated surface currents.

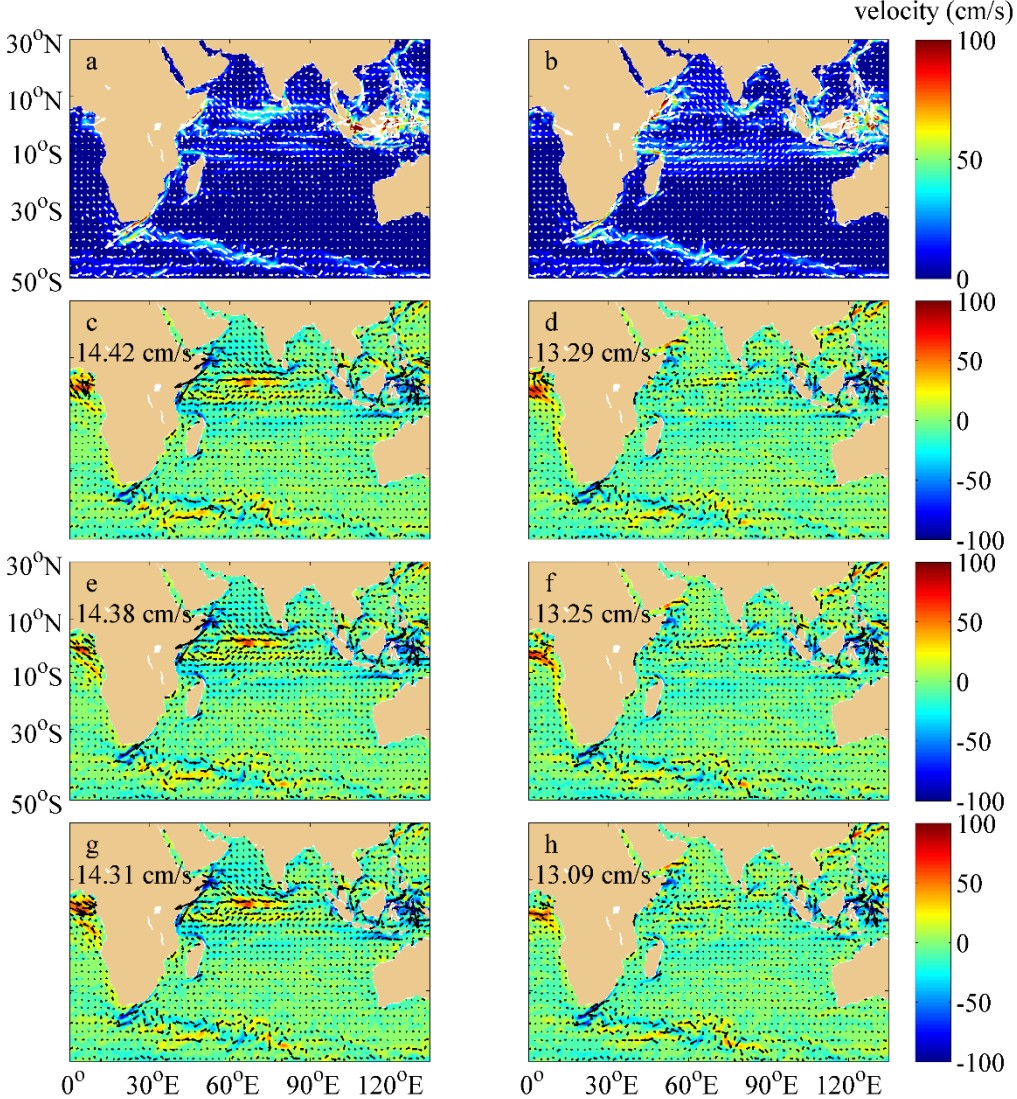

**Figure 13: The distribution of the surface currents from the OSCAR climatologic data and the differences by subtracting the OSCAR data from the mean results from Exp 1 (c, d), Exp 2 (e, f) and Exp 3 (g, h) in January (c, e and g) and July (d, f and h). RMSEs of the surface velocities are shown on the upper left corner of the sub-figures. White arrows in Sub-figures a and b represent the surface current vectors, and black arrows in Sub-figures c – h represent the surface current difference vectors. Deep yellow areas correspond to the lands**

Furthermore, we calculated the three-dimensional vertical vorticity and the eddy kinetic energy (EKE) in Exp 1 – Exp 5 to evaluate the effects of the mixing induced by the NBSWs and the ITs on the meso-scale eddy activity. However, the difference of the vertical vorticity and the EKE among the five experiments was too complicated to summarize some dynamic processes and physical mechanisms. The reason should be that the climatological modeling in this study, on one hand, may be inappropriate to analyze the meso- or small-scale processes because of the relatively coarse resolution and smoothed surface

forcing, open boundary conditions and topography data; on the other hand, the induced vertical mixing may not be a key mechanism for the eddy activity, previous studies indicated that the surface waves affect the eddies through the interaction among the turbulence, the circulation and the Langmuir circulation when the Turbulent Langmuir number is small (Jayne and Marotzke, 2002; Romero et al., 2021), and the subharmonic instability may transfer the energy from the internal tides to the shear-induced turbulent diapycnal mixing (MacKinnon and Gregg, 2005; Pinkel and Sun, 2013). Especially in the east region of the tropical Indian Ocean, the effects of the ITF on the meso- or small-scale processes have not yet been simulated exactly in the existing OGCMs (e.g. Nagai et al., 2017; Santoso et al., 2022). Additional improvements of the mixing schemes and the ocean modelling will be studied further in future.

## 5. Discussion

We evaluate the impacts of three different mixing schemes, including the NBSW-generated turbulent mixing, the WTFR-induced mixing, and the IT-generated turbulent mixing, on the upper-ocean thermal structure simulation in the IO. The comparisons of the temperature structure and the MLDs between the WOA13 data, which is regarded as the observations, and the model results imply that the simulation is significantly improved by incorporating the NBSW- and IT-generated turbulent mixing into the MASNUM ocean circulation model, but the effects of the WTFR is trivial, the simulated MLDs are even deteriorated in some regions. However, based on numerical experiments, Yang et al. (2019) demonstrated that the WTFR may play an important role in the SST cooling, if the wind and surface waves are strong. During the period of the tropical cyclone Nepartak passage, the simulated SST cooling distribution and the cooling amplitude are more consistent with the observations, if the WTFR-induced mixing scheme is incorporated, which present waring/cooling effects on the left/right sides of the typhoon track (Yu et al., 2020). The effects of the WTFR under the typhoon conditions will be further examined in the future work.

In addition, the three mixing schemes are incorporated into the MASNUM model as a part of the vertical diffusive terms, thus avoiding the issues that may be resulted from adding the mixing coefficients to those from the M-Y 2.5 scheme directly. The analysis of the numerical results indicates that the NBSW (and WTFR, sometimes) leads to improve simulations of upper-ocean temperature structure and MLDs due to the enhanced mixing that draws the warmer water from the surface to the subsurface layers with depths from about 10 m to 40 m, then the IT, which can improve the simulations further, may enhance the mixing that draws the warmer water from the subsurface layers to the ocean interior (Fig. 14). In summary, the combination of the NBSW- and IT-generated turbulent mixing results in a better match with the observations of upper-ocean temperature structure and MLDs. The mixing schemes introduced in this study contain the effects of the surface waves and internal tides, which are thought to be the supplement of the physical mechanism for the vertical mixing processes in the OGCMs, because the original turbulent mixing schemes, such as the M-Y 2.5 scheme, neglected the interaction between the surface waves and the currents (Huang and Qiao, 2010; Huang et al., 2011). The M-Y 2.5 mixing scheme combined with the NBSW- and IT-

induced mixing schemes should become more complete for modeling the vertical mixing processes. In our opinion, it is important to study the NBSW- and IT-induced mixing for promoting the development of the ocean and coupling models.


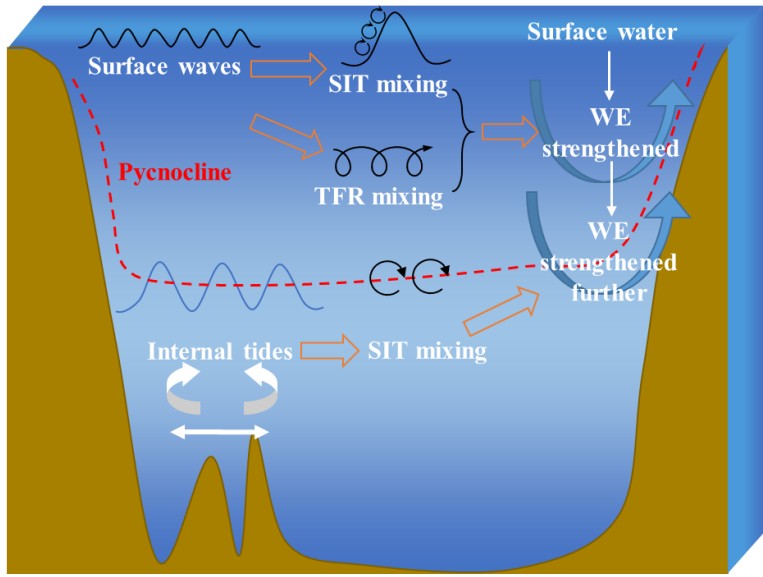

**Figure 14: Sketch of the enhancing processes of the vertical mixing induced by three different mechanisms, including the NBSW-generated turbulent mixing, the WTFR-induced mixing, and the IT-generated turbulent mixing. SIT and TFR mixing represent the shear induced turbulent and transport flux residue mixing, respectively. WE means the water exchange.**

It is worth noting that the circulation and the temperature structure of the IO have been not yet characterized by the ocean
model in the present study because of the non-ignorable difference between the WOA13 data and the simulation results. The
RMSEs in the NIO, including the Arabian Sea and the Bay of Bengal, are even generally larger than 1.2 ℃. Direct modeling
of the ITs in the experiments of this study is inappropriate. Firstly, the horizontal and vertical resolution should be insufficient
to simulate the generation and propagation of the ITs because of the relatively coarse topography and coastlines. The modeling
area of the whole IO should be also too large. Secondly, the MASNUM ocean model used in this study has not yet included
the tidal forcing and tidal open boundary conditions, so the conversion from barotropic to baroclinic energy cannot be described
exactly. Finally, the climatologic experiments are not good at simulating the ITs, because the multi-year mean surface forcing
should be very smooth and partly lack the local small- and meso-scale information. The climatologic current, temperature and
salinity input in the open boundary is also inappropriate for the IT modeling. Therefore, higher horizontal resolution and more
vertical layers, on one hand, will be designed in following experiments to describe the finer structure and features of the IO,
on the other hand, the surface forcing and the lateral boundary conditions with higher spatio- and temporal-resolution, such as
the ERA5 hourly reanalysis data (https://cds.climate.copernicus.eu/cdsapp#!/dataset/reanalysis-era5-single-levels?tab=form),
the Climate Forecast System Version 2 (CFSv2) (http://rda.ucar.edu/datasets/ds094.0/) and the global HYbrid Coordinate
Ocean Model/Navy Coupled Ocean Data Assimilation (HYCOM/NCODA) product
(https://www.hycom.org/data/glba0pt08/expt-90pt8), and more optimization and improvement of the real-time hindcast
experimental design will be used to simulate the IO more accurately. It is worth noting that the NBSW and IT can improve the

simulation obviously in the Arabian Sea, but do not always work in the Bay of Bengal, which is a hot spot for generation of the ITs. This is implies that the IT-induced mixing scheme may not be appropriate in the marginal-sea simulation containing full of the small- and meso-scale processes.

We have to admit that the issues about the simulation in the IO cannot be solved entirely when the NBSW- and IT-induced mixing schemes are adopted, but it should be more convenient to improve the ocean modeling further because the mixing schemes are independent with the ocean models. A multi-scale-process coupling model, including atmosphere, ocean current, tide, surface-wave, and internal-wave/tide component models, will be established in future for accurate and high-resolution ocean and atmosphere modeling. The NBSW- and IT-induced mixing schemes and the related results in this study are helpful and valuable for establishing the coupling model.

## 6. Conclusions

This study uses the MASNUM ocean circulation model for testing and validating the effects of three different mixing schemes, including the NBSW-generated turbulent mixing, the WTFR-induced mixing and the IT-generated turbulent mixing, on the upper-ocean thermal structure simulation in the IO. The major findings are summarized as follows.

1). The diffusive terms calculated by the NBSW-generated turbulent mixing is dominant if the depth is less than 30 m, while the WTFR-induced mixing is extremely weak because the values are about 4 to 6 orders smaller than the NBSW. In the ocean interior with depths from 40 m to 130 m, the diffusive terms calculated by the IT-generated turbulent mixing are the largest ones in regions with large topographic relief.

2). The effects of these schemes on the upper-ocean simulation are tested. The results show that the simulated thermal structure and MLDs and surface currents are both improved by the NBSW, because of the enhanced mixing in the sea surface, while the effects of the WTFR is trivial.

3). The IT may strengthen the vertical mixing of the ocean interior and improve the simulation further. In summary, the combination of the NBSW and IT may strengthen the vertical mixing and improve the upper-ocean simulation.

## Appendix A: The velocity shear module of the internal tide

The internal-tide-induced mixing plays an important role in the vertical and horizontal distribution of water mass properties. Based on the Navier-Stokes equations, the solvability simplification is realized based on the spatio-temporal scale, controlling mechanism and actual characteristics of the ITs. The IT is considered to be weakly nonlinear, the shear terms of the larger-scale motions in the equations are linear approximately, and the molecular and turbulent mixing terms in the equations are too small to be ignored. The f-plane and layered approximation for the larger-scale motions are also adopted into the equations. Thus, the simplified equations can be written as

$$\frac{\partial u_{IT1}}{\partial x_1} + \frac{\partial u_{IT2}}{\partial x_2} + \frac{\partial u_{IT3}}{\partial x_3} + \gamma u_{IT1} = 0 , \tag{12}$$

$$\frac{\partial u_{IT1}}{\partial t} - f u_{IT2} - 2U_1 \gamma u_{IT2} = -\frac{\partial}{\partial x_1}\left(\frac{p_{IT}}{\rho_0}\right) , \tag{13}$$

$$\frac{\partial u_{IT2}}{\partial t} + \left( f + 2\gamma U_2 + \frac{\partial U_2}{\partial x_1} \right) u_{IT1} + \frac{\partial U_2}{\partial x_3} u_{IT3} = -\frac{\partial}{\partial x_2}\left(\frac{p_{IT}}{\rho_0}\right) , \tag{14}$$

$$\frac{\partial u_{IT3}}{\partial t} = -\frac{\partial}{\partial x_3}\left(\frac{p_{IT}}{\rho_0}\right) - g\left(\frac{\rho_{IT}}{\rho_0}\right) , \tag{15}$$

$$\frac{\partial}{\partial t}\left(\frac{\rho_{IT}}{\rho_0}\right) + \frac{\partial}{\partial x_1}\left(\frac{\rho_{IT}}{\rho_0}\right) u_{IT1} + \frac{\partial}{\partial x_3}\left(\frac{\rho_{IT}}{\rho_0}\right) u_{IT3} = 0 , \tag{16}$$

where $u_{IT i}$ ($i$=1, 2, 3), $\rho_{IT}$ and $p_{IT}$ denote the three dimensional velocity, density and pressure of the internal tide, respectively. $f$ is the Coriolis parameter, $\gamma$ is regarded as the curvature of the larger-scale motions including mesoscale eddies, gyre and so on. $U_i$ represent the velocity of the larger-scale motions. The Fourier kernel function is used to transform the differential equations (12) to (16) to the algebraic equations, for example, the relation between $u_{IT1}$ and its Fourier kernel function $\mu_{IT1}$ can be expressed as

$$u_{IT1} = \iint\limits_{\vec{k}} \eta_{IT} \mu_{IT1} \exp\{i(k_1 x_1 + k_2 x_2 - \omega t)\} dk_1 dk_2 , \tag{17}$$

where $\eta_{IT}$ is the SSH amplitude of the internal tide, $\omega$ is the frequency. The dispersion relation between the frequency and the wavenumber can be written as

$$\omega^2 \left(\frac{N_0^2}{\omega^2} - 1\right)^{1/2} = g\left(\frac{\rho_{IT}}{\rho_0}\right) k \frac{\sin\left\{\int\limits_{-H}^{0}\left(\frac{N^2}{\omega^2} - 1\right)^{1/2} k dx_3\right\}}{\cos\left\{\int\limits_{-H}^{0}\left(\frac{N^2}{\omega^2} - 1\right)^{1/2} k dx_3\right\}} , \tag{18}$$

$$\frac{\partial \omega}{\partial x_3} = 0 . \tag{19}$$

Based on the analytical expression of the three dimensional velocity (derived from the Fourier kernel functions) and Eqs. (18) and (19), the velocity shear module can be expressed analytically as

$$\left(\sum_{i=1}^{3}\sum_{j=1}^{3}\left|\frac{\partial u_{IT i}}{\partial x_j}\right|\right)^2 = \left\{2\iint\limits_{\vec{k}} \eta_{IW} k^2 \omega^2 \frac{\left(\frac{N^2}{2\omega^2}\right)^2 - \left(\frac{N^2}{2\omega^2} - 1\right)^2 \cos\left\{2\int\limits_{-H}^{x_3}\left(\frac{N^2}{\omega^2} - 1\right)^{1/2} k dx_3\right\}}{\sin^2\left\{\int\limits_{-H}^{0}\left(\frac{N^2}{\omega^2} - 1\right)^{1/2} k dx_3\right\}} dk_1 dk_2\right\}^{1/2} , \tag{20}$$

which is consistent with Eq. (5). Similar to Gregg (1989) and Gregg and Kunze (1991), the mixing terms including the viscosity and diffusivity can be calculated from the velocity shear modules as shown in Eq. (4).


*Code and data availability*. The MASNUM ocean circulation and wave spectrum models can be downloaded at https://doi.org/10.5281/zenodo.6717314 (Han et al., 2022) and https://doi.org/10.5281/zenodo.6719479 (Yuan et al., 2022), respectively. All configuration scripts, pre-processing and post-processing subroutines are included in these repositories. The data used in the ocean modeling, including the topography, the surface forcing, the open boundary, etc., and the results can
be downloaded at https://doi.org/10.5281/zenodo.6749788 (Zhuang, 2022).

*Author contributions*. ZZ wrote the paper with the help of all the co-authors. QZ, YY and ZS provided constructive feedback on the manuscript. YY designed and developed the theoretical basis of the improved vertical mixing scheme. CZ, XZ, TZ and JX gave help and advice in data processing and numerical experiments.


*Competing interests*. The authors declare that they have no conflict of interest.

*Financial support*. This work is supported by the National Natural Science Foundation of China (Grant No. 42106031), Basic Scientific Fund for National Public Research Institutes of China (Grant No. 2020Q04), National Programme on Global Change
and Air-sea Interactions (phase II) - "Distribution and Evolution of Ocean Dynamic Processes" (Grant No. GASI-04-WLHY-01), National Program on Global Change and Air-Sea Interaction (Phase II) - "Parameterization assessment for interactions of the ocean dynamic system" (Grant No. GASI-04-WLHY-02), Shandong Provincial Natural Science Foundation, China (Grant No. ZR202102240074), National Natural Science Foundation of China (Grant No. 42006008).

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
