# Peer review of "Improved upper-ocean thermo-dynamical structure modeling with combined effects of surface waves and $M_2$ internal tides on vertical mixing: a case study for the Indian Ocean"

_Geoscientific Model Development, 2022_

## Referee Comment (RC1)

**Title:** "Improved ocean circulation modeling with combined effects of surface waves and M2 internal tides on vertical mixing: a case study for the Indian Ocean"

**Summary**

The main result of this study is to investigate the contribution of surface wave- and internal tides-induced vertical mixing in the upper ocean of the Indian Ocean. By adding three mixing schemes (vertical diffusive terms) into ocean circulation model, namely non-breaking surface-wave-generated turbulent mixing, the mixing induced by the wave transport flux residue, and the internal-tide-generated turbulent mixing schemes, the role of three vertical mixing schemes is quantified by switching off each diffusive term. Especially, the surface wave mainly improves the vertical mixing in the sea surface while the internal tide mainly contributes to the vertical mixing in the ocean interior. Improvement of upper ocean temperature structure is observed when all three schemes are combined.

**Recommendation:** Major revision.

The authors have presented a clear description of three mixing schemes, model experiments, and results that shows significant improvement of upper ocean thermal structure in the Indian Ocean. The study would be of importance to local oceanography, and it will also help improving the vertical mixing parameterization scheme that is used in the low-resolution models. However, a first impression of this paper is that the topic, i.e., what scientific object the authors are intended to improve, is not very clear, which could be related to the presentation of the introduction. The results also need more explanation. I would recommend major revision this time.

Here are some specific points below:

**Major point:**

1. The introduction part is not well-written, especially with respect to the relationship between vertical mixing process and upper ocean dynamic and thermodynamic processes in the Indian Ocean. For example, the paragraph from Lines 65 to 75 mainly introduces the importance of surface wave and internal tide in the Indian Ocean. But, how they are related to the upper thermal structure, and most importantly, the upper ocean dynamics, haven't been well-documented in this paragraph. In 2.4, the authors have clearly stated that the three vertical mixing schemes has been added to the momentum equation, which means that the ocean dynamic process is also influenced by three mixing schemes. Therefore, the relationship between vertical mixing (due to surface wave and internal tide) and upper ocean dynamic process should be documented in the introduction.

2. Although the authors have stated the reason for not presenting the circulation pattern in the Indian Ocean (Lines from 350 to 355), the performance of ocean dynamic process should be presented (at least discussed) in the Result Section. As mentioned in the Major point 1, the momentum equation in the ocean model of this study is also influenced by three mixing schemes, and it in turn affects the thermal dynamic process (upper ocean temperature structure). Hence, the role of those three schemes on the ocean dynamic process, e.g., general circulation (mean state), and eddy activity (anomaly), should be investigated. Especially for the latter, since it is directly related to the parameterization of turbulent mixing, and is an important factor to the tracer conservation (Jayne and Marotzke, 2002).

Reference:

Jayne, S. R., and Marotzke, J. 2002: The oceanic eddy heat transport. J. Phys. Oceanogr., 32(12), 3328–3345. doi: 10.1175/1520-0485(2002)032<3328:TOEHT>2.0.CO;2.

**Minor points:**

1. Line 60. Why choose M2 internal tides?

2. Lines 155-160: A short description of the Mellor-Yamada scheme is needed for the readers to understand the difference between the experiment 1 and other experiments in the Result Section.

3. Line 225: Please provide the reason why you choose the 10.5°S Section.

4. Line 245: The EIO only occurs once in this manuscript. Hence it is not necessary to define this abbreviation.

5. The thermocline structure at the equatorial region (normally defined as the depth of the 20 degree temperature, referred as Z20C) is one of the distinctive feature in the Indian Ocean. Because of the weak westerly at the equator, the Indian Ocean shows deeper and reversed slope of Z20C as compared to its counterpart in the Pacific Ocean. The mean state structure of Z20C is very important for the Indian Ocean air-sea interaction (e.g., Indian Ocean Dipole). Therefore, it would be worthwhile to find out how the Z20C responds to those three mixing schemes. I would suggest the authors to show the upper-ocean thermal structure at the Equator in Section 4.2.

6. The fontsize of text in some figures should be made larger (e.g., Figure 8, Figure 9).

---

## Referee Comment (RC2)

**"Improved ocean circulation modeling with combined effects of surface waves and M2 internal tides on vertical mixing: a case study for the Indian Ocean" By Zhuang et al.**

In this study, the authors incorporated three mixing schemes into the ocean general circulation model, namely non-breaking surface-wave-generated turbulent mixing(NBSW), the mixing induced by the wave transport flux residue(WTFR), and the internal tide-generated turbulent mixing(IT) along with Mellor-Yamada 2.5 mixing scheme. This study of quantifying the role of wave and tide-induced mixing in an ocean model is a timely and valuable contribution. However, the authors are unable to represent it in terms of value addition to its scientific contributions. There are many gaps in this study starting with ocean model configurations and their different experiments. The introduction lacks the present status of the state of the art model's mixing schemes with details and its drawbacks in the Indian Ocean. The authors are unable to give the scientific objectives to be achieved in this study as compared to the previous works. The representation of the internal tide-generated turbulent mixing is not new, in fact, it's been introduced by Simmons et al. (2004) in a global Ocean General Circulation model. The author did not mention this work and its related works (Nagai and Hibiya (2015). Also, the authors presented the results only up to 130 m which does not represent insight into the mixing process related to internal tides since its effect could be seen in the deeper layers.  A very recent study by Lozovatsky et al. (2022) showed that the observed eddy diffusivity in the ocean pycnocline over the southeastern BoB is likely related to internal-wave generated turbulence. In line-121-22 the authors wrote "….., the mode-1 M2 internal tides, which mainly originate from regions with steep topographic gradients, are considered…." . Doesn't it imply that the mixing will be more over the steep topographic gradients?. But the author did not show any results related to this. The authors implemented the mixing schemes in the momentum equations.  This implementation will also affect the dynamics as well. But the authors did not show any results on whether any changes are there in the circulations. The authors should show a few results about how the upper ocean currents improved with implementations of NBSW, IT, and WTFR mixing schemes. It will be good if the authors also can show spatial comparisons of model-simulated temperature diffusivities with Argo observations (Whalen et al. 2012).  I am unable to recommend this manuscript for publication in this form. However, it can be considered for publication if they address my above queries and the below comments.

1. Line 173-174: "The initial temperature and salinity are interpolated based on the annually mean Levitus data with the horizontal resolution of 1° by 1° and 33 vertical layers.." Which Levitus data authors have used? Should give the version and reference.
2. The author used a regional model in which the lateral boundary condition is very important for any basin-scale model, particularly for the Indian Ocean which is affected by the Indonesian Throughflow in the eastern boundary. The author did not give any

details about how the boundary condition is prescribed. Is it a boundary condition with a sponge layer? The authors should provide the details about the lateral boundary conditions used in this study.

3. Line 175-180: The initialization strategy and the experimental details are also not very clear. It looks like the author used a cold start and then inter-annual forcing from NCEP/NCAR (1948-2021). This means its inter-annual simulations. On the other hand, they wrote "The model is integrated from the quiescent state for 10 climatological years. The simulated temperature in the last 1 year is compared with the monthly World Ocean Atlas 2013 (WOA13) climatologic data". This implies it's only 10 years of simulations. It's confusing what experiments the authors exactly carried out. It seems 10 years of simulation may not be sufficient to reach the steady-state. The authors should give the evidence that the model reached steady-state in $10^{th}$ year of simulation.

4. The author used MASNUM wave spectrum model simulations to get the inputs for the NBSW parameterizations scheme they incorporated. But how good the model simulations compare with observations?

5. In Figures 2c and 3c authors represented it as the IT-generated turbulent mixing scheme based on Exp-3 but in this experiment, NBSW is also included, then how can it be an IT-generated turbulent mixing scheme?

6. In Figures 2 and 3 for the vertical profiles of the monthly mean vertical temperature diffusive terms, the author choose to show the results for 10.5 °S, and for the temperature comparison, they showed 30.5 °S. What is the physical basis to choose these sections? Authors should show such results for the Arabian Sea and Bay of Bengal as well.

7. In Figure 4 in exp1& 4 why the model does show the cooler temperature in the thermocline depth region? In general, over the Indian Ocean, almost all forced model shows warm bias (Rahaman et al. 2020). Although the thermocline bias was reduced in Exp 2 and 3, it became reversed with similar magnitude why does it so? Why there is no difference between exp-1 and exp-4 in Figures 4 and 5? Does it mean WTFR does not impact temperature simulations? Authors should show such results for the Arabian Sea and Bay of Bengal as well.

8. Figure 8 What is the physical basis of choosing the different zone? Looks like the present defined zones will not give true representation, for example in zone 1 since the dynamics and thermodynamics are different in the Arabian Sea, Bay of Bengal and South China Sea, hence the mixing characteristics are also different. I suggest excluding the regions outside of the Indian Ocean such as South China Sea and Atlantic Ocean as included in the present zone 2 and zone 3. I also suggest the author should select the zones based on past studies or based on the dynamics and thermodynamic properties of the Indian Ocean basin.

9. How the RMSE is statistically robust when the authors used the seasonal cycle and computed the RMSE?

10. As already pointed out in the case of the thermocline in the MLD bias given in Figure 9 for Exp-1 too looks not consistent with the previous works. In general OGCMs simulates deeper MLD in the Indian Ocean (de Boyer Montégut et al. 2007). A very recent study by Pottapinjara et al. (2022) too shows similar results. Hence, how the MLD simulation, in this case, shows shallower than observations? The authors need to explain why the model simulated MLD is shallower as compared to observations.  Also, the criteria used to compute MLD is not very widely used. The authors did not provide any reference to compute MLD or any explanation why they choose the 1 °C criterion to compute MLD.

---

## Author Comment (AC2)

**Response to Comments from Referee #1**

We thank the referee for careful reviews and constructive comments in improving the original manuscript. Below is our point-to-point reply to these comments (the reviewer's comments are in blue, and out responses are in black).

Summary

The main result of this study is to investigate the contribution of surface wave- and internal tides-induced vertical mixing in the upper ocean of the Indian Ocean. By adding three mixing schemes (vertical diffusive terms) into ocean circulation model, namely non-breaking surface-wave-generated turbulent mixing, the mixing induced by the wave transport flux residue, and the internal-tide-generated turbulent mixing schemes, the role of three vertical mixing schemes is quantified by switching off each diffusive term. Especially, the surface wave mainly improves the vertical mixing in the sea surface while the internal tide mainly contributes to the vertical mixing in the ocean interior. Improvement of upper ocean temperature structure is observed when all three schemes are combined.

Recommendation: Major revision.

The authors have presented a clear description of three mixing schemes, model experiments, and results that shows significant improvement of upper ocean thermal structure in the Indian Ocean. The study would be of importance to local oceanography, and it will also help improving the vertical mixing parameterization scheme that is used in the low-resolution models. However, a first impression of this paper is that the topic, i.e., what scientific object the authors are intended to improve, is not very clear, which could be related to the presentation of the introduction. The results also need more explanation. I would recommend major revision this time.

According to the reviewer's suggestion, the topic has been revised as "Improved

upper-ocean thermo-dynamical structure modeling with combined effects of surface waves and $M_2$ internal tides on vertical mixing: a case study for the Indian Ocean" which should be clearer to show the scientific object we are intended to improve in this study. In addition, more explanation has been added in the results. The Sub-section "4.4 Effects on simulation of ocean currents" has been added in the revised version to evaluate the effects on the ocean dynamic structure of the NBSWs and the ITs. Please refer to the revised manuscript for details.

Here are some specific points below:

Major point:

**1.** The introduction part is not well-written, especially with respect to the relationship between vertical mixing process and upper ocean dynamic and thermodynamic processes in the Indian Ocean. For example, the paragraph from Lines 65 to 75 mainly introduces the importance of surface wave and internal tide in the Indian Ocean. But, how they are related to the upper thermal structure, and most importantly, the upper ocean dynamics, haven't been well-documented in this paragraph. In 2.4, the authors have clearly stated that the three vertical mixing schemes has been added to the momentum equation, which means that the ocean dynamic process is also influenced by three mixing schemes. Therefore, the relationship between vertical mixing (due to surface wave and internal tide) and upper ocean dynamic process should be documented in the introduction.

As the reviewer suggested, the effects of the surface waves and internal tides on the ocean dynamic processes have been documented in the 'Introduction' Section. The relationship between the surface wave and internal tide-induced mixing and the dynamic processes in the Indian Ocean has been also introduced.

Generally, the turbulent mixing induced by the wave-current interaction can improve the large-scale ocean circulation modeling. For the small- and meso-scale motions, the effects of the surface waves were also significant by modifying the surface

current gradient variability and the eddy transport when the Turbulent Langmuir number is small, and the effects will become larger when the model resolution increases.

The internal wave energy in the ocean interior, which generates the turbulence processes and the diapycnal diffusivity is redistributed from large- to small-scale motions by the wave-current interactions. The parameterization of the turbulent mixing induced by the internal waves was introduced into the ocean models and makes the simulated mixing coefficients and dynamic processes, including horizontal currents and meridional overturning circulation, agree better with the Large Eddy Simulation (LES) results or observations than the original schemes.

Please refer to the revised manuscript for details, of which the content is marked in red.

**2.** Although the authors have stated the reason for not presenting the circulation pattern in the Indian Ocean (Lines from 350 to 355), the performance of ocean dynamic process should be presented (at least discussed) in the Result Section. As mentioned in the Major point 1, the momentum equation in the ocean model of this study is also influenced by three mixing schemes, and it in turn affects the thermal dynamic process (upper ocean temperature structure). Hence, the role of those three schemes on the ocean dynamic process, e.g., general circulation (mean state), and eddy activity (anomaly), should be investigated. Especially for the latter, since it is directly related to the parameterization of turbulent mixing, and is an important factor to the tracer conservation (Jayne and Marotzke, 2002).

Reference:

Jayne, S. R., and Marotzke, J. 2002: The oceanic eddy heat transport. J. Phys. Oceanogr., 32(12), 3328–3345. doi: 10.1175/1520-0485(2002)032<3328:TOEHT>2.0.CO;2.

We agree with the reviewer's suggestion about analyzing the effects of the NBSWs and ITs on the ocean dynamic processes. We have added Sub-section "4.4 Effects on simulation of ocean currents" to discuss the effects of the NBSWs and ITs on the ocean

dynamic processes. Because the WTFR-induced mixing has a little effects on the simulated results, there is little difference between the simulation with and without the WTFR-induced mixing scheme, so we only compared the results in Exp1 – Exp 3. The monthly mean Ocean Surface Current Analyses Real-time (OSCAR) data, which are able to provide accurate estimates of the surface time-mean circulation, were used in the comparison as a reference. The comparison results showed that the NBSW and the IT somewhat improve the simulated surface currents.

[Figure]

Figure: The distribution of the surface currents from the OSCAR climatologic data and the differences by subtracting the OSCAR data from the mean results from Exp 1 (c, d), Exp 2 (e, f) and Exp 3 (g, h) in January (c, e and g) and July (d, f and h). RMSEs of the surface velocities are

shown on the upper left corner of the sub-figures. White arrows in Sub-figures a and b represent the surface current vectors, and black arrows in Sub-figures c – h represent the surface current difference vectors. Deep yellow areas correspond to the lands

Furthermore, we have calculated the three-dimensional vertical vorticity and the eddy kinetic energy (EKE) in Exp 1 – Exp 5 to evaluate the effects of the NBSWs and the ITs on the meso-scale eddy activity. However, the difference among the five experiments was too complicated to summarize some dynamic processes and physical mechanisms. Therefore, the analysis of the simulated eddy activity was omitted in this study. The reason should be that the climatologic modeling in this study, on one hand, may be inappropriate to analyze the meso- or small-scale processes because of the relatively coarse and smoothed surface forcing, open boundary conditions and topography; on the other hand, the induced vertical mixing may not be a key mechanism for the eddy activity, previous studies indicated that the surface waves affect the eddies through the interaction among the turbulence, the circulation and the Langmuir circulation when the Turbulent Langmuir number is small (Jayne and Marotzke, 2002; Romero et al., 2021), and the subharmonic instability may transfer the energy from the internal tides to the shear-induced turbulent diapycnal mixing (MacKinnon and Gregg, 2005; Pinkel and Sun, 2013). Additional improvements of the mixing schemes used in this article will be studied further in future. These analysis and explanation have been added in the sub-section 4.4.

Minor points:

1. Line 60. Why choose $M_2$ internal tides?

Many thanks for this question, which is important for clarifying further the motivation of this study. In our opinions, there should be three reasons. Firstly, as one of the main tidal constituents ($M_2$, $S_2$, $N_2$, $O_1$, and $K_1$), $M_2$ internal tides have the largest energy among the semi-diurnal tides, therefore, the turbulence generated by $M_2$ internal tides should be dominant and typical for studying the mixing processes. Secondly, $M_2$ internal tides are ubiquitous in the world oceans, and lose little energy in propagating

across its critical latitudes (28.88S/N) (Zhao et al., 2016). Finally, this study is still preliminary research on the contribution of surface wave- and internal tides-induced vertical mixing in the upper ocean, we just chose one of the main tidal constituents to test the effects. Other constituents such as $S_2$, $N_2$, $O_1$, and $K_1$ will be evaluated in future. The explanation has been added in the revised manuscript.

2. Lines 155-160: A short description of the Mellor-Yamada scheme is needed for the readers to understand the difference between the experiment 1 and other experiments in the Result Section.

A short description of the M-Y 2.5 scheme has been added in the revised manuscript. The M-Y 2.5 scheme is a level-2.5 turbulence model based on the modification of the material derivative and diffusion terms. The mixing coefficients $K_m$ and $K_h$ can be calculated as the turbulence characteristics multiplied by a stability function associated with the Richardson number. One of the major deficiencies of the M-Y 2.5 scheme is the neglect of the surface-wave effect. Please refer to the revised version for details.

3. Line 225: Please provide the reason why you choose the 10.5°S Section.

There are two reasons for the choice of the 10.5°S Section. Firstly, the Madagascar-Mascarene regions (0° ~ 25°S in the West Indian Ocean) are considered to be a hot spot for generation of semidiurnal internal tides. Both of the surface waves and internal tides should grow fully and become large enough for the comparison of the diffusive terms. Secondly, the 10.5°S Section is typical to the different pattern among the diffusion terms obviously, which include that, the $k_h$ calculated from the classic M-Y 2.5 scheme is relatively small, the NBSW-generated turbulent mixing is the largest in the ocean surface and decays exponentially with the depth, and the IT-generated turbulent mixing is obviously larger than other schemes in the ocean interior. The explanation has been added in the revised version.

4. Line 245: The EIO only occurs once in this manuscript. Hence it is not necessary to define this abbreviation.

We agree with the reviewer and accept the suggestion. The abbreviation "EIO" has been deleted in the revised version.

5. The thermocline structure at the equatorial region (normally defined as the depth of the 20 degree temperature, referred as Z20C) is one of the distinctive feature in the Indian Ocean. Because of the weak westerly at the equator, the Indian Ocean shows deeper and reversed slope of Z20C as compared to its counterpart in the Pacific Ocean. The mean state structure of Z20C is very important for the Indian Ocean air-sea interaction (e.g., Indian Ocean Dipole). Therefore, it would be worthwhile to find out how the Z20C responds to those three mixing schemes. I would suggest the authors to show the upper-ocean thermal structure at the Equator in Section 4.2.

Many thanks for the reviewer's constructive comments about the thermocline structure. The variation of the depths of 20 °C isothermal (Z20) have been discussed in Subsection 4.2. As another indicator of the thermal structure in the upper-100 m layers, the depths of 26 °C isothermal (Z26) are also analyzed. New figures (Figures 9 in the revised version) are plotted to show the comparison of the Z20 and Z26 depths along the equator from the WOA13 data and the model results. One can see that the Z20 and Z26 depths are both shallow in the west and deep in the east, the simulation of the thermal structure in the five experiments depict this pattern successfully. Comparisons of the RMSEs show that the NBSW- and IT-generated turbulence mixing can improve the simulated Z26 depths as two of the key factors, but have a negative effects on the Z20 depth modeling, the reason is that the Z20 isothermal simulated in the benchmark experiment (Exp 1) is generally deeper than the WOA13 data, this should be caused by inaccurate topography and open boundary conditions, more optimization and improvement of the experimental design will be implemented in future work to make the simulated results more accurate. Please refer to the revised version for details.

[Figure]

Figure: The comparison of the Z20 and Z26 depths along the equator from the WOA13 data and the model results. a and b: the Z20 (solid curves) and Z26 (dashed curves) depths from the WOA13 data and the model results (Exp 1 ~ Exp 5) in January and July, respectively. The RMSEs of the Z26 (c) and Z20 (d) depths along the equator are also plotted.

6. The fontsize of text in some figures should be made larger (e.g., Figure 8, Figure 9).

Figures 2 ~ 9 in the original manuscript have been replotted to be more clear and more readable.

---

## Author Response (AR1)

**A point-by-point listing of response for each of the reviewers' and the Chief Editor's comments**

We thank the reviewers and the Chief Editor for careful reviews and constructive comments in improving the original manuscript. Below is our point-to-point reply to these comments (the reviewers' and the Chief Editor's comments are in blue, our responses are in black, and our changes in manuscript are in red).

**All of the changes in the revised manuscript are marked using 'track changes' in Word. Please refer to the marked-up manuscript version showing the changes made in detail.**

**Response to Comments from Referee #1**

Summary

The main result of this study is to investigate the contribution of surface wave- and internal tides-induced vertical mixing in the upper ocean of the Indian Ocean. By adding three mixing schemes (vertical diffusive terms) into ocean circulation model, namely non-breaking surface-wave-generated turbulent mixing, the mixing induced by the wave transport flux residue, and the internal-tide-generated turbulent mixing schemes, the role of three vertical mixing schemes is quantified by switching off each diffusive term. Especially, the surface wave mainly improves the vertical mixing in the sea surface while the internal tide mainly contributes to the vertical mixing in the ocean interior. Improvement of upper ocean temperature structure is observed when all three schemes are combined.

Recommendation: Major revision.

The authors have presented a clear description of three mixing schemes, model experiments, and results that shows significant improvement of upper ocean thermal

structure in the Indian Ocean. The study would be of importance to local oceanography, and it will also help improving the vertical mixing parameterization scheme that is used in the low-resolution models. However, a first impression of this paper is that the topic, i.e., what scientific object the authors are intended to improve, is not very clear, which could be related to the presentation of the introduction. The results also need more explanation. I would recommend major revision this time.

According to the reviewer's suggestion, the topic has been revised to become clearer to show the scientific object we are intended to improve in this study. In addition, more explanation has been added in the results. The sub-section "4.4 Effects on simulation of ocean currents" has been added in the revised manuscript to evaluate the effects on the ocean dynamic structure of the NBSWs and the ITs.

The topic has been revised as "Improved upper-ocean thermo-dynamical structure modeling with combined effects of surface waves and $M_2$ internal tides on vertical mixing: a case study for the Indian Ocean". Section 4.4 has been added in Line 497 – 539 of the revised manuscript.

Here are some specific points below:

Major point:

**1.** The introduction part is not well-written, especially with respect to the relationship between vertical mixing process and upper ocean dynamic and thermodynamic processes in the Indian Ocean. For example, the paragraph from Lines 65 to 75 mainly introduces the importance of surface wave and internal tide in the Indian Ocean. But, how they are related to the upper thermal structure, and most importantly, the upper ocean dynamics, haven't been well-documented in this paragraph. In 2.4, the authors have clearly stated that the three vertical mixing schemes has been added to the momentum equation, which means that the ocean dynamic process is also influenced by three mixing schemes. Therefore, the relationship between vertical mixing (due to surface wave and internal tide) and upper ocean dynamic process should be documented

As the reviewer suggested, the effects of the surface waves and internal tides on the ocean dynamic processes have been documented in the Section 1 'Introduction'. The relationship between the surface wave and internal tide-induced mixing and the dynamic processes in the Indian Ocean has been also introduced.

Generally, the turbulent mixing induced by the wave-current interaction can improve the large-scale ocean circulation modeling. For the small- and meso-scale motions, the effects of the surface waves were also significant by modifying the surface current gradient variability and the eddy transport when the Turbulent Langmuir number is small, and the effects will become larger when the model resolution increases.

The internal wave energy in the ocean interior, which generates the turbulence processes and the diapycnal diffusivity is redistributed from large- to small-scale motions by the wave-current interactions. The parameterization of the turbulent mixing induced by the internal waves was introduced into the ocean models and makes the simulated mixing coefficients and dynamic processes, including horizontal currents and meridional overturning circulation, agree better with the Large Eddy Simulation (LES) results or observations than the original schemes.

The introduction has been added in Line 50 – 62, 76 – 85, and 100 – 103 of the revised manuscript.

**2.** Although the authors have stated the reason for not presenting the circulation pattern in the Indian Ocean (Lines from 350 to 355), the performance of ocean dynamic process should be presented (at least discussed) in the Result Section. As mentioned in the Major point 1, the momentum equation in the ocean model of this study is also influenced by three mixing schemes, and it in turn affects the thermal dynamic process (upper ocean temperature structure). Hence, the role of those three schemes on the ocean dynamic process, e.g., general circulation (mean state), and eddy activity (anomaly), should be investigated. Especially for the latter, since it is directly related to

the parameterization of turbulent mixing, and is an important factor to the tracer conservation (Jayne and Marotzke, 2002).

Reference:

Jayne, S. R., and Marotzke, J. 2002: The oceanic eddy heat transport. J. Phys. Oceanogr., 32(12), 3328–3345. doi: 10.1175/1520-0485(2002)032<3328:TOEHT>2.0.CO;2.

We agree with the reviewer's suggestion about analyzing the effects of the NBSWs and ITs on the ocean dynamic processes. The sub-section "4.4 Effects on simulation of ocean currents" has been added to discuss the effects of the NBSWs and ITs on the ocean dynamic processes. Only the results simulated in Exp 1 – Exp 3 are discussed in detail and the results in Exp 4 and Exp 5 are omitted here because the effects of the WTFR-induced mixing are relatively small. The monthly mean Ocean Surface Current Analyses Real-time (OSCAR) data, which are able to provide accurate estimates of the surface time-mean circulation, were used in the comparison as a reference. The comparison results showed that the NBSW and the IT somewhat improve the simulated surface currents.

Furthermore, we have calculated the three-dimensional vertical vorticity and the eddy kinetic energy (EKE) in Exp 1 – Exp 5 to evaluate the effects of the NBSWs and the ITs on the meso-scale eddy activity. However, the difference among the five experiments was too complicated to summarize some dynamic processes and physical mechanisms. Therefore, the analysis of the simulated eddy activity was omitted in this study. The reason should be that the climatological modeling in this study, on one hand, may be inappropriate to analyze the meso- or small-scale processes because of the relatively coarse and smoothed surface forcing, open boundary conditions and topography; on the other hand, the induced vertical mixing may not be a key mechanism for the eddy activity, previous studies indicated that the surface waves affect the eddies through the interaction among the turbulence, the circulation and the Langmuir circulation when the Turbulent Langmuir number is small (Jayne and Marotzke, 2002; Romero et al., 2021), and the subharmonic instability may transfer the energy from the

internal tides to the shear-induced turbulent diapycnal mixing (MacKinnon and Gregg, 2005; Pinkel and Sun, 2013). Additional improvements of the mixing schemes used in this article will be studied further in future.

The analysis and explanation as a new sub-section (Section 4.4) have been added in Line 497 – 539 of the revised manuscript.

Minor points:

1. Line 60. Why choose $M_2$ internal tides?

This question is very important because it is helpful for clarifying further the motivation of this study. In our opinions, there should be three reasons. Firstly, as one of the main tidal constituents ($M_2$, $S_2$, $N_2$, $O_1$, and $K_1$), $M_2$ internal tides have the largest energy among the semi-diurnal tides, therefore, the turbulence generated by $M_2$ internal tides should be dominant and typical for studying the mixing processes. Secondly, $M_2$ internal tides are ubiquitous in the world oceans, and lose little energy in propagating across its critical latitudes (28.88S/N) (Zhao et al., 2016). Finally, this study is still preliminary research on the contribution of surface wave- and internal tides-induced vertical mixing in the upper ocean, we just chose one of the main tidal constituents to test the effects. Other constituents such as $S_2$, $N_2$, $O_1$, and $K_1$ will be evaluated in future.

The explanation has been added in Line 103 – 110 of the revised manuscript.

2. Lines 155-160: A short description of the Mellor-Yamada scheme is needed for the readers to understand the difference between the experiment 1 and other experiments in the Result Section.

As the reviewer suggested, a short description of the M-Y 2.5 scheme has been added in the revised manuscript. The M-Y 2.5 scheme is a level-2.5 turbulence model based on the modification of the material derivative and diffusion terms. The mixing coefficients $K_m$ and $K_h$ can be calculated as the turbulence characteristics multiplied by a stability function associated with the Richardson number. One of the major

deficiencies of the M-Y 2.5 scheme is the neglect of the surface-wave effect.

The detailed introduction has been added in Line 202 – 209 of the revised manuscript.

3. Line 225: Please provide the reason why you choose the 10.5°S Section.

There are two reasons for the choice of the 10.5°S Section. Firstly, the Madagascar-Mascarene regions (0° ~ 25°S in the West Indian Ocean) are considered to be a hot spot for generation of semidiurnal internal tides. Both of the surface waves and internal tides should grow fully and become large enough for the comparison of the diffusive terms. Secondly, the 10.5°S Section is typical to the different pattern among the diffusion terms obviously, which include that the NBSW-generated turbulent mixing is the largest in the ocean surface and decays exponentially with the depth, and the IT-generated turbulent mixing is obviously larger than other schemes in the ocean interior.

The explanation has been added in Line 302 – 305 of the revised manuscript.

4. Line 245: The EIO only occurs once in this manuscript. Hence it is not necessary to define this abbreviation.

We agree with the reviewer and accept the suggestion.

The abbreviation "EIO" has been deleted in the revised manuscript.

5. The thermocline structure at the equatorial region (normally defined as the depth of the 20 degree temperature, referred as Z20C) is one of the distinctive feature in the Indian Ocean. Because of the weak westerly at the equator, the Indian Ocean shows deeper and reversed slope of Z20C as compared to its counterpart in the Pacific Ocean. The mean state structure of Z20C is very important for the Indian Ocean air-sea interaction (e.g., Indian Ocean Dipole). Therefore, it would be worthwhile to find out how the Z20C responds to those three mixing schemes. I would suggest the authors to show the upper-ocean thermal structure at the Equator in Section 4.2.

Many thanks for the reviewer's constructive comments about the thermocline structure. The variation of the depths of 20 ℃ isothermal (Z20) have been discussed in Subsection 4.2. As another indicator of the thermal structure in the upper-100 m layers, the depths of 26 ℃ isothermal (Z26) are also analyzed. New figures (Figures 9 in the revised version) are plotted to show the comparison of the Z20 and Z26 depths along the equator from the WOA13 data and the model results. One can see that the Z20 and Z26 depths are both shallow in the west and deep in the east, the simulation of the thermal structure in the five experiments depict this pattern successfully. Comparisons of the RMSEs show that the NBSW- and IT-generated turbulence mixing can improve the simulated Z26 depths as two of the key factors, but have a negative effects on the Z20 depth modeling, the reason is that the Z20 isothermal simulated in the benchmark experiment (Exp 1) is generally deeper than the WOA13 data, this should be caused by inaccurate topography and open boundary conditions, more optimization and improvement of the experimental design will be implemented in future work to make the simulated results more accurate.

The description and analysis have been added in Line 433 – 463 of the revised manuscript.

6. The fontsize of text in some figures should be made larger (e.g., Figure 8, Figure 9).

We agree with the reviewer and accept the suggestion.

Figures 2 ~ 9 in the original manuscript have been replotted, the fontsize of text in the figures have been made larger, the figures should become clearer and more readable.

**Response to Comments from Referee #2**

In this study, the authors incorporated three mixing schemes into the ocean general circulation model, namely non-breaking surface-wave-generated turbulent mixing(NBSW), the mixing induced by the wave transport flux residue(WTFR), and the internal tide-generated turbulent mixing(IT) along with Mellor-Yamada 2.5 mixing scheme. This study of quantifying the role of wave and tide-induced mixing in an ocean model is a timely and valuable contribution. However, the authors are unable to represent it in terms of value addition to its scientific contributions. There are many gaps in this study starting with ocean model configurations and their different experiments.

Many thanks for the reviewer's comments about the contribution of this study. More detailed description about the model configurations and experimental design has been added in the revised manuscript. The MASNUM ocean and wave models should become more practicality and repeatability. The codes have been uploaded on the Zenodo: https://doi.org/10.5281/zenodo.6717314 and https://doi.org/10.5281/zenodo.6719479, the required data, including the topography, the surface forcing, the open boundary, etc., are also uploaded on the Zenodo: https://doi.org/10.5281/zenodo.6749788. All configuration scripts, preprocessing and post processing subroutines are included in these repositories. The three mixing schemes were incorporated into the ocean model as independent subroutines, which will be applied into other ocean models conveniently.

Furthermore, addition to the scientific value, the results in this study are helpful to improve the accuracy and timeliness of the global ocean numerical prediction for the national or regional forecasting agencies, because the MASNUM ocean model is able to depict more complete physical processes. In our opinion, it is important to study the NBSW- and IT-induced mixing for promoting the development of the ocean and coupling models.

The explanation and description have been added and revised in Line 226 – 242 (the detailed description about the model configurations), 559 – 562, and 625 – 629 of the revised manuscript.

The introduction lacks the present status of the state of the art model's mixing schemes with details and its drawbacks in the Indian Ocean. The authors are unable to give the scientific objectives to be achieved in this study as compared to the previous works. The representation of the internal tide-generated turbulent mixing is not new, in fact, it's been introduced by Simmons et al. (2004) in a global Ocean General Circulation model. The author did not mention this work and its related works (Nagai and Hibiya, 2015).

Many thanks for the reviewer's constructive suggestions about the introduction of the IT-generated turbulent mixing. Previous studies, such as Simmons et al. (2004) and Nagai and Hibiya, (2015), constructed baroclinic ocean models to compute the energy flux from barotropic tides into internal waves. The Navier-Stokes equations with accurate tidal potential forcing, tidal open boundary conditions and non-hydrostatic approximation (especially for the regional modeling) can be calculated to simulate the generation, development, propagation and dissipation processes of the internal tides in high-resolution numerical experiments. The induced turbulent mixing coefficients then can be estimated in terms of the local dissipation efficiency, the barotropic to baroclinic energy conversion and the buoyancy frequency. In fact, the estimation of the IT-generated turbulent mixing in these previous studies was implicit.

On the contrary, we attempt to derive an analytic and explicit expression of the viscosity and diffusivity (mixing coefficients) induced by the NBSWs and ITs, based on the theory about the turbulence dynamics and the surface and internal wave statistics. The three mixing schemes introduced in this study can be calculated directly in terms of some parameters of the surface waves and ITs. The error caused by the calculation deviation of the internal tides will be avoided. There should be no need to simulate the ITs accurately in a high resolution modeling using the ocean circulation model.

In addition, it is worth noting that direct modeling of the ITs in the experiments of this study is inappropriate. There are three main reasons. Firstly, the horizontal and vertical resolutions should be insufficient to simulate the generation and propagation of the ITs because of the relatively rough topography and coastlines. The modeling area of the whole IO should be also too large, some regional high-resolution simulations (such as Bay of Bengal, east of the Madagascar Island) will be implemented to simulate the ITs and the related patterns in future works. Secondly, the MASNUM ocean model used in this study has not yet included the tidal forcing and tidal open boundary conditions, so the conversion from barotropic to baroclinic energy cannot be described exactly. Finally, the climatologic experiments are not good at simulating the ITs, because the multi-year mean (climatologic) surface forcing should be very smooth and partly lack the local small-scale information. The climatologic current, temperature and salinity input in the open boundary is also inappropriate for the IT modeling. Therefore, more optimization and improvement of the real-time hindcast experimental design will be implemented in future work to make the simulated results more accurate.

The explanation and discussion have been added in Line 115 – 123 and 569 – 576 of the revised manuscript.

Also, the authors presented the results only up to 130 m which does not represent insight into the mixing process related to internal tides since its effect could be seen in the deeper layers. A very recent study by Lozovatsky et al. (2022) showed that the observed eddy diffusivity in the ocean pycnocline over the southeastern BoB is likely related to internal-wave generated turbulence.

As the reviewer's suggested, we have compared the thermal structure and the velocity fields in the region with depths from 130 m to 1000 m (the region with depths larger than 200 m is often called the deep ocean). However, the difference of the simulated temperature in Exp 3 and 5 from the WOA13 data is generally the most in the whole IO, this is caused by that the simulated temperature in the deep ocean (from about 300 m to about 1000 m) is too warm. The reason is that the Haney equation (Haney, 1971) was used to modify the

climatologic surface heat flux and improve the large-scale thermal coupling of ocean and atmosphere, but a disadvantage of the Haney modifying method is the destruction of the heat balance, so excessive heat may be transmitted into the ocean interior. Some more accurate surface forcing data, such as ERA5 and GFS, will be used in future simulation. For the current simulation, the mean velocities, the eddy kinetic energy and the vertical vorticity have been calculated, but the difference among the five experiments was too complicated (irregular distribution of positive and negative difference) to summarize some dynamic processes and physical mechanisms. This should be related to thermal and salinity structure. It is worth noting that the vertical mixing in the upper ocean (0 ~ 100 m) is the main focus of this study. Additional improvements of the MASNUM model and the mixing schemes will be studied further in future.

Lozovatsky et al. (2022) demonstrated that the internal wave instabilities appear to be a dominant mechanism for generating the energetic mixing based on the analysis of the in-situ observations of the turbulent kinetic energy dissipation rate and buoyancy frequency profiles. Actually, designing a universal and flexible IT-induced mixing scheme for ocean modeling based on the in-situ observations still needs a lot of works. The three schemes introduced in this study are just preliminary research on the contribution of the upper-ocean vertical mixing.

The explanation and discussion have been added in Line 422 – 432 of the revised manuscript.

In line-121-22 the authors wrote "….., the mode-1 M2 internal tides, which mainly originate from regions with steep topographic gradients, are considered…." . Doesn't it imply that the mixing will be more over the steep topographic gradients? But the author did not show any results related to this.

As we explained above, the MASNUM ocean model has not been used to directly simulate the ITs. On one hand, the topography in the experiments has been smoothed using the dual-step five-point-involved spatial smoothing method (Han, 2014) to make the calculation more stable; on the other hand, the experiments with relatively low resolution

and climatologic surface forcing are also inappropriate to directly simulate the ITs. Therefore, the topographic gradients were not considered to be a key factor in this study. Instead of estimation of the IT-induced mixing coefficients based on the accurate and high-resolution simulation of the IT processes, the NBSW- and IT-induced mixing schemes as independent subroutines were incorporated into the ocean model in this study.

The explanation has been added in Line 222 – 225 of the revised manuscript.

The authors implemented the mixing schemes in the momentum equations. This implementation will also affect the dynamics as well. But the authors did not show any results on whether any changes are there in the circulations. The authors should show a few results about how the upper ocean currents improved with implementations of NBSW, IT, and WTFR mixing schemes.

As the reviewer and reviewer #1 suggested, we have added sub-section "4.4 Effects on simulation of ocean currents" to discuss the effects of the NBSWs and ITs on the ocean dynamic processes. Only the results simulated in Exp 1 – Exp 3 are discussed in detail and the results in Exp 4 and Exp 5 are omitted here because the effects of the WTFR-induced mixing are relatively small. The monthly mean Ocean Surface Current Analyses Real-time (OSCAR) data, which are able to provide accurate estimates of the surface time-mean circulation, were used in the comparison as a reference. The comparison results showed that the NBSW and the IT somewhat improve the simulated surface currents.

The analysis and explanation as a new sub-section (Section 4.4) have been added in Line 497 – 539 of the revised manuscript.

It will be good if the authors also can show spatial comparisons of model-simulated temperature diffusivities with Argo observations (Whalen et al. 2012).

Many thanks for the reviewer's suggestions about the comparisons of the model results with Argo observations. We designed climatologic experiments in this study, so all of the model simulations including the temperature diffusivities can be regarded as the multi-year

mean results. In our opinion, it is inappropriate for our simulations to be compared with the Argo data, because there should be considerable difference between the climatologic data and the real-time in-situ observations. The WOA13 data, which is the multi-year (1955 - 2012) mean results, will be a good choice to evaluate the ocean model.

The explanation has been added in Line 285 – 289 of the revised manuscript.

I am unable to recommend this manuscript for publication in this form. However, it can be considered for publication if they address my above queries and the below comments.

1. Line 173-174: "The initial temperature and salinity are interpolated based on the annually mean Levitus data with the horizontal resolution of 1° by 1° and 33 vertical layers.." Which Levitus data authors have used? Should give the version and reference.

The Levitus94 Ocean Climatology data were used in this study. The references are shown as follows:

Levitus S., R. Burgett and T.P. Boyer. 1994. World Ocean Atlas 1994. Volume 3: Salinity. NOAA Atlas NESDIS 3. U.S. Department of Commerce, Washington, D.C. 99 pp.

Levitus S. and T.P. Boyer. 1994. World Ocean Atlas 1994 Volume 4: Temperature. NOAA Atlas NESDIS 4. U.S. Department of Commerce, Washington, D.C. 117 pp.

The version and references of the Levitus data has been added in Line 226 – 227 of the revised manuscript.

2. The author used a regional model in which the lateral boundary condition is very important for any basin-scale model, particularly for the Indian Ocean which is affected by the Indonesian Throughflow in the eastern boundary. The author did not give any details about how the boundary condition is prescribed. Is it a boundary condition with a sponge layer? The authors should provide the details about the lateral boundary conditions used in this study.

We agree with the reviewer's suggestions, the lateral boundary condition is very important for the basin-scale model. We have described the calculation of the lateral

boundary conditions in detail. The gravity-wave radiation conditions (Chapman, 1985) were used as the lateral boundary conditions. The simulated variables, including velocities, temperature, salinity and SSH, on the lateral boundary grids are calculated in an explicit numerical form. In the explicit form, the values of the related variables obtained from the daily global climatologic model results by the MASNUM model with the horizontal resolution of 1°/2 by 1°/2 are also used. The lateral boundary conditions are time-dependent with an updating period of 1 day.

Reference:

Chapman, D. C. 1985. Numerical Treatment of Cross-Shelf Open Boundaries in a Barotropic Coastal Ocean Model. Journal of Physical Oceanography, 15(8), 1060-1075.

The description has been added in Line 227 – 232 of the revised manuscript.

3. Line 175-180: The initialization strategy and the experimental details are also not very clear. It looks like the author used a cold start and then inter-annual forcing from NCEP/NCAR (1948-2021). This means its inter-annual simulations. On the other hand, they wrote "The model is integrated from the quiescent state for 10 climatological years. The simulated temperature in the last 1 year is compared with the monthly World Ocean Atlas 2013 (WOA13) climatologic data". This implies it's only 10 years of simulations. It's confusing what experiments the authors exactly carried out. It seems 10 years of simulation may not be sufficient to reach the steady-state. The authors should give the evidence that the model reached steady-state in 10th year of simulation.

The description about the time interval of the simulation is not clear. This unclear description will make the readers confused. We have revised this part. At first, we calculated the multi-year monthly mean (climatologic) surface forcing results based on the time series of the NCEP/NCAR data (1948-2021). Then the model is driven by the monthly mean surface forcing results, which repeats in every climatologic year.

In our opinion, the time interval of 10 years should be appropriate for ocean simulation from the quiescent state to a relatively stable circulation background. The average kinetic

energy was estimated in the experiments of this study, which can be regarded as a model stability index. The results show that obviously large fluctuation of the average kinetic energy occurs in the first two years, and the average kinetic energy become stable gradually in the third year and is completely steady from the forth to the tenth year. The conclusion is similar to many previous studies (e.g. Xia et al., 2006; Qiao et al., 2010; Han, 2014; Yu et al., 2020).

The explanation has been added and revised in Line 234 – 238 and 243 – 247 of the revised manuscript.

4.  The author used MASNUM wave spectrum model simulations to get the inputs for the NBSW parameterizations scheme they incorporated. But how good the model simulations compare with observations?

The MASNUM wave spectrum model was constructed by Yuan et al., (1991). The wave model has been validated by the observations (Yu et al., 1997) and widely accepted in ocean engineering and numerical simulation (e.g. Qiao et al., 1999; Xia et al., 2006; Qiao et al., 2010; Shi et al., 2016; Yang et al., 2019; Yu et al., 2020; Sun et al., 2021). The results showed that the simulated significant wave height (SWH) and mean wave period (MWP) were consistent with the satellite observations. Actually, the configuration of the wave model is relatively simple, and the model design in this study is almost the same as that in Xia et al., (2006) and Qiao et al., (2010). Therefore, we believe that the experiment using the MASNUM wave model is able to characterize the spatial pattern and variation of the surface waves.

Reference:

Yuan, Y., Hua, F., Pan, Z., and Sun, L. 1991. LAGFD-WAM numerical wave model - I Basic physical model, Acta Oceanologica Sinica, 10, 483-488

Yu, W., Qiao F., Yuan Y., Pan Z. 1997. Numerical modeling of wind and waves for Typhoon Betty (8710), Acta Oceanologica Sinica, 16, 459– 473

Qiao, F., Chen S., Li C., Zhao W., Pan Z. 1999. The study of wind, wave, current extreme parameters and climatic characters of the South China Sea, J. Mar. Technol., 33, 61–68

Xia, C., Qiao F., Yang Y., Ma J., Yuan Y. 2006. Three-dimensional structure of the summertime circulation in the Yellow Sea from a wave-tide-circulation coupled model, Journal of Geophysical Research Oceans, 111, C11S03

Qiao, F., Yuan Y., Ezer T., et al. 2010. A three-dimensional surface wave–ocean circulation coupled model and its initial testing, Ocean Dynamics, 60(5), 1339-1355

Shi, Y., Wu K., Yang Y. 2016. Preliminary Results of Assessing the Mixing of Wave Transport Flux Residualin the Upper Ocean with ROMS, Journal of Ocean University of China, 15(2), 193-200

Yang, Y., Shi Y., Yu C., et al. 2019. Study on surface wave-induced mixing of transport flux residue under typhoon conditions, Journal of Oceanology and Limnology, 37(6), 1837-1845

Yu, C., Yang Y., Yin X., et al. 2020. Impact of Enhanced Wave-Induced Mixing on the Ocean Upper Mixed Layer during Typhoon Nepartak in a Regional Model of the Northwest Pacific Ocean, Remote Sensing, 12, 2808

Sun, M., Du J., Yang Y., Yin X. 2021. Evaluation of Assimilation in the MASNUM Wave Model Based on Jason-3 and CFOSAT, Remote Sensing, 13(19), 3833

The explanation and introduction have been added in Line 252 – 255 and 259 – 261 of the revised manuscript.

5. In Figures 2c and 3c authors represented it as the IT-generated turbulent mixing scheme based on Exp-3 but in this experiment, NBSW is also included, then how can it be an IT-generated turbulent mixing scheme?

Many thanks for this comment. The description is not very clear and will make the readers confusing. The IT-generated turbulent mixing coefficients can be calculated independently, so Figures 2c and 3c exactly show the diffusive terms relating to the ITs which are a part of the diffusive terms in Exp 3.

The explanation has been added in Line 300 of the revised manuscript.

6. In Figures 2 and 3 for the vertical profiles of the monthly mean vertical temperature diffusive terms, the author choose to show the results for 5 °S, and for the temperature comparison, they showed 30.5 °S. What is the physical basis to choose these sections? Authors should show such results for the Arabian Sea and Bay of Bengal as well.

We chose the 10.5°S Section for the diffusive term comparisons. As the response to the reviewer #1, there are two reasons for the choice of the 10.5°S Section. Firstly, the Madagascar-Mascarene regions (0° ~ 25°S in the West Indian Ocean) are considered to be a hot spot for generation of semidiurnal internal tides. Both of the surface waves and internal tides should grow fully and become large enough for the comparison of the diffusive terms. Secondly, the 10.5°S Section is typical to the different pattern among the diffusion terms obviously, which include that, the $k_h$ calculated from the classic M-Y 2.5 scheme is relatively small, the NBSW-generated turbulent mixing is the largest in the ocean surface and decays exponentially with the depth, and the IT-generated turbulent mixing is obviously larger than other schemes in the ocean interior.

We chose the 30.5°S Section for the temperature comparisons. Along the 10.5°S Section, there is non-ignorable difference between the WOA13 data and the simulation results in the East Indian Ocean, we have analyzed the reasons in the manuscript and more optimization and improvement of the experimental design will be implemented in future work to make the simulated results more accurate. Furthermore, the effect of the three schemes in the tropical area are relatively non-obvious, which is regarded as a long-standing issue (Qiao et al., 2010; Zhuang et al., 2020). Therefore, we have chosen another more typical transect (the 30.5°S Section) to show the effects of the three schemes on the temperature modeling.

As the reviewer suggested, the temperature structure along the zonal transects located in the north and middle of the Arabian Sea and the Bay of Bengal was analyzed, the results are very similar to those along the 7.5° N Section, which is located in the south of the Arabian Sea and the Bay of Bengal. Furthermore, the climatologic patterns

of the temperature and current in the Arabian Sea and Bay of Bengal are similar because of the relatively low resolution and the monthly mean surface forcing. Some local meso- and small-scale features in the nearshore areas should be non-obvious. Therefore, we think that the temperature structure along the 7.5° N Section should be enough to show the pattern in the NIO.

The explanation and statement have been added in Line 302 – 305 and 326 – 332 of the revised manuscript.

7. In Figure 4 in exp1& 4 why the model does show the cooler temperature in the thermocline depth region? In general, over the Indian Ocean, almost all forced model shows warm bias (Rahaman et al. 2020). Although the thermocline bias was reduced in Exp 2 and 3, it became reversed with similar magnitude why does it so? Why there is no difference between exp-1 and exp-4 in Figures 4 and 5? Does it mean WTFR does not impact temperature simulations? Authors should show such results for the Arabian Sea and Bay of Bengal as well.

The reason for the cooler temperature in the thermocline depth region in Exp 1 should be that the multi-year monthly mean surface forcing fields were smaller than the actual values, which leads to insufficient heat transfer from the atmosphere to the ocean. After 10 climatologic years modeling, the temperature in the ocean interior became cooler obviously than the WOA13 data.

Additionally, the NBSWs and ITs enhanced the vertical mixing as well as the heat transfer, so more heat entered into the ocean interior and the SST became cooler. Additionally, the Haney equation modified the surface forcing and the solar radiation will increase continuously in the ocean surface. Therefore, the accumulation of the heat during the 10-year modeling will make the temperature bias in Exp 2 and Exp 3 reversed with similar magnitude in Exp 1.

As we discussed in the manuscript, the simulation is slightly improved in Exp 4 compared with Exp 1 because the WTFR-induced mixing is remarkably weaker than the

NBSW-generated turbulent mixing (the values of the WTFR-induced diffusion terms are about 4 to 6 orders smaller than those for the NBSW). Previous studies (Yang et al., 2019; Yu et al., 2021) demonstrated that the WTFR may play an important role in the SST cooling, if the wind and surface waves are strong. Therefore, the effects of the WTFR under the typhoon conditions will be further examined in the future work.

As we explained above, we think that the temperature structure along the 7.5° N Section, which is located in the south of the Arabian Sea and the Bay of Bengal, should be adequate to show the pattern in the NIO.

The explanation and discussion have been added and revised in Line 362 – 373 and 330 – 332 the revised manuscript.

8. Figure 8. What is the physical basis of choosing the different zone? Looks like the present defined zones will not give true representation, for example in zone 1 since the dynamics and thermodynamics are different in the Arabian Sea, Bay of Bengal and South China Sea, hence the mixing characteristics are also different. I suggest excluding the regions outside of the Indian Ocean such as South China Sea and Atlantic Ocean as included in the present zone 2 and zone 3. I also suggest the author should select the zones based on past studies or based on the dynamics and thermodynamic properties of the Indian Ocean basin.

We have replotted Figures 2 ~ 7 that the regions outside of the IO were removed. Actually, the RMSEs of the temperature shown in Figures 8 were calculated based on the simulated temperature only in the whole IO. The zone partition of the IO in this study were designed based on the previous studies and the dynamic patterns of the IO. On one hand, previous studies (e.g. Talley et al., 2011; E. Kumar et al., 2013; P. Kumar et al., 2019) showed different zone partitioning criteria which often included the NIO, the SIO and the tropical regions. On the other hand, the principal upper ocean flow regimes of the IO are the subtropical gyre of the SIO and the monsoonally forced circulation of the tropics and NIO. The whole effects of the Indonesian throughflow (ITF) should be also considered. Taking the above factors into account, the whole IO was divided into three parts, Zone 1

represented the NIO including the Arabian Sea and the Bay of Bengal, Zone 2 represented the tropics and the subtropical regions in the SIO containing the whole effects of the subtropical gyre and the ITF, and Zone 3 represented the region on the south of Zone 2 in the SIO.

The description has been added and revised in Line 382 – 389 of the revised manuscript.

9. How the RMSE is statistically robust when the authors used the seasonal cycle and computed the RMSE?

The monthly mean temperature in the upper-100 m layers between the WOA13 data and the model results were used to calculate the RMSEs as the following expression

$$\text{RMSE} = \sqrt{\frac{\sum_{i=1}^{im}\sum_{j=1}^{jm}\sum_{k=1}^{ks}\left(t_{m\ i,j,k} - t_{w\ i,j,k}\right)^2}{im \times jm \times ks}}$$

where $t_m$ and $t_w$ represent the model results and the WOA13 data of the monthly mean temperature, im, jm and ks means the number of grids in the whole IO in horizontal and vertical directions. The RMSE can be regarded as a spatially average deviation of the three-dimensional temperature fields. Therefore, in our opinion, the RMSE is statistically robust because the calculation result is unique if the spatial range of the monthly mean results is determined.

The discussion and introduction have been added in Line 375 – 381 of the revised manuscript.

10. As already pointed out in the case of the thermocline in the MLD bias given in Figure 9 for Exp-1 too looks not consistent with the previous works. In general OGCMs simulates deeper MLD in the Indian Ocean (de Boyer Montégut et al. 2007). A very recent study by Pottapinjara et al. (2022) too shows similar results. Hence, how the MLD simulation, in this case, shows shallower than observations? The authors need to explain why the model simulated MLD is shallower as compared to observations. Also, the criteria used to compute MLD is not very widely used. The authors did not provide

any reference to compute MLD or any explanation why they choose the 1 °C criterion to compute MLD.

Many thanks for the comments about the MLD simulation. The simulated MLDs were generally shallower than the observations globally because of underestimation of the vertical mixing in the upper ocean, especially during summer. The conclusion is similar to previous studies, such as Kantha and Clayson 1994; Mellor, (2001), Qiao et al., (2010), Shu et al., (2011), Huang et al., (2012), Wang et al., (2019). The accumulation of the weak vertical mixing during the 10-year climatologic modeling will make more heat staying in the surface layer, which will lead to warmer SST and shallower MLDs. In fact, from Figures 10 one can see that, the obviously shallower MLDs are generally in the Antarctic Circumpolar Current (ACC) regions, where the simulated vertical mixing from the original experiment is weak dramatically.

Another reason for the shallower MLD simulation, which is not consistent with previous studies including de Boyer Montégut et al., (2007) and Pottapinjara et al., (2022), should be different methods used to define the MLD. The threshold criterion, which is used in this study, is the most widely favored and simplest method for finding the MLD (Kara et al., 2000; de Boyer Montegut et al., 2004). In threshold method, the MLD is defined as the depth at which the temperature or density profiles change by a predefined amount relative to a surface reference value. Various threshold criteria were used to determine the MLD globally, such as 0.2 °C in de Boyer Montégut et al., (2004), 0.5 °C in Monterey and Levitus, (1997), 0.8 °C in Kara et al., (2000) and 1.0 °C in Qiao et al., (2010). Therefore, we chose one of the threshold criteria to define the MLD and attempt to make the effects of the NBSWs and ITs on the simulated MLD more obvious. Adopting the threshold criterion of 1.0 °C, the simulated MLDs were shallower than the WOA13 data because of the warmer SST and cooler temperature in the ocean interior.

Reference:

Kantha, H. L., and A. C. Clayson. 2004. On the effect of surface gravity waves on mixing in the oceanic mixed layer, Ocean Modelling, 6(2), 101-124

Mellor, G. L. 2001. One dimensional, ocean surface layer modeling: a problem and a solution. J Phys Oceanogr 31:790–809

Qiao, F., Y. Yuan, T. Ezer, C. Xia, Y. Yang, X. Lü, and Z. Song. 2010. A three-dimensional surface wave–ocean circulation coupled model and its initial testing, Ocean Dynamics, 60(5), 1339-1355

Shu, Q., F. Qiao, Z. Song, C. Xia, and Y. Yang. 2011. Improvement of MOM4 by including surface wave-induced vertical mixing, Ocean Modelling, 40(1), 42-51

Huang, C. J., F. Qiao, Q. Shu, and Z. Song. 2012. Evaluating austral summer mixed-layer response to surface wave–induced mixing in the Southern Ocean, Journal of Geophysical Research Oceans, 117, C00J18

Wang, S., Q. Wang, Q. Shu, P. Scholz, G. Lohmann, and F. Qiao. 2019. Improving the Upper-ocean Temperature in an Ocean Climate Model (FESOM 1.4): Shortwave Penetration vs. Mixing Induced by Non-breaking Surface Waves, Journal of Advances in Modeling Earth Systems, 11, 1-13

de Boyer Montégut, C., J. Vialard, S. S. C. Shenoi, D. Shankar, F. Durand, C. Ethé, and G. Madec. 2007. Simulated Seasonal and Interannual Variability of the Mixed Layer Heat Budget in the Northern Indian Ocean, Journal of Climate, 20(13), 3249-3268

Monterey, G., S. Levitus. 1997. Seasonal Variability of Mixed Layer Depth for the World Ocean. NOAA Atlas NESDIS 14, 96 pp

Kara, A. B., A. J. Wallcraft, and H. E. Hurlburt. 2003. Climatological SST and MLD predictions from a global layered ocean model with an embedded mixed layer, Journal of Atmospheric and Oceanic Technology, 20(11), 1616-1632

The introduction about finding the MLD has been added and revised in Line 469 – 475 of the revised manuscript. The explanation and discussion have been added in Line 479 – 486 of the revised manuscript.

**Response to Comments from Chief Editor**

After checking your manuscript, it has come to our attention that it does not comply with our Code and Data Policy to many levels.

https://www.geoscientific-model-development.net/policies/code_and_data_policy.html

Therefore, we have to request you several amendments (listed below) and full compliance with our policy. Failure to fix these issues will result in the rejection of your work for publication in our journal.

We apologized that our original manuscript did not comply with the Code and Data Policy to many levels. We have read the Code and Data Policy carefully and published our code and data in compliance with the policy. We have also revised the Code and Data availability section of the manuscript. The MASNUM ocean circulation model V2.0 is available on Zenodo: https://doi.org/10.5281/zenodo.6717314. The MASNUM wave spectrum model V3.0 is available on Zenodo: https://doi.org/10.5281/zenodo.6719479. All of the data used in this article can be downloaded on Zenodo: https://doi.org/10.5281/zenodo.6749788.

The 'Code and data availability' has been revised in Line 625 – 635 of the revised manuscript.

First, you have archived your code and part of your data on GitHub. However, GitHub is not a suitable repository (we make it clear in our policy). GitHub itself instructs authors to use other alternatives for long-term archival and publishing, such as Zenodo. Therefore, please, publish your code and data in one of the appropriate repositories.

As the Chief Editor suggests, we have published our code and data on the Zenodo for long-term archival.

The repositories in 'Code and data availability' have been updated.

Secondly, for data access, you cite repositories in third-party servers that are not suitable for publication purposes (NOAA, Columbia, etc.) You must upload to one of the suitable repositories the data that you have used in your work instead of citing generic web pages that are continuously updated. Then, you must include in the Code and Data availability section of your paper the link and DOI of the new repositories.

As the Chief Editor suggests, we have uploaded all of the data required in the modeling and evaluation of the results in a new repository of the Zenodo. The related DOI has been added in the Code and Data availability section.

Third, both the MASNUM wave model and the oceanic model are provided without a license. If you do not include a license, the code continues to be your property and can not be used by others, despite any statement on being free to use. Therefore, when uploading the model's code to the repository, you could want to choose a free software/open-source (FLOSS) license. In our editorial guidelines, we recommend the GPLv3. You only need to include the file 'https://www.gnu.org/licenses/gpl-3.0.txt' as LICENSE.txt with your code. Also, you can choose other options that Zenodo provides: GPLv2, Apache License, MIT License, etc.

As the Chief Editor suggests, the GPLv3 license file has been uploaded on the Zenodo with our codes which was named as "license.txt".

Finally, you have provided data in the Github repository in two proprietary compression formats (.rar and .mat), which specification is not open. In this way, future compatibility and access to the data of your work are not assured. Please, save and publish the data using open formats such as ZIP, NetCDF or HDF to solve this issue.

According to the Chief Editor's comments, all of the data have been saved in the NetCDF format and named as "**.nc". All of the data and model codes used in this article have been compressed and stored in the ZIP format and uploaded on the Zenodo.

Please, reply as soon as possible to this comment. All these issues should have been addressed before beginning the Discussion period; however, it is now close to its end.

Juan A. Añel

Geosci. Model Dev. Executive Editor

**Citation**: https://doi.org/10.5194/gmd-2022-110-CEC1

We thanks again for the Chief Editor's careful reviews and constructive comments. According to the comments and suggestions, we have tried our best to publish our code and data in compliance with the Code and Data Policy, and revise the 'Code and Data availability' section of the manuscript.

Other revisions:

We tried our best to improve the writing English. The manuscript has been carefully checked and revised as follows.

Line 1: 'M2' has been replaced by '$M_2$';

Line 27 – 28: 'the thermal structure and the mixed layer depth' has been replaced by 'the thermal structure, the mixed layer depth and the surface currents';

Line 45: '(NBSWs)' has been added;

Line 71: '(ITs)' has been added;

Line 130: 'non-breaking surface waves' has been replaced by 'NBSWs';

Line 132: 'non-breaking wave-induced' has been replaced by 'NBSW-induced';

Line 137: 'significant wave height' has been replaced by 'SWH'; 'the' has been deleted;

Line 156: '(IT)' has been deleted;

Line 200: 'on the right side' has been added;

Line 300: 'Figs. 2 and 3' has been replaced by 'Figures 2 and 3';

Line 317: 'Deep yellow areas correspond to the lands' has been added;

Line 322: 'fileds' has been replaced by 'fields';

Line 335: 'transfered' has been replaced by 'transferred';

Line 343: 'White' has been replaced by 'Deep yellow';

Line 477: 'Figure 9' has been replaced by 'Figures 10';

Line 494: 'Figure 9' has been replaced by Figure 10;

Line 495 – 496: 'Dark' has been replaced by 'Deep yellow';

Line 564: 'Figure 10' has been replaced by 'Figure 12';

Line 565 – 566: 'SIT and TFR mixing represent the shear induced turbulent and transport flux residue mixing, respectively. WE means the water exchange.' has been added;

Line 582 – 583: 'and more optimization and improvement of the real-time hindcast experimental design' has been added;

Line 587: 'aresummarized' has been replaced by 'are summarized';

Line 593: 'and surface currents' has been added;

Line 600: 'internal tide' has been replaced by 'IT'; 'nolinear' has been replaced by 'nonlinear';

Line 636: 'coauthors' has been replaced by 'co-authors'.

---

## Referee Report (RR1)

As the former Reviewer #1, I believe that the authors have responded to my suggestions and comments, and those of the other reviewers, satisfactorily. However, there are some further minor points:

* Former minor point 1: The explanation should be placed at the end of Line 70, "In this study, we analyze the effects of the turbulent mixing generated by the M2 internal tides on the ocean circulation."
* Former minor point 3: Please provide references for your first and second arguments.
* Fig. 1: Please add the unit and also check the rest of the figures for similar mistakes.

Thank you to the authors for responding to all of the reviewers so thoroughly, and congratulations on your manuscript. I am happy to recommend publication in the Geoscientific Model Development after the small corrections noted above.

---

## Referee Report (RR2)

**"Improved ocean circulation modeling with combined effects of surface waves and M2 internal tides on vertical mixing: a case study for the Indian Ocean" By Zhuang et al.**

In this study, the authors incorporated three mixing schemes into the ocean general circulation model, namely non-breaking surface-wave-generated turbulent mixing(NBSW), the mixing induced by the wave transport flux residue(WTFR), and the internal tide-generated turbulent mixing(IT) along with Mellor-Yamada 2.5 mixing scheme. This study of quantifying the role of wave and tide-induced mixing in an ocean model is a timely and valuable contribution. However, the authors are unable to represent it in terms of value addition to its scientific contributions. There are many gaps in this study starting with ocean model configurations and their different experiments.

Author partially responded to my above comment. In terms of scientific contribution author respnsed "Furthermore, addition to the scientific value, the results in this study are helpful to improve the accuracy and timeliness of the global ocean numerical prediction for the national or regional forecasting agencies, because the MASNUM ocean model is able to depict more complete physical processes. In our opinion, it is important to study the NBSW- and IT-induced mixing for promoting the development of the ocean and coupling models."

How does the accuracy and timeliness of the global ocean numerical prediction for the national or regional forecasting agencies are helpful ??  the MASNUM ocean model is able to depict more complete physical processes HOW ???

The introduction lacks the present status of the state of the art model's mixing schemes with details and its drawbacks in the Indian Ocean. The authors are unable to give the scientific objectives to be achieved in this study as compared to the previous works. The representation of the internal tide-generated turbulent mixing is not new, in fact, it's been introduced by Simmons et al. (2004) in a global Ocean General Circulation model. The author did not mention this work and its related works (Nagai and Hibiya (2015).

Again same fro my above comments. Author did not give any insight about what are outstanding issues in terms of mixing schems in the ocean model with respect to Indian Ocean.For example overflow schems in the Indian Ocean models to represnt Red Sea salty water yet an outstanding issues. I agree that  the internal tide-generated turbulent mixing is not explecitely implimented but yet it shows significant improvement, however, in the presnt study as such no significant improvement can be found.

Also, the authors presented the results only up to 130 m which does not represent insight into the mixing process related to internal tides since its effect could be seen in the deeper layers.  A very

recent study by Lozovatsky et al. (2022) showed that the observed eddy diffusivity in the ocean pycnocline over the southeastern BoB is likely related to internal-wave generated turbulence.

Authors gave explanation that below 130 m the model simulations  shows too warm when compared with WOA13.  How much warm ? what is the exact value ? 3,4 5 C ????
 The explanation given for this warm simulation as " The reason is that the Haney equation (Haney, 1971) was used to modify the  climatologic surface heat flux and improve the large-scale thermal coupling of ocean and atmosphere, but a disadvantage of the Haney modifying method is the destruction of the heat balance, so excessive heat may be transmitted into the ocean interior."
This explanation of temperature restoration is the only reason as given by the authors may not be correct. It seems MASNUM model has fundamental  problem to reproduce it even in the control simulation. Author should give concreat scientific evidence what the exact cause of warm temperature simulation. The other existing Indian ocean regiionl model simulation based on MOM, ROMS ,HYCOM etc does not show too warm temperature below 130 m as the authors mentioned.

I understand  its climatological simulation but the eddy diffusivity in the ocean pycnocline as a model diagonostics can be obtained from all these sensetivity experiments. This will give some preliminary idea how  much its differs with respect to instantanious values given in Lozovatsky et al. (2022)

In line-121-22 the authors wrote "….., the mode-1 M2 internal tides, which mainly originate from regions with steep topographic gradients, are considered…." . Doesn't it imply that the mixing will be more over the steep topographic gradients?. But the author did not show any results related to this.

Reply: Explnation with respect to my above comment looks fine.

 The authors implemented the mixing schemes in the momentum equations.   This implementation will also affect the dynamics as well. But the authors did not show any results on whether any changes are there in the circulations. The authors should show a few results about how the upper ocean currents improved with implementations of NBSW, IT, and WTFR mixing schemes.

With reference to my above comment ,although  the author gave comparion plot with OSCAR but its very hard to see any changes  between Exp-1, Exp-2 and Exp-3. It looks no significant change of circulation is induced with the inclusion of the stated schems  ecplicitely as a subroutine in the momentum eqation.

 It will be good if the authors also can show spatial comparisons of model-simulated temperature diffusivities with Argo observations (Whalen et al. 2012).  I am unable to recommend this manuscript for publication in this form. However, it can be considered for publication if they address my above queries and the below comments.

The argumet given for not to give the comparion results not acceptable. I agree this will not give exact values but at least will give the spatial distribution pattern. Author must show the spatial comparion.

1. Line 173-174: "The initial temperature and salinity are interpolated based on the annually mean Levitus data with the horizontal resolution of 1° by 1° and 33 vertical layers.." Which Levitus data authors have used? Should give the version and reference.
Reply: Levitus94 data for Indian Ocean model initilization may not give realisit climatological spatial pattern. Its too old and I guess hardly any representative data went onto this. I think author must use the recent WOA atlas may be 2013 or 2018 for the initilization.

2. The author used a regional model in which the lateral boundary condition is very important for any basin-scale model, particularly for the Indian Ocean which is affected by the Indonesian Throughflow in the eastern boundary. The author did not give any details about how the boundary condition is prescribed. Is it a boundary condition with a sponge layer? The authors should provide the details about the lateral boundary conditions used in this study.
Reply: Explnation with respect to my above comment looks fine.

3. Line 175-180: The initialization strategy and the experimental details are also not very clear. It looks like the author used a cold start and then inter-annual forcing from NCEP/NCAR (1948-2021). This means its inter-annual simulations. On the other hand, they wrote "The model is integrated from the quiescent state for 10 climatological years. The simulated temperature in the last 1 year is compared with the monthly World Ocean Atlas 2013 (WOA13) climatologic data" . This implies it's only 10 years of simulations. It's confusing what experiments the authors exactly carried out. It seems 10 years of simulation may not be sufficient to reach the steady-state. The authors should give the evidence that the model reached steady-state in $10^{th}$ year of simulation.
Reply: Explnation with respect to my above comment looks fine.

4. The author used MASNUM wave spectrum model simulations to get the inputs for the NBSW parameterizations scheme they incorporated. But how good the model simulations compare with observations?
Reply: Explnation with respect to my above comment looks fine.

5. In Figures 2c and 3c authors represented it as the IT-generated turbulent mixing scheme based on Exp-3 but in this experiment, NBSW is also included, then how can it be an IT-generated turbulent mixing scheme?
Reply: Explnation with respect to my above comment looks fine.

6. In Figures 2 and 3 for the vertical profiles of the monthly mean vertical temperature diffusive terms, the author choose to show the results for 10.5 °S, and for the temperature comparison, they showed 30.5 °S. What is the physical basis to choose these sections? Authors should show such results for the Arabian Sea and Bay of Bengal as well.

Reply: Partially responded. If the authors compared it for Arabian Sea and Bay of Bengal, then why they did not gave the figures ?

7. In Figure 4 in exp1& 4 why the model does show the cooler temperature in the thermocline depth region? In general, over the Indian Ocean, almost all forced model shows warm bias (Rahaman et al. 2020). Although the thermocline bias was reduced in Exp 2 and 3, it became reversed with similar magnitude why does it so? Why there is no difference between exp-1 and exp-4 in Figures 4 and 5? Does it mean WTFR does not impact temperature simulations? Authors should show such results for the Arabian Sea and Bay of Bengal as well.

Reply: The reson given for cooler therocline temp in Exp-1 as " The reason for the cooler temperature in the thermocline depth region in Exp 1 should be that the multi-year monthly mean surface forcing fields were smaller than the actual values, which leads to insufficient heat transfer from the atmosphere to the ocean. After 10 climatologic years modeling, the temperature in the ocean interior became cooler obviously than the WOA13 data." This explanation does not show any scientific argumnet. Not fully convinced. How does multi-year monthly mean surface forcing fields were smaller than the actual values will affec the simulation ? The author made the forcing climatology I guess from 1948-2021.
Similarly the explnation why the bias reversed in Exp-2 with respect to Exp-1 is also not complete. Author should give scientiifc evidence may be the complete heat budget for the explnation.

8. Figure 8 What is the physical basis of choosing the different zone? Looks like the present defined zones will not give true representation, for example in zone 1 since the dynamics and thermodynamics are different in the Arabian Sea, Bay of Bengal and South China Sea, hence the mixing characteristics are also different. I suggest excluding the regions outside of the Indian Ocean such as South China Sea and Atlantic Ocean as included in the present zone 2 and zone 3. I also suggest the author should select the zones based on past studies or based on the dynamics and thermodynamic properties of the Indian Ocean basin.

Reply: In that case there should be a zone in the equatorial region. It will not be good to include ITF in zone 2, zone 2 must be devided in to two zone.

9. How the RMSE is statistically robust when the authors used the seasonal cycle and computed the RMSE?

10. As already pointed out in the case of the thermocline in the MLD bias given in Figure 9 for Exp-1 too looks not consistent with the previous works. In general OGCMs simulates deeper MLD in the Indian Ocean (de Boyer Montégut et al. 2007). A very recent study by

Pottapinjara et al. (2022) too shows similar results. Hence, how the MLD simulation, in this case, shows shallower than observations? The authors need to explain why the model simulated MLD is shallower as compared to observations. Also, the criteria used to compute MLD is not very widely used. The authors did not provide any reference to compute MLD or any explanation why they choose the 1 °C criterion to compute MLD.

Reply:"........ In fact, from Figures 10 one can see that, the obviously shallower MLDs are generally in the Antarctic Circumpolar Current (ACC) regions, where the simulated vertical mixing from the original experiment is weak dramatically." What about the Indian Ocean partcularly in the Arabian Sea and Bay of Bengal.

---

## Author Response (AR2)

A point-by-point listing of response for each of the reviewers' comments

We thank the reviewers for careful reviews and constructive comments in improving the manuscript. Below is our point-to-point reply to these comments (the reviewers' comments in the last review are in black, the reviewers' new comments are in blue, our responses are in red, and our changes in manuscript are in red and italics).

**All of the changes in the revised manuscript are marked using 'track changes' in Word. Please refer to the marked-up manuscript version showing the changes made in detail.**

**Response to Comments from Referee #1**

As the former Reviewer #1, I believe that the authors have responded to my suggestions and comments, and those of the other reviewers, satisfactorily. However, there are some further minor points:

\* Former minor point 1: The explanation should be placed at the end of Line 70, "In this study, we analyze the effects of the turbulent mixing generated by the M2 internal tides on the ocean circulation."

We agree with the reviewer's suggestion. The explanation has been placed at the end of the sentence "In this study, we analyze the effects of the turbulent mixing generated by the M2 internal tides on the ocean circulation." in the original manuscript.

*Please refer to Line 74-81 and 110-117 in the revised manuscript.*

\* Former minor point 3: Please provide references for your first and second arguments.

Many thanks for the suggestion. The related references have been added.

*The references about the reasons for the choice of the 10.5°S Section have been added in Line 314 and 316-317 of the revised manuscript.*

\* Fig. 1: Please add the unit and also check the rest of the figures for similar mistakes.

Many thanks for the reviewer's suggestion. We agree with the reviewer and accept the suggestion. The figures have been replotted with adding the units.

*Figures 2-8, 10-13 in the revised manuscript have been replotted.*

Thank you to the authors for responding to all of the reviewers so thoroughly, and congratulations on your manuscript. I am happy to recommend publication in the Geoscientific Model Development after the small corrections noted above.

Many thanks for the reviewer's encouragement and comments. We tried our best to revise the manuscript following the reviewers' suggestion and comments.

**Response to Comments from Referee #2**

**"Improved ocean circulation modeling with combined effects of surface waves and M2 internal tides on vertical mixing: a case study for the Indian Ocean" By Zhuang et al.**

In this study, the authors incorporated three mixing schemes into the ocean general circulation model, namely non-breaking surface-wave-generated turbulent mixing(NBSW), the mixing induced by the wave transport flux residue(WTFR), and the internal tide-generated turbulent mixing(IT) along with Mellor-Yamada 2.5 mixing scheme. This study of quantifying the role of wave and tide-induced mixing in an ocean model is a timely and valuable contribution. However, the authors are unable to represent it in terms of value addition to its scientific contributions. There are many gaps in this study starting with ocean model configurations and their different experiments.

Author partially responded to my above comment. In terms of scientific contribution author represented "Furthermore, addition to the scientific value, the results in this study are helpful to improve the accuracy and timeliness of the global ocean numerical prediction for the national or regional forecasting agencies, because

the MASNUM ocean model is able to depict more complete physical processes. In our opinion, it is important to study the NBSW- and IT-induced mixing for promoting the development of the ocean and coupling models." How does the accuracy and timeliness of the global ocean numerical prediction for the national or regional forecasting agencies are helpful?? The MASNUM ocean model is able to depict more complete physical processes, HOW???

The statements about the scientific contribution were not very correct. It is inappropriate to present "the results in this study are helpful to improve the accuracy and timeliness of the global ocean numerical prediction for the national or regional forecasting agencies". Actually, we would like to say that the mixing schemes introduced in this study contain the effects of the surface waves and internal tides, which are thought to be the supplement of the physical mechanism for the vertical mixing processes in the OGCMs, because the original turbulent mixing schemes, such as the M-Y 2.5 scheme, neglected the interaction between the surface waves and the currents (Huang et al., 2011; Qiao and Huang, 2012). The M-Y 2.5 mixing scheme combined with the NBSW- and IT-induced mixing schemes should become more complete for modeling the vertical mixing processes.

*The statements have been revised in Line 631-639 of the revised manuscript.*

The introduction lacks the present status of the state of the art model's mixing schemes with details and its drawbacks in the Indian Ocean. The authors are unable to give the scientific objectives to be achieved in this study as compared to the previous works. The representation of the internal tide-generated turbulent mixing is not new, in fact, it's been introduced by Simmons et al. (2004) in a global Ocean General Circulation model. The author did not mention this work and its related works (Nagai and Hibiya (2015).

Again same for my above comments. Author did not give any insight about what are outstanding issues in terms of mixing schemes in the ocean model with respect to Indian Ocean. For example overflow schemes in the Indian Ocean models to represent Red Sea salty water yet an outstanding issues. I agree that the internal tide-generated

turbulent mixing is not explicitly implemented but yet it shows significant improvement, however, in the present study as such no significant improvement can be found.

We agree with the reviewer. The direct modeling of the internal tides is an effective and important way to study the internal tide-generated turbulent mixing, but there still some issues need to be solved to improve the simulation, including more accurate wind stress and simulated temperature and current structure, the establishment of a reasonable non-hydrostatic ocean model, and the parameterization of the interaction between different tidal constituents (Nagai and Hibiya, 2015). However, it should be noted that there should be a disadvantage for this direct modeling, which is that the simulated internal tide processes will become inaccurate if the temperature and current structure cannot be modeled accurately.

We have to admit that the issues mentioned above cannot be solved entirely when the NBSW- and IT-induced mixing schemes introduced in this study are adopted. The present study just provided another way and preliminary attempt to study the mixing processes induced by the internal tides. It should be more convenient to improve the simulation further because the mixing schemes are independent with the ocean models. The research team including the authors is developing an internal-wave/tide spectrum model. And a multi-scale process coupling model, including atmosphere, ocean current, tide, surface-wave, and internal-wave/tide component models, will be established in future for the accurate and high-resolution ocean modeling. The NBSW- and IT-induced mixing schemes and the related results in this study are helpful and valuable for establishing the coupling model.

*The explanations have been revised in Line 127-134 and 664-669 of the revised manuscript.*

Also, the authors presented the results only up to 130 m which does not represent insight into the mixing process related to internal tides since its effect could be seen in the deeper layers. A very recent study by Lozovatsky et al. (2022) showed that the observed eddy diffusivity in the ocean pycnocline over the southeastern BoB is likely related to internal-wave generated turbulence.

Authors gave explanation that below 130 m the model simulations shows too warm when compared with WOA13. How much warm? What is the exact value? 3, 4, 5 °C ???? The explanation given for this warm simulation as " The reason is that the Haney equation (Haney, 1971) was used to modify the climatologic surface heat flux and improve the large-scale thermal coupling of ocean and atmosphere, but a disadvantage of the Haney modifying method is the destruction of the heat balance, so excessive heat may be transmitted into the ocean interior." This explanation of temperature restoration is the only reason as given by the authors may not be correct. It seems MASNUM model has fundamental problem to reproduce it even in the control simulation. Author should give concrete scientific evidence what the exact cause of warm temperature simulation.

Many thanks for the reviewer's comments. The comparisons between the simulated temperature and the WOA13 data are analyzed detailedly again, and shown in Figures 1 – 3 with the depths from 0 to 300 m in January. Along the 30.5°S Transect, most of the deviations appear in the upper-100 m layers, the simulated temperature in Exp 1 is cooler than the WOA13 data and improved dramatically in Exp 2 and 3 when the NBSW- and IT-induced mixing schemes are adopted. The only special region is that on the southwest of the Australia (100° - 120° E), in which the simulated temperature is warmer than the WOA13 data when the depth is deeper than 150 m. This indicates that the simulated temperature is often cooler than the WOA13 data in the ACC region and the south of the IO. The mean deviations of the temperature in the upper-300 m regions are also shown in the figures.

However, along the 0.5° S (near the equator) and the 7.5° N Transects, the simulated temperature is warmer obviously than the WOA13 data when the depth is deeper than 120 m, and even the NBSW- and IT-induced mixing schemes cannot work anymore. Therefore, the distribution pattern of the temperature in the ocean interior (from 100 m to 300 m or deeper) seems to appear as cooler in the SIO and warmer in the NIO and the tropics.

The intermediate and deep water masses in the IO are often effected by the Southern Ocean including Antarctic Intermediate Water (AAIW), Circumpolar Deep Water (CDW), North Atlantic Deep Water (NADW), etc. These cooler water in the

Southern Ocean is carried by the meridional overturning circulation into the IO throughout south of the South Equatorial Current in the subtropical Indian Ocean, but the situation does not appear in the simulated current fields (Figures 4). Therefore, in addition to the relatively coarse and smoothed surface forcing mentioned in the last response, the another important reason should be that it is hard to simulate accurately the meridional overturning circulation in the present experiments, especially the meridional transport of the heat (cooler water in the Southern Ocean) from south (Southern Ocean) to north (South Indian Ocean). This makes the simulated temperature warmer than the WOA13 data when the depth is deeper than about 120 m along the 0.5° S (near the equator) and the 7.5° N Transects. More optimization and improvement of the real-time experimental design will be implemented in future work to solve the related issues.

[Figure]

Figure 1. The vertical temperature profiles along 30.5° S in January. Left-top: The temperature structure from the monthly WOA13 data (units: ℃). Remaining 3 subfigures: the difference of the temperature calculated by subtracting the monthly mean results simulated in Exp 1 - Exp 3 from the monthly WOA13 data, respectively. The mean deviations of the temperature in the upper-300 m regions are also given as red text

[Figure]

[Figure]

Figure 2. The same as Fig. 1, but along 0.5° S

[Figure]

Figure 3. The same as Fig. 1, but along 7.5° N

[Figure]

Figure 4. Left: adjusted steric height (related geostrophic streamfunction) (10 m²/s²) with the depth of 200 m (Reid, 2003). Middle: Net northward (meridional) transport (Sv) for the Indian Ocean at 33° S, integrated from the bottom to the top (Ganachaud et al., 2000). Right: Simulated horizontal currents in Exp 1.

Following the reviewer's comments, the initialization design is also important for the ocean modeling. The comparison of the annual mean temperature between the Levitus94 data and the WOA 13 data is shown in Figures 5 and 6. One can see that the

WOA13 data contains more meso-scale information than the Levitus94 data. The comparison shows that the temperature from the Levitus94 data is cooler obviously than that from the WOA13 data in the ACC regions (45° - 75° E, 35° - 50° S), while warmer generally in the whole IO with the depth from 200 to 500 m. Therefore, the inaccurate initial field should be also one of the reasons why the simulated temperature in the ocean interior is different from the WOA13 data. A series of the high-resolution real-time numerical experiments for the circulation in the IO will be carried out to examine the influence of different initial fields, parameterization schemes, surface fluxes, and open boundary conditions in future.

[Figure]

[Figure]

Figure 5. The comparison of the horizontal distribution of the temperature from the

WOA13 and Levitus94 data.

[Figure]

Figure 6. The comparison of the vertical distribution of the temperature from the

WOA13 and Levitus data.

*The explanations and analysis have been revised in Line 389-393 and 445-475 of*

*the revised manuscript. Figures 1-3 have not been plotted in the revised manuscript*

*because the detailed analysis for the deep ocean was not performed further here and*

*the vertical mixing in the upper ocean (0 ~ 100 m) is the main focus of this study.*

*Figures 5 and 6 have also not been plotted in the revised manuscript because of the relatively minor role in the research on the vertical mixing and the limitation of the length of the article.*

Reference:

Reid, J.L., 2003. On the total geostrophic circulation of the Indian Ocean: Flow patterns, tracers and transports. Progr. Oceanogr. 56, 137-186.

Ganachaud, A., Wunsch, C., Marotzke, J., Toole, J., 2000. Meridional overturning and large-scale circulation of the Indian Ocean. J. Geophys. Res. 105, 26117e26134.

The other existing Indian ocean regional model simulation based on MOM, ROMS, HYCOM etc does not show too warm temperature below 130 m as the authors mentioned.

In our opinion, the MASNUM ocean circulation model is suitable for the ocean modeling in the IO. Han (2014) and Han and Yuan (2014) have tested the modeling ability of the MASNUM model compared with the POM, the results showed that the MASNUM model could produce quite identical simulation results as the existing models with only half computer cost. The effects of the NBSW and IT on the vertical mixing processes are the main part of this study, but regrettably, the simulations especially in the control experiment seems to be unable to obtain satisfactory results because of the relatively coarse model design. We believe that high-resolution real-time numerical experiments based on the MASNUM ocean model developed in future will obtain more accurate simulation of the temperature and currents in the IO.

*The explanations have been added in Line 228-230 of the revised manuscript.*

I understand its climatological simulation but the eddy diffusivity in the ocean pycnocline as a model diagnostics can be obtained from all these sensitivity experiments. This will give some preliminary idea how much its differs with respect to instantaneous values given in Lozovatsky et al. (2022).

Following the reviewer's suggestion, the eddy diffusivity ( $K_N = {}^{\gamma\varepsilon}\!/_{N^2}$ ) can be characterized by the vertical mixing coefficients (km, Bus and Bui are viscosity corresponding to the momentum equations, kh, Bts = 2Bus and Bti = 2Bui are diffusivity to the tracer equations). The km and kh are calculated by M-Y 2.5 scheme, Bus and Bts are calculated by NBSW-induced mixing scheme, and Bui and Bti are calculated by IT-induced mixing scheme. The comparisons of the eddy diffusivity are shown in Figures 7 and 8. The eddy diffusivity of the WTFR is omitted here because the expression of the diffusive term is not in a standard form, which means that we can not obtain the eddy diffusivity directly from the expression of the diffusive term.

From Figures 7 and 8, one can see that the vertical distribution are very similar to that of the diffusive terms (Figures 2 and 3 in the manuscript). Especially in January, Bts is the largest in the upper-30 m layers and Bti is larger generally in the ocean interior with the depth deeper than about 40 m. Kh and Bts decay with the depth below the sea surface, the delay rate of Bts is slower obviously than Kh, so Bts is larger than Kh in the ocean interior. The high-value layers ($>10^{-5}$ m$^2$/s) of the Kh are as thin as about 20 m in January, and up to about 80 m partially in July, while the high-value layers of the Bts are generally about 70-100 m both in January and July. When the depth is larger than 40 m, the value of Bti appear to be about $10^{-5}$-$10^{-3}$ m$^2$/s.

[Figure]

Figure 7. Vertical profiles of the diffusivity in logarithmic scale along 10.5° S in
January, including Kh (a), Bts (b), and Bti (c)

[Figure]

Figure 8. The same as Fig. 7, but in July

*The explanations and the description of the values of the eddy diffusivity have been added in Line 325-331 of the revised manuscript. Figures 7 and 8 have not been plotted in the revised manuscript because they are very similar to Figures 2 and 3 in the manuscript.*

In line-121-22 the authors wrote "….., the mode-1 M2 internal tides, which mainly originate from regions with steep topographic gradients, are considered…." . Doesn't it imply that the mixing will be more over the steep topographic gradients?. But the author did not show any results related to this.

Reply: Explanation with respect to my above comment looks fine.

Many thanks for the reviewer's suggestion. We tried our best to revise the manuscript.

The authors implemented the mixing schemes in the momentum equations. This implementation will also affect the dynamics as well. But the authors did not show any results on whether any changes are there in the circulations. The authors should show a few results about how the upper ocean currents improved with implementations of NBSW, IT, and WTFR mixing schemes.

With reference to my above comment, although the author gave comparison plot with OSCAR but it's very hard to see any changes between Exp-1, Exp-2 and Exp-3. It looks no significant change of circulation is induced with the inclusion of the stated schemes explicitly as a subroutine in the momentum equation.

On one hand, only the simulated surface current fields were compared with the OSCAR data, the RMSEs were reduced slightly because the surface currents are controlled dominantly by the surface wind stresses, the interaction between the ocean and atmosphere, etc., rather than the vertical mixing in the upper layers. Even so, the NBSW- and IT-induced mixing schemes still can partially improve the simulation of the surface currents (RMSE decreased in Exp 3 compared with Exp 1). On the other hand, regrettably, the simulated currents in the ocean interior were not improved when the NBSW- and IT-induced mixing schemes were adopted. This implies that the

currents are complicated and vertical mixing is just one of the influencing factors. Unlike the temperature, the NBSW- and IT-induced mixing schemes are unable to improve the simulated currents obviously at present (may be applicable to the thermohaline circulation, which is the oceanic deep circulation system with global scale in a relatively steady state), more accurate mixing schemes or other ways for parameterizing the physical mechanism will be developed in future.

It will be good if the authors also can show spatial comparisons of model-simulated temperature diffusivities with Argo observations (Whalen et al. 2012). I am unable to recommend this manuscript for publication in this form. However, it can be considered for publication if they address my above queries and the below comments.

The argument given for not to give the comparison results not acceptable. I agree this will not give exact values but at least will give the spatial distribution pattern. Author must show the spatial comparison.

Many thanks for the reviewer's suggestion, the existing Argo-derived gridded products, which are named Barnes objective analysis (BOA)-Argo datasets (Li et al., 2017), are chosen to validate the simulated temperature structure. The climatologic monthly mean BOA-Argo data (multi-year mean from 2004 to 2014) are used and can be downloaded directly from ftp://data.argo.org.cn/pub/ARGO/BOA_Argo/. The BOA-Argo data with 49 vertical levels from the surface to 1950 m depth is produced based on refined Barnes successive corrections by adopting flexible response functions. A series of error analyses are adopted to minimize errors induced by non-uniform spatial distribution of Argo observations. These response functions allow BOA-Argo to capture a greater portion of mesoscale and large-scale signals while compressing small-sale and high-frequency noise the performance of the BOA-Argo dataset demonstrates both an accuracy and retainment of mesoscale features. Generally, BOA-Argo seems compare well with other global gridded data sets (Li et al., 2017).

Figures 9 – 12 shows the comparison of the temperature structure between the monthly BOA-Argo data and the model results. The patterns are similar to those from the WOA13 data (please see Figures 4a, 5a, 6a and 7a in the manuscript, and Figures

9a, 10a, 11a and 12a). The difference between the BOA-Argo data and the model results along 30.5° S is also similar to the WOA13 data. Compared with Exp 1, the difference for Exp 2 often decreases remarkably, the difference for Exp 3 is much smaller than that of Exp 1 and Exp 2 because of the incorporation of the IT-generated turbulent mixing, especially in the layers with depths between 20 m and 50 m. In addition, the improvement of the NBSW and IT along 7.5° N is not obvious (RMSEs decrease a little), this conclusion is also similar to that for the WOA13 data. This implies that the three mixing schemes introduced in this study may not be appropriate in the marginal sea simulation that if full of small- and meso-scale processes. In order to solve the issues about the accuracy, we attempt to design the high-resolution real-time numerical modeling experiments in the North Indian Ocean (or Arabian Sea and the Bay of Bengal only), as well as the finer simulation of the surface waves and more accurate estimation of the ITs.

[Figure]

Figure 9. The vertical temperature profiles along 30.5° S in January. (a) The temperature structure from the monthly BOA-Argo data (units: °C). (b) - (f) The difference of the temperature calculated by subtracting the monthly mean results simulated in Exp 1 - Exp 5 from the monthly BOA-Argo data, respectively. The

[Figure]

Figure 10. The same as Fig. 9, but in July

[Figure]

Figure 11. The same as Fig. 9, but along 7.5° N

[Figure]

Figure 12. The same as Fig. 9, but along 7.5° N and in July

*The analysis has been added in Line 507-532 of the revised manuscript. Figures 9 and 11 have been added in the revised manuscript.*

Reference:

Li H, Xu F, Zhou Wet al. 2017. Development of a global gridded Argo data set with Barnes successive corrections. J Geophys Res Oceans 122(2):866–889

1. Line 173-174: "The initial temperature and salinity are interpolated based on the annually mean Levitus data with the horizontal resolution of 1° by 1° and 33 vertical layers.." Which Levitus data authors have used? Should give the version and reference.

Reply: Levitus94 data for Indian Ocean model initialization may not give realist climatological spatial pattern. It's too old and I guess hardly any representative data went onto this. I think author must use the recent WOA atlas may be 2013 or 2018 for the initialization.

We agree with the reviewer's comments. As we explained above (please see the

analysis of Figures 5 and 6 in the response), the WOA13 data contains more meso-scale information than the Levitus94 data. The temperature from the Levitus94 data is cooler than the WOA13 data in the ACC region, while generally warmer in the IO with the depth from 200 to 500 m. The difference of the initial fields between the Levitus94 and WOA13 data may be one of the factors to make the simulated temperature in the IO warmer than the WOA13 data (Figures 2 and 3).

It is worth noting that, we attempt to carry out the experiments using WOA13 as the initial fields, this means that the works about the simulation and analysis should be done all over again. And we also find other factors that can improve the modeling. Therefore, now we (together with other colleagues) are carrying out a series of the **high-resolution real-time** numerical experiments for the circulation in the IO, to examine the influence of different parameterization (including vertical and horizontal mixing schemes), surface fluxes (different heat and momentum fluxes), open boundary conditions (quasi-global modeling results or HYCOM reanalysis products), initial fields (newly added following the reviewer's suggestion, the WOA18, BOA-Argo or HYCOM reanalysis products will be used). The model design was implemented referencing some previous studies such as Nagai and Hibiya, (2015).

*Please refer to Line 389-392 and 458-470 of the revised manuscript.*

2. The author used a regional model in which the lateral boundary condition is very important for any basin-scale model, particularly for the Indian Ocean which is affected by the Indonesian Throughflow in the eastern boundary. The author did not give any details about how the boundary condition is prescribed. Is it a boundary condition with a sponge layer? The authors should provide the details about the lateral boundary conditions used in this study.

Reply: Explanation with respect to my above comment looks fine.

Many thanks for the reviewer's suggestion. We tried our best to revise the manuscript.

3. Line 175-180: The initialization strategy and the experimental details are also

not very clear. It looks like the author used a cold start and then inter-annual forcing from NCEP/NCAR (1948-2021). This means its inter-annual simulations. On the other hand, they wrote "The model is integrated from the quiescent state for 10 climatological years. The simulated temperature in the last 1 year is compared with the monthly World Ocean Atlas 2013 (WOA13) climatologic data" . This implies it's only 10 years of simulations. It's confusing what experiments the authors exactly carried out. It seems 10 years of simulation may not be sufficient to reach the steady-state. The authors should give the evidence that the model reached steady-state in 10th year of simulation.

Reply: Explanation with respect to my above comment looks fine.

Many thanks for the reviewer's suggestion. We tried our best to revise the manuscript.

4. The author used MASNUM wave spectrum model simulations to get the inputs for the NBSW parameterizations scheme they incorporated. But how good the model simulations compare with observations?

Reply: Explanation with respect to my above comment looks fine.

Many thanks for the reviewer's suggestion. We tried our best to revise the manuscript.

5. In Figures 2c and 3c authors represented it as the IT-generated turbulent mixing scheme based on Exp-3 but in this experiment, NBSW is also included, then how can it be an IT generated turbulent mixing scheme?

Reply: Explanation with respect to my above comment looks fine.

Many thanks for the reviewer's suggestion. We tried our best to revise the manuscript.

6. In Figures 2 and 3 for the vertical profiles of the monthly mean vertical temperature diffusive terms, the author choose to show the results for 10.5 °S, and for the temperature comparison, they showed 30.5 °S. What is the physical basis to choose these sections? Authors should show such results for the Arabian Sea and Bay of Bengal

as well.

Reply: Partially responded. If the authors compared it for Arabian Sea and Bay of Bengal, then why they did not gave the figures?

Following the reviewer's comments, the simulated temperature structure in the Arabian Sea and the Bay of Bengal is compared with the WOA13 data. Three transects, including 11.5° N, 15.5° N and 19.5° N, are chosen to show the vertical distribution of the difference of the temperature (Figures 13 – 18). In July (Figures 14, 16 and 18), the NBSW and IT can improve the simulation obviously in the Arabian Sea, but do not work in the Bay of Bengal, especially the difference for Exp 3 (NBSW+IT) or Exp 5 (NBSW+IT+WTFR) became larger in the Bay of Bengal, which is a hot spot for generation of the ITs. This implies that the IT-induced mixing scheme may not be appropriate in the marginal sea simulation that is full of small- and meso-scale processes. In January (Figures 13, 15 and 17), the NBSW and the IT even have a negative effects on the modeling. In summary, the simulated temperature structure along three transects, which are located in the Arabian Sea and the Bay of Bengal, can be improved by the NBSW and IT partially, further test based on high-resolution and real-time experiments will be implemented in future. As we mentioned above, we attempt to design the numerical modeling experiments in the Arabian Sea and the Bay of Bengal only (or the North Indian Ocean), as well as the finer simulation of the surface waves and more accurate estimation of the ITs.

[Figure]

Figure 13. The vertical temperature profiles along 11.5° N in January. (a) The temperature structure from the monthly WOA13 data (units: °C). (b) - (f) The difference of the temperature calculated by subtracting the monthly mean results simulated in Exp 1 - Exp 5 from the monthly WOA13 data, respectively. The RMSE of the temperature in the upper-100 m regions between the WOA13 data and the model results are given. Deep yellow areas correspond to the lands

[Figure]

Figure 14. The same as Fig. 13, but in July

[Figure]

Figure 15. The same as Fig. 13, but along 15.5° N

[Figure]

Figure 16. The same as Fig. 13, but along 15.5° N and in July

[Figure]

Figure 17. The same as Fig. 13, but along 19.5° N

[Figure]

Figure 18. The same as Fig. 13, but along 19.5° N and in July

*The discussion has been added in Line 660-664 of the revised manuscript.*

7. In Figure 4 in exp1& 4 why the model does show the cooler temperature in the thermocline depth region? In general, over the Indian Ocean, almost all forced model shows warm bias (Rahaman et al. 2020). Although the thermocline bias was reduced in Exp 2 and 3, it became reversed with similar magnitude why does it so? Why there is no difference between exp-1 and exp-4 in Figures 4 and 5? Does it mean WTFR does not impact temperature simulations? Authors should show such results for the Arabian Sea and Bay of Bengal as well.

Reply: The reason given for cooler thermocline temp in Exp-1 as " The reason for the cooler temperature in the thermocline depth region in Exp 1 should be that the multi-year monthly mean surface forcing fields were smaller than the actual values, which leads to insufficient heat transfer from the atmosphere to the ocean. After 10 climatologic years modeling, the temperature in the ocean interior became cooler obviously than the WOA13 data." This explanation does not show any scientific argument. Not fully convinced. How does multiyear monthly mean surface forcing

fields were smaller than the actual values will affect the simulation? The author made the forcing climatology I guess from 1948-2021. Similarly the explanation why the bias reversed in Exp-2 with respect to Exp-1 is also not complete. Author should give scientific evidence may be the complete heat budget for the explanation.

We agree with the reviewer, the explanation has been revised. The simulated temperature along 30.5° S is generally cooler than the WOA13 data in the thermocline depth region (see Figures 4 in the manuscript). We think that in addition to the explanation mentioned in the last response, the comparisons between the Levitus94 and WOA13 data (Figures 5 and 6) can give another better answer. The 30.5° S transect is near the ACC region, in which the Levitus94 temperature is cooler about 3℃ than the WOA13 temperature from surface to 200m depth layers, and warmer about 0.5℃ in the NIO and tropics when the depth is deeper than 200 m. The comparisons of the temperature in the upper-300 layers (Figures 1 – 3) indicate that, although the simulated temperature is cooler than the WOA13 data along 30.5° S, the simulated temperature become warmer than the WOA13 data along 0.5° S and 7.5° N, inaccurate simulation of the meridional overturning circulation causes abnormal heat distribution (heat accumulates in the north of the IO and tropic region).

When the NBSW- and IT-induced mixing schemes were adopted, the vertical mixing was enhanced and carried the surface warm water (see upper-50 m regions in Figure 4b of the manuscript) to mix together with the cooler water below, so more heat entered into the ocean interior and the SST became cool. The climatologic surface heat fluxes used in these experiments are often small because of smoothing, the Haney method will 'bring' sufficient heat fluxes to make the SST maintain at a normal magnitude.

We agree that the complete heat budget should be analyzed to understand the mechanism, but it is a little inappropriate for the ocean modeling only, because the heat fluxes are not strictly conserved (the climatologic reanalysis product are used and adjusted by the Haney method). The new experiments will be carried out based on the atmosphere-ocean coupling models (also including current-tide-wave coupling) in future, we believe that the heat budget analysis must be one of the most important tasks.

8. Figure 8 What is the physical basis of choosing the different zone? Looks like the present defined zones will not give true representation, for example in zone 1 since the dynamics and thermodynamics are different in the Arabian Sea, Bay of Bengal and South China Sea, hence the mixing characteristics are also different. I suggest excluding the regions outside of the Indian Ocean such as South China Sea and Atlantic Ocean as included in the present zone 2 and zone 3. I also suggest the author should select the zones based on past studies or based on the dynamics and thermodynamic properties of the Indian Ocean basin.

Reply: In that case there should be a zone in the equatorial region. It will not be good to include ITF in zone 2, zone 2 must be divided into two zones.

Many thanks for the reviewer's suggestion. The ITF forms into a narrow westward flow centered at about 12° S, within the South Equatorial Current (SEC), when it enters into the IO. The SEC carries the ITF waters westward across the IO. There is a complete cyclonic circulation system between the equator and 20° S, consisting of the westward SEC on the south side, the eastward South Equatorial Countercurrent (SECC) on the north side, and a northward western boundary current (East African Coastal Current; EACC). Furthermore, the effects of the $M_2$ internal tides are produced throughout the whole west region in Zone 2 (northern regions around the Madagascar Island). Therefore, in our opinion, the zone partitioning in this study should be appropriate.

*The explanations have been added in Line 411-414 of the revised manuscript.*

9. How the RMSE is statistically robust when the authors used the seasonal cycle and computed the RMSE?

10. As already pointed out in the case of the thermocline in the MLD bias given in Figure 9 for Exp-1 too looks not consistent with the previous works. In general OGCMs simulates deeper MLD in the Indian Ocean (de Boyer Montégut et al. 2007). A very recent study by Pottapinjara et al. (2022) too shows similar results. Hence, how the MLD simulation, in this case, shows shallower than observations? The authors need to

explain why the model simulated MLD is shallower as compared to observations. Also, the criteria used to compute MLD is not very widely used. The authors did not provide any reference to compute MLD or any explanation why they choose the 1 °C criterion to compute MLD.

Reply:"........ In fact, from Figures 10 one can see that, the obviously shallower MLDs are generally in the Antarctic Circumpolar Current (ACC) regions, where the simulated vertical mixing from the original experiment is weak dramatically." What about the Indian Ocean particularly in the Arabian Sea and Bay of Bengal.

In addition to the ACC regions, the obviously shallower MLDs also appear in the east regions of the Arabian Sea because of the weak vertical mixing. Furthermore, the simulated MLDs in most of the tropical and southern regions of the IO are shallower partially than the WOA13 data.

*The analysis has been added in Line 553-556 of the revised manuscript.*